# Do LLMs Recognize Your Preferences? Evaluating Personalized Preference Following in LLMs

**Siyan Zhao**[2*], **Mingyi Hong**[1,3], **Yang Liu**[1], **Devamanyu Hazarika**[1], **Kaixiang Lin**[1] [†]

[1]Amazon AGI, [2]UCLA, [3]University of Minnesota

`siyanz@cs.ucla.edu, mhong@umn.edu, devamanyu@u.nus.edu`
`{yangliud, kaixianl}@amazon.com`

## ABSTRACT

Large Language Models (LLMs) are increasingly used as chatbots, yet their ability to personalize responses to user preferences remains limited. We introduce PREFEVAL, a benchmark for evaluating LLMs' ability to infer, memorize and adhere to user preferences in a long-context conversational setting. PREFEVAL comprises 3,000 manually curated user preference and query pairs spanning 20 topics. PREFEVAL contains user personalization or preference information in both explicit and implicit forms, and evaluates LLM performance using a generation and a classification task. With PREFEVAL, we evaluated the aforementioned preference following capabilities of 10 open-source and proprietary LLMs in multi-session conversations with varying context lengths up to 100k tokens. We benchmark with various prompting, iterative feedback, and retrieval-augmented generation methods. Our benchmarking effort reveals that state-of-the-art LLMs face significant challenges in proactively following users' preferences during conversations. In particular, in zero-shot settings, preference following accuracy falls below 10% at merely 10 turns (∼3k tokens) across most evaluated models. Even with advanced prompting and retrieval methods, preference following still deteriorates in long-context conversations. Furthermore, we show that fine-tuning on PREFEVAL significantly improves performance. We believe PREFEVAL serves as a valuable resource for measuring, understanding, and enhancing LLMs' preference following abilities, paving the way for personalized conversational agents. Our code and dataset are available at https://prefeval.github.io/.

## 1 INTRODUCTION

The pursuit of personal chatbots that can remember your favorite cuisine, propose binge-worthy series tailored to your tastes, and avoid suggestions that conflict with your dietary restrictions has been a longstanding desire for many. While LLM-based chatbots such as Claude (Bai et al., 2022) and GPT-4 (Achiam et al., 2023) have substantially advanced natural language processing capabilities, their ability to proactively provide personalized interactions that are scalable to millions of users remains limited (Salemi et al., 2023; Li et al., 2023a; Jang et al., 2023; Tan et al., 2024; Liu et al., 2024a; Zhuang et al., 2024; Li et al., 2024a; Shaikh et al., 2024; Lee et al., 2024). For example, if a user says, *"I don't like jazz,"* and later asks for travel recommendations in New Orleans, a personalized chatbot should avoid suggesting jazz-related attractions, which are popular there. Achieving this level of proactive personalization poses a challenge when scaled to millions of users with diverse real-life preferences. Rather than building separate models for each, it's more scalable to create a single adaptable chatbot that can dynamically understand and accommodate these preferences in real-time. This brings us to the central evaluation goal of our benchmark:

*Can LLMs infer, remember and follow personalized preferences?*

This ability is crucial for user satisfaction and engagement during conversations. Current LLMs, however, are primarily optimized and evaluated for general-purpose tasks. Our study will reveal that we lack a comprehensive understanding of their ability to proactively follow and apply user

---

[*]Work done while as an intern at Amazon.

[†]Corresponds to: `siyanz@cs.ucla.edu, kaixianl@amazon.com`

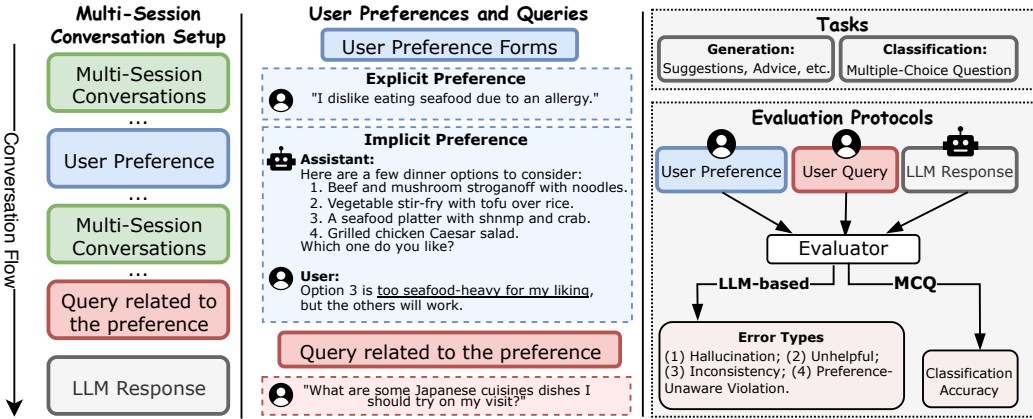

Figure 1: PREFEVAL setup overview. Key components from left to right: 1) **Multi-Session Conversation Setup**: PREFEVAL evaluates LLMs' ability to follow user preferences in multi-session conversation, challenging LLMs to handle preference inference, long-range retrieval, and context-aware preference following simultaneously. 2) **Preferences and Queries**: User preferences can be expressed through both explicit and implicit forms. Queries are designed such that a non-personalized answer would inadvertently conflict with user preferences, testing the LLM's adherence. 3) **Tasks and Evaluations**: PREFEVAL includes generation and classification tasks. Generation tasks are evaluated using an LLM-based evaluator to measure preference following accuracy and analyze error types. Classification tasks enable quicker evaluation through multiple-choice questions (MCQ). The two tasks' performances are highly correlated as shown in Fig 11.

preferences over conversations. Addressing this gap is essential for advancing LLMs toward truly personalized and scalable conversational agents.

Effective personalization in conversation settings also requires robust long-context abilities, as it requires aggregating and adhering to user preferences over extended interaction histories. A personal assistant should proactively infer, memorize, and adhere to these preferences over long horizon, ensuring responses are not only relevant but also aligned with the user's preferences. Additionally, users express preferences in various forms—explicitly or implicitly—making it challenging for the assistant to recognize them accurately. Users may also present multiple or conflicting preferences within a single conversation. The assistant must navigate these nuances for delivering a truly personalized experience.

In response to these challenges, we introduce PREFEVAL, a benchmark to evaluate, understand, and improve LLMs' capacity for preference following in conversation settings. Our benchmark consists of 3,000 manually curated preference-query pairs across diverse daily life topics, incorporating preference forms which are explicitly stated or implicitly revealed, and it includes both generation and classification tasks. With PREFEVAL, we assess 10 open-source and proprietary LLMs with varying context lengths of up to 100k tokens using various methods, analyzing their adaptability to conflicting and multiple user preferences, and demonstrate how finetuning on this dataset enhances performance. Our contributions can be summarized as follows:

- We introduce a novel, comprehensive benchmark for evaluating LLMs' preference following capabilities in conversational contexts, encompassing 3,000 manually curated question-preference pairs across 20 topics and 3 preference forms.

- We conduct an extensive evaluation of 10 state-of-the-art LLMs, including Claude, Mistral, GPT4 and the LLaMA series, utilizing various context lengths and assessment methods such as prompting, iterative feedback (Bai et al., 2022), and retrieval-augmented generation (Lewis et al., 2020).

- Our benchmark results show that, without explicit prompting, the preference following precision (evaluated using various measures we developed in this work) falls below 10% in zero-shot settings for 10-turn conversations of 3k tokens. Even with more advanced methods, performance still deteriorates with longer contexts.

- We uncover critical limitations in current LLMs through extensive error analysis, including their inability to recognize and proactively apply user preferences in long-context conversation settings.

- We find that, counterintuitively, multiple stated preferences within a conversation lead to improved adherence, even in the presence of conflicting preferences. We conjecture it is due to reinforced attention on user preferences. Moreover, fine-tuning on PREFEVAL further enhances preference following and generalizes well to longer contexts.

## 2 THE PREFEVAL DATASET

### 2.1 PROBLEM FORMULATION

The goal of our benchmark is to assess how well LLMs can personalize their responses to user preferences in conversational settings, with the overall setup shown in Figure 1. Let $\mathcal{C}$ denote a conversation comprising $m$ turns $\{(u_1, b_1), (u_2, b_2), \ldots, (u_m, b_m)\}$, where $u_i$ and $b_i$ represent the user's message and LLM's response at turn $i$, respectively. Each conversation $\mathcal{C}$ is divided into sessions $\mathcal{S} = \{s_1, s_2, \ldots, s_k\}$, where each session $s$ represents a coherent dialogue segment focused on a specific topic and consists of contiguous turns $s = \{(u_i, b_i), \ldots, (u_{i+l}, b_{i+l})\} \subseteq \mathcal{C}$.

At the core of our evaluation are *preference-query pairs* $(p, q)$. Here, $p$ refers to a user preference disclosed at some point in the conversation, while $q$ represents a query related to this preference, posed at the end of the conversation for the LLM to respond. The user preference $p$ can be explicitly expressed as a single utterance or implicitly revealed through a multi-turn dialogue. The query $q$ is the user's message at turn $m$, i.e., $q = u_m$, and the LLM's response is $b_m$. The query $q$ is constructed such that a generic, non-personalized response would likely violate the previously stated preference $p$. We evaluate whether $b_m$ adheres to the preference $p$. To simulate real-world conversational complexity, we include unrelated contextual turns between $p$ and $q$. These intervening turns act as potential distractions, emulating the natural flow of dialogue, where multiple topics may be discussed across a single conversation. In such cases, maintaining awareness of the user's earlier preference $p$ becomes crucial for appropriately responding to the later query $q$. We evaluate the LLM's ability to navigate this complexity and maintain personalization, measuring how well it responds to $q$ in line with the user's preference $p$, even amidst unrelated sessions.

To perform well on our benchmark, LLMs should demonstrate four key capabilities: (1) **Preference Inference**—the capacity to accurately infer user preferences through dialogue, whether explicitly stated or implicitly revealed; (2) **Long-Context Retrieval**—the ability to track and recall user preferences across long conversation; (3) **Preference Following**—the ability to generate responses that are both contextually relevant and aligned with the user's preferences when knowing the preference; and (4) **Personalization Proactiveness**—the initiative and ability to know when and how to utilize the first three capabilities to deliver personalized responses, rather than focusing solely on general question answering. We will show in our benchmarking results that, LLMs that miss a subset of these capabilities perform poorly.

### 2.2 PREFEVAL STATISTICS

PREFEVAL consists of 1,000 unique preference-query pairs, each with three preference forms (§2.3), resulting in 3000 preference-query pairs. These pairs were manually curated with the assistance of GPT-4, Claude 3 Sonnet, and Claude 3.5 Sonnet (see Appendix A.14 for detailed data construction methodology). The preferences cover day-to-day topics such as travel, shopping, entertainment, and more, as shown in the topic distribution in Figure 2. We intersperse unrelated contextual conversation turns between the disclosure of a user preference and the final query, with context lengths extending up to 100k tokens (§2.4). For each pair $(p, q)$, we consider two tasks: a generation task and a classification task. In the generation task, the LLM is required to generate a long-form response to the user's query. In the classification task, the LLM is presented with four options related to the query, with one option aligned with the user's preference, and the LLM is tasked to select the correct option.

### 2.3 PREFERENCE FORMS

User preferences can be expressed in various forms. In our benchmark, we consider three distinct methods of preference construction and expression: (1) **Explicit Preference.** The user directly expresses their preference to the LLM in a single sentence within a single conversational turn (Examples in Table 14). (2) **Implicit Choice-Based Dialogue.** The user's preference is inferred over the course of a two-turn dialogue. In this setup, the user initiates a preference-related query, and the

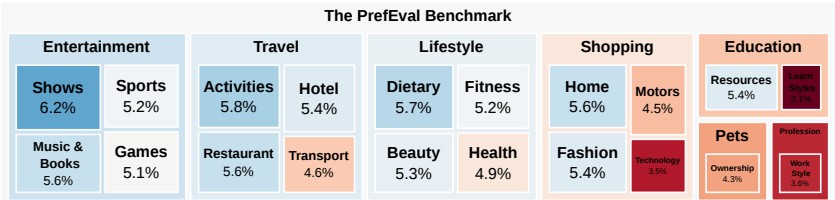

Figure 2: Distribution of domains and topics within PREFEVAL, which are commonly encountered during conversations with chatbots where users seek recommendations, suggestions, and advice.

assistant presents multiple options, some of which violate the user's preference while others align with it. The user can either agree with or reject one or more options, implicitly revealing their preference through these choices (Examples in Table 15). (3) **Implicit Persona-Driven Dialogue.** To simulate more nuanced, implicit preferences, this form of elicitation unfolds over 4–8 turns. The dialogue primarily revolves around a persona-driven topic, with a randomly assigned persona guiding the conversation. The user's preference is subtly revealed in a single sentence during the dialogue, while it is not the main focus of the conversation, making the inference and reasoning process more challenging (Examples in Table 16).

## 2.4 MULTI-SESSION CONVERSATIONAL CONTEXT

To simulate realistic conversational dynamics, we incorporate multi-session turns from the *LMSYS-Chat-1M* dataset (Zheng et al., 2023), consisting of one million real interactions between users and 25 state-of-the-art language models across various topics. We randomly select multi-session context up to a length of 100k tokens and intersperse these conversation turns between the disclosure of a user preference and the final query. This presents a challenge for the LLM to accurately infer, retain, and retrieve user preferences while navigating unrelated dialogues, assessing the LLM's ability to handle long-context preference following.

## 2.5 TASK TYPES AND EVALUATION PROTOCOLS

We offer two task types for each preference-query pair, each with its corresponding evaluation protocol. By including both generation and classification tasks, we aim to thoroughly assess LLMs' ability to understand and adhere to user preferences in diverse contexts.

**Generation Task and LLM-based Evaluators.** In the generation task, the LLM generates a response in reply to the user's query. To evaluate preference following in the generation task, we apply the "LLM-as-a-judge" framework using Claude 3 Sonnet. Specifically, we employ four independent evaluators to check the response against four binary metrics. Each evaluator is provided with a detailed prompt (see Appendix Sec A.12) containing the definitions and examples for each metric. These checks are then aggregated into four distinct error types, following the rules outlined in Table 1. Preference-following accuracy is defined as the absence of *any* error type in the generated response. The four error types are: (1) **Preference-Unaware Violation:** The LLM provides generic recommendations that contradict the user's stated preference due to unawareness of user preference. (2) **Preference Hallucination Violation:** The response fabricates or misattributes preferences, diverging from the user's true preference and violates the true preference. (3) **Inconsistent Violation:** The response acknowledges the correct preference but generates contradicting response. (4) **Unhelpful Response:** The response lacks relevant recommendations or fails to address the query due to poor recall of the user's preference. To validate our LLM-based evaluation method, we manually checked 200 randomly sampled evaluations, with an observed 5% error rate. This demonstrates strong agreement between human judgment and LLM-based assessments with Claude 3 Sonnet.

**Classification Task and MCQ Accuracy.** In the classification task, the user presents a final query along with four potential options and asks the LLM to select the one that aligns with their preference. Only one option follows the user's stated preference, while the remaining options conflict with it. The LLM's preference-following accuracy is determined by whether it selects the correct option. This task facilitates faster automatic evaluation, eliminating the need for costly human or LLM-based assessments by focusing on a single-choice response.

Table 1: Error type aggregation rules: Evaluators perform binary checks for each metric listed in the column headers. The results are then aggregated according to the table rules to classify the error into one of the four defined types.

| Error Type | Violate Preference? | Acknowledge Preference? | Hallucinate Preference? | Helpful Response? |
|---|---|---|---|---|
| Preference-Unaware Violation | Yes | No | N/A | Yes |
| Preference Hallucination Violation | Yes | Yes | Yes | Yes |
| Inconsistency Violation | Yes | Yes | No | Yes |
| Unhelpful Response | No | Yes/No | N/A | No |

## 3 EXPERIMENTS

Our experiments encompassed a comprehensive evaluation and analysis of both open-source and proprietary state-of-the-art LLMs across multiple dimensions, including preference forms, task types, various methods to improve preference following, an examination of how LLMs adapt to multiple and dynamically changing preferences, and fine-tuning open-source models to enhance preference following. We aim to investigate the following questions: (1) What is the performance of current SOTA LLMs in multi-session long-context preference following with zero-shot, prompting and RAG baselines? (§3.2, §3.3, §3.4). (2) How do LLMs performance change with different forms of user preference expressions? (§3.3). (3) What causes LLMs to fail in preference following, and what are the prevalent error types? (§3.5). (4) How capable are LLMs in simultaneously accommodating multiple user preferences and adapting to dynamically changing user preferences? (§3.6). (5) How does finetuning on PREFEVAL improve the preference adherence ability of LLMs? (§3.7)

### 3.1 MODELS AND METHODS

We extensively evaluate a variety of state-of-the-art LLMs, including Claude 3 Sonnet, Claude 3 Haiku, Mistral 7b Instruct, Mistral 8x7b Instruct, LLaMA 3 8b Instruct, and LLaMA 3 70b Instruct. We also assess more recent models Claude 3.5 Sonnet, GPT-o1-preview, and Gemini-1.5-pro in specific settings. We investigate methods to explicitly help LLMs focus on the preference-following task, with the aim of understanding if they can proactively adapt to user preferences: (1) **Zero-shot:** The default case, where the LLM directly answers the question without any additional prompting. (2) **Reminder:** Before answering the question, the LLM is provided with a reminder sentence to consider the user's previously stated preference in answering. (3) **Self-Critic:** The LLM generates an initial *zero-shot* response to the question, critiques whether it has followed the user's preference, and then generates a revised response taking the critique into account, similar to *intrinsic self-correction* (Huang et al.). (4) **Few-Shot Chain-of-Thought (CoT):** The LLM is given few-shot examples of CoT reasoning of how to follow the user's preference right before answering the question. (5) **Retrieval-Augmented Generation (RAG):** A sentence embedding model is used to retrieve the most similar conversation exchanges to the question, which are then provided to the LLM in the prompt. Please refer to §A.4 for detailed prompt examples for each method. For the experiments, if not explicitly stated otherwise, we place the user's preference at the beginning of the conversation and query at the end.

### 3.2 EXPLICIT PREFERENCE FOLLOWING

**SOTA LLMs exhibit limited proactivity in adhering to user preferences.** As shown in the generation task performance in Figure 3, all LLMs exhibit substantial declines in preference-following accuracy as the dialogue length increases between the user's stated preferences and the final queries in zero-shot settings—where no specific prompting is provided. Accuracy drops steeply from approximately 80% to below 30% as the number of conversation turns increases to merely 5. As the turns extend from 30 to 300, accuracy falls close to zero across all models. This suggests that LLMs often provide recommendations that conflict with the user's previously stated preferences, even when expressed only a few turns earlier. Such behavior is detrimental to user satisfaction and engagement. Additionally, more advanced LLMs continue to struggle with preference following in zero-shot settings, even within the first 10 turns. As shown in Table 2, GPT-o1 has 50% accuracy while Claude 3.5 Sonnet and Gemini 1.5 Pro have near-zero preference following.

Table 2: **Preference-following accuracy** across SOTA LLMs on the *travel restaurant* generation task. Accuracy is evaluated using two context lengths between the user's preference and query. We show results for Zero-shot and Reminder (best prompting method). The o-series models may involve additional test-time compute due to their "thinking" phase.

| | 10 Turns / ~3k tokens | | 300 Turns / ~103k tokens | |
|---|---|---|---|---|
| | **Zero-shot** | **Reminder** | **Zero-shot** | **Reminder** |
| Claude-3.5-Sonnet | 0.07 | 0.45 | 0.02 | 0.02 |
| Gemini-1.5-Pro | 0.07 | 0.91 | 0.09 | 0.05 |
| GPT-4o | 0.07 | 0.98 | 0.05 | 0.23 |
| GPT-4.1 | 0.50 | 0.98 | 0.11 | 0.70 |
| o1-preview | 0.50 | 0.98 | **0.14** | **0.98** |
| o4-mini | 0.23 | 0.98 | 0.02 | 0.54 |
| o3 | **0.71** | **1.0** | 0.05 | 0.43 |

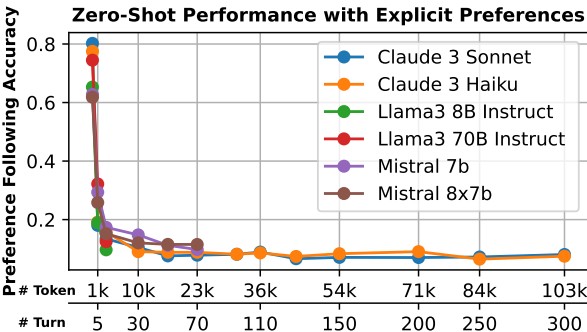

Figure 3: Zero-shot performance of LLMs with explicit preferences, averaged across 20 topics. The x-axis represents the dialogue length between the user's stated preference and the final query, measured by both the number of tokens in the prompt and the number of conversation turns. All LLMs exhibit a rapid decline in accuracy as the number of turns increases.

**How do different methods improve preference following performance?** LLMs lack the proactiveness to follow user preference in the zero-shot setting. We evaluated four methods for enhancing preference following across models (§3.1) that force the LLMs to utilize their abilities in *Preference Inference*, *Long-Context Retrieval* and *Preference Following*. As shown in Figure 4, all methods outperformed the zero-shot baseline. Among prompting techniques, the Reminder method, which simply reminds the model to consider the user's previously stated preferences (see prompt in §A.4), surprisingly outperformed more complex methods such as Self-Critic and CoT. Interestingly, performance trends varied across models: in Claude models, Self-Critic initially outperformed CoT but fell behind as the number of conversation turns increased, whereas the reverse was observed for Llama models. RAG consistently performed the best across most models, indicating that adherence to user preferences may hinge on retrieval capabilities. However, in models like Claude 3 Sonnet and Mistral 8x7b, Reminder performed comparably or even surpassed RAG, suggesting that these models may have strong intrinsic *Long-Context Retrieval* abilities.

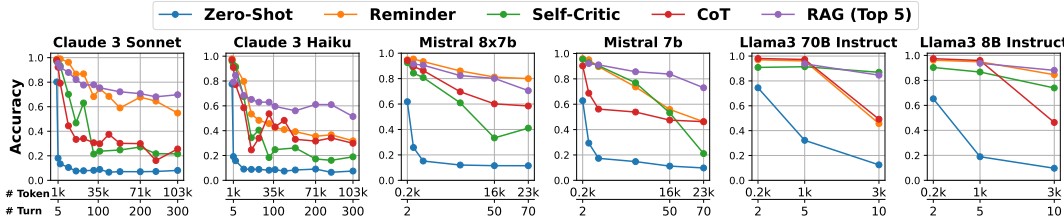

Figure 4: Performance comparison of 5 methods across 6 LLMs with explicit preferences on the generation task. Both Reminder and RAG consistently achieve the highest accuracy across models. Notably, Reminder outperforms more complex techniques such as Self-Critic and CoT.

## 3.3 IMPLICIT PREFERENCE FOLLOWING

**Implicit Preferences Add Complexity to Preference Inference.** The ability to infer preferences is especially critical when users implicitly reveal their preferences through a dialogue. PREFEVAL includes three forms of preferences, two of which are implicitly conveyed through conversation (§2.3). We evaluate how these different forms impact model performance using the best prompt-

---

[0]Note that GPT-o1-preview may not be a fair comparison with other models as it may require more test-time compute with a "thinking" phase.

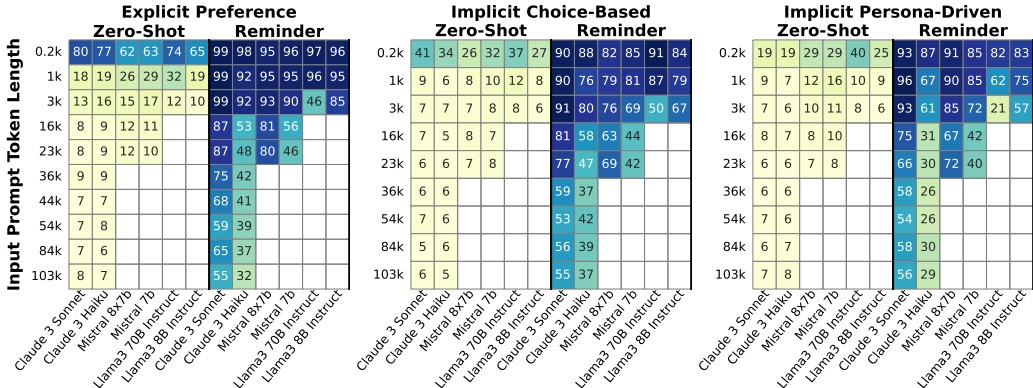

Figure 5: Comparison of 3 preference forms for 6 LLMs on the generation task, across varying lengths between the stated preference and query. Note that the user preference is stated in the first turn. Results show that implicit preferences are more challenging to infer than explicit preferences.

ing method—Reminder. As shown in Figure 5, implicit preferences add additional complexity to preference-following tasks, even with shorter input lengths. Models exhibit varying capabilities in handling these complexities: for instance, Claude and Llama struggle more with *implicit persona-driven* preferences compared to *implicit choice-based* preferences, while Mistral models show the opposite trend. These differences indicate that each model has unique strengths and weaknesses in inferring and processing distinct preference forms.

### 3.4 CLASSIFICATION TASK

**Classification Task Brings Fast Evaluation with High Correlation to Generation Task Results.**
We introduce a classification task for each preference-query pair to complement the generation task (§2.5). This reduces reliance on costly human evaluations and LLM-based evaluators. We task LLMs to select the aligned option based on user preferences and get classification accuracy. Results in Figure 6 show higher overall accuracy compared to the generation task, reflecting the classification task's simpler nature. The Self-Critic method, with Claude models and Mistral-8x7B, underperforms the zero-shot baseline. Upon examining their critiques and revisions, we observe that when models fail to retrieve user preferences, the revisions tend to hallucinate and select random options distinct from the initial response. We conjecture that iterative self-feedback mechanisms may be less effective for classification tasks (which require structured output) compared to generation tasks. While both RAG and Reminder consistently achieve high accuracy, RAG demonstrates more advantage over Reminder in the classification task compared to the generation task. To further understand the relationship between the generation and classification tasks, we perform a correlation analysis between LLM-based preference following accuracy and classification-based preference alignment results, as illustrated in Figure 11. Across six models, five methods, and 12 dialogue lengths between the preference and query, each data point in the scatter plot is averaged over 20 topics. We observe a correlation coefficient of 0.73, indicating a strong positive correlation between the two evaluation methodologies. This correlation suggests that despite task differences, models adhering to preferences in one setting perform similarly in the other. Classification tasks could thus serve as an efficient proxy for evaluating preference following in complex generation scenarios across models and methods.

### 3.5 ERROR TYPE ANALYSIS

**What causes LLMs to fail and what capabilities are missing in preference following?** LLM-based evaluations allow us to efficiently analyze error types defined in §2.5 and we analyze them with the key capabilities defined in §2.1. Figure 7 shows the distribution of error types across two LLMs and five methods at turn 10. In zero-shot settings, LLMs generally lack awareness of user preferences, leading to high *Preference-Unaware Violations*. With advanced methods, this error type percentage drops, indicating a lack of *Personalization Proactiveness* ability in the zero-shot setting. However, with these methods, *Inconsistency Violations* appears, indicating that even when preferences are correctly retrieved, LLMs struggle to generate aligned responses, lacking the *Preference*

**Classification Task Accuracy (Explicit Preference)**

| Input Prompt Token Length | Zero-Shot | | | | | | Reminder | | | | | | Self-Critic | | | | | | CoT | | | | | | RAG (Top 5) | | | | | |
|---|---|---|---|---|---|---|---|---|---|---|---|---|---|---|---|---|---|---|---|---|---|---|---|---|---|---|---|---|---|---|
| | Claude 3 Sonnet | Claude 3 Haiku | Mistral 8x7b | Mistral 7b | Llama3 70B Instruct | Llama3 8B Instruct | Claude 3 Sonnet | Claude 3 Haiku | Mistral 8x7b | Mistral 7b | Llama3 70B Instruct | Llama3 8B Instruct | Claude 3 Sonnet | Claude 3 Haiku | Mistral 8x7b | Mistral 7b | Llama3 70B Instruct | Llama3 8B Instruct | Claude 3 Sonnet | Claude 3 Haiku | Mistral 8x7b | Mistral 7b | Llama3 70B Instruct | Llama3 8B Instruct | Claude 3 Sonnet | Claude 3 Haiku | Mistral 8x7b | Mistral 7b | Llama3 70B Instruct | Llama3 8B Instruct |
| 0.2k | 95 | 92 | 91 | 87 | 90 | 85 | 98 | 98 | 96 | 95 | 98 | 94 | 91 | 87 | 73 | 75 | 90 | 79 | 96 | 96 | 94 | 90 | 95 | 83 | | | | | | |
| 1k | 80 | 66 | 78 | 66 | 70 | 55 | 98 | 95 | 94 | 85 | 95 | 80 | 85 | 70 | 60 | 65 | 86 | 73 | 93 | 88 | 87 | 79 | 86 | 67 | 98 | 98 | 95 | 87 | 96 | 87 |
| 3k | 74 | 59 | 59 | 57 | 48 | 42 | 96 | 92 | 84 | 75 | 76 | 59 | 77 | 58 | 54 | 58 | 57 | 62 | 90 | 83 | 79 | 73 | 69 | 50 | 97 | 97 | 91 | 86 | 95 | 80 |
| 10k | 64 | 51 | 50 | 50 | | | 93 | 85 | 67 | 63 | | | 59 | 39 | 43 | 51 | | | 81 | 75 | 65 | 69 | | | 92 | 93 | 84 | 81 | | |
| 16k | 58 | 51 | 51 | 42 | | | 88 | 79 | 68 | 56 | | | 44 | 32 | 47 | 52 | | | 75 | 68 | 63 | 56 | | | 89 | 91 | 82 | 80 | | |
| 23k | 57 | 48 | 50 | 47 | | | 90 | 80 | 67 | 56 | | | 53 | 33 | 49 | 48 | | | 82 | 75 | 69 | 62 | | | 88 | 88 | 80 | 75 | | |
| 31k | 50 | 45 | | | | | 79 | 69 | | | | | 41 | 27 | | | | | 74 | 66 | | | | | 83 | 84 | | | | |
| 36k | 55 | 44 | | | | | 85 | 74 | | | | | 43 | 32 | | | | | 77 | 66 | | | | | 84 | 85 | | | | |
| 54k | 48 | 41 | | | | | 74 | 64 | | | | | 34 | 28 | | | | | 72 | 64 | | | | | 82 | 80 | | | | |
| 71k | 53 | 46 | | | | | 82 | 70 | | | | | 40 | 32 | | | | | 76 | 66 | | | | | 80 | 81 | | | | |
| 84k | 53 | 46 | | | | | 82 | 76 | | | | | 37 | 28 | | | | | 75 | 68 | | | | | 81 | 81 | | | | |
| 103k | 49 | 43 | | | | | 74 | 68 | | | | | 37 | 26 | | | | | 71 | 65 | | | | | 78 | 74 | | | | |

Figure 6: Performance of the classification task across models and methods on the explicit preferences dataset across various input lengths between the user's preference and the final query. Results are averaged over 20 topics.

*Following* capability. Interestingly, while prompting methods are introduced to enhance preference following, they inadvertently incentivize LLMs to hallucinate preferences. RAG reduces the hallucination error percentage for Mistral 8x7b. Additionally, Claude 3 Sonnet exhibits a high percentage of *Unhelpful* errors with prompting methods, often refusing to answer queries due to a perceived lack of context regarding user preferences, which is undesirable for effective conversational personalization. We include further discussions on long-context error type analysis with more LLMs and how they scale with turns in Appendix A.11.

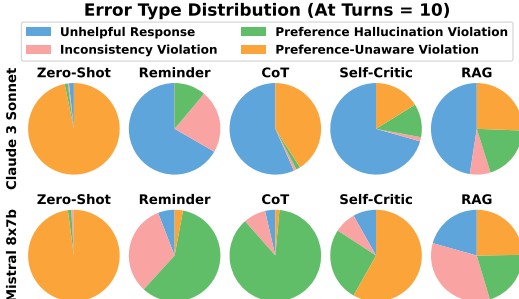

Figure 7: Distribution of 4 disjoint error types across 2 LLMs and 5 methods in the generation task with explicit preferences, where the gap between the preference and the query is 10 turns. Prompting methods introduce more hallucination and unhelpful errors, as LLMs either fabricate user preferences or refuse to respond due to failures in recalling the preferences, which is undesirable for personalized conversational chatbots.

## 3.6 DYNAMIC PREFERENCE FOLLOWING

In real-world scenarios, users may express multiple preferences within a single conversation, and these preferences can evolve or even conflict as the dialogue progresses across different sessions. We explore how the presence of multiple preferences and conflicting preferences affect the model's ability to maintain adherence.

**Impact of Multiple Preferences Stated in Conversation.** We investigated whether introducing non-conflicting preferences from different topics throughout a conversation would affect the model's ability to follow an initial preference. To evaluate this, we evenly inserted additional preferences at various points in the conversation and measured the model's adherence by asking a query related to the first preference. As shown in Figure 8, the results indicate that adherence accuracy for the initial preference increases as more preferences are introduced, even when these preferences are unrelated to the first. We conjecture that when more preferences are presented, the model may be implicitly encouraged to treat the cumulative set of preferences as a broader constraint on its outputs. Consequently, introducing multiple preferences may help the model pay more attention to each user-stated preference throughout the conversation, leading to improved personalization. **Effect of Conflicting Preferences Stated in Conversation.** We also analyzed how LLMs adapt to conflicting user preferences within a conversation. We generated conflicting preferences for the original preferences

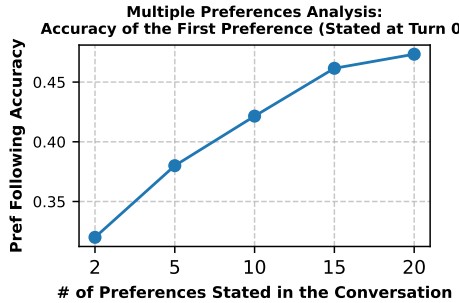

Figure 8: Introducing multiple preferences throughout a conversation improves adherence to the initial preference. Results are using Claude 3 Sonnet with Reminder prompting.

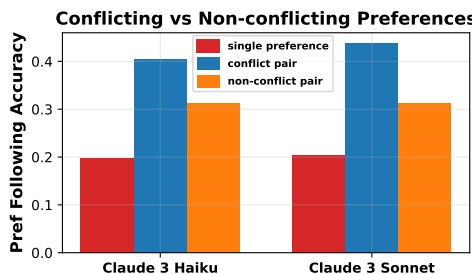

Figure 9: Effect of adding conflicting versus non-conflicting preferences on adherence. The red bar indicates the performance when only the original preference is present. Results are averaged over five topics using a fixed 100-turn conversation.

across five topics using Claude 3.5 Sonnet. These conflicting preferences were inserted alongside the original preferences at predetermined positions in the conversation, while the rest of the conversation was kept constant. For comparison, we also tested the impact of inserting non-conflicting preferences. In both settings, the original preferences were stated later than the conflicting one, and we will test the model's preference following to the original preference. As shown in Figure 9, when conflicting preferences are introduced, the model actually demonstrates improved adherence to the original preference compared to when non-conflicting preferences are added. Comparing the adherence performance to the baseline scenario where only the original preference is present (indicated by the red bar), both the conflicting and non-conflicting preference scenarios achieve better adherence than the single-preference scenario, reinforcing the observation from the previous section that introducing multiple preferences encourages the model to better retain and follow preferences. These findings suggest that conflicting preferences do not necessarily pose a challenge for the LLM but rather reinforce its capacity to track and adapt to evolving user preferences.

### 3.7 FINETUNING ON PREFEVAL TO IMPROVE PREFERENCE FOLLOWING

**Enhancing Preference Following through Supervised Fine-Tuning (SFT) on PREFEVAL**: We fine-tuned the Mistral-7B model using SFT on 80% of the topics in PREFEVAL and evaluated it on the remaining unseen 20% topics for the generation task. For SFT data, we used Mistral-7B's responses generated using the Reminder method, without any contextual turns inserted between the stated preference and the query. During training, to simulate conversational preference following, we inserted 0, 5, or 10 contextual turns between the preference and query, resulting in training data of 2, 7, or 12 turns (where the preference, query, and response constitute 2 turns). After fine-tuning, the model demonstrated significant improvement in the zero-shot setting, surpassing the previous best-performing method (RAG), as shown in Figure 10. One notable benefit is improved length generalization – the capability to follow preferences for longer contexts compared to training. When the model is trained with 10-turn contextual turns, it generalizes to 70-turn contexts much more effectively than when trained with fewer contexts. This suggests that SFT enhances both the ability to handle long-context retrieval and the capacity to infer and follow user preferences over extended conversations.

To understand the mechanisms behind this improvement, we analyzed the attention patterns before and after fine-tuning. Our analysis reveals that the SFT model consistently exhibits increased attention to preference-related information during generation, with improvements in preference region attention up to 4.97% across test examples. See Appendix A.13 for detailed attention score visualizations and analysis.

## 4 RELATED WORK

**LLM Personalization and Benchmarks.** Early personalization efforts focused on dialogue systems mimicking user styles (Zhang et al., 2018; Mazaré et al., 2018; Wu et al., 2021; Zhong et al., 2022) and tasks like news headline (Ao et al., 2021) and review generation (Li & Tuzhilin, 2019).

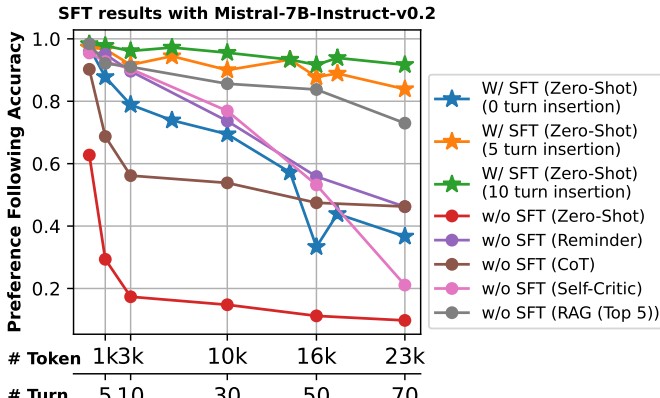

Figure 10: Performance on the 20% unseen test topics: After fine-tuning, the Mistral-7B model exhibits superior preference following accuracy compared to other baseline models. It also shows enhanced length generalization when trained with longer context interleavings, demonstrating its ability to handle extended conversational turns more effectively.

Recent LLM personalization benchmark works include LAMP (Salemi et al., 2023), which emphasizes explicit user profile conditioning through retrieval-augmented techniques, RPBench-Auto's character-based role-playing tasks (Boson AI, 2024), TIMECHARA's temporal consistency in character representation (Ahn et al., 2024), and RoleLLM's fine-grained role-playing framework (Wang et al., 2024b). While these often focus on stylistic preferences or single-turn tasks (Lee et al., 2024; Li et al., 2024a; Jang et al., 2023; Zhao et al., 2023), our work addresses lifestyle preferences and extend to long-context, multi-turn conversations. **Long Context LLM and Benchmarks.** With extended context windows (Reid et al., 2024), long-context LLMs have emerged (Agarwal et al., 2024; Bertsch et al., 2024). Existing benchmarks primarily evaluate information retrieval capabilities through tasks requiring models to locate specific facts or answers (Zhang et al., 2024; Wang et al., 2024a; An et al., 2023; Li et al., 2023b), including tasks like question-answering, retrieval, fact reasoning, and coding (An et al., 2023; Bai et al., 2023; Kočiský et al., 2018; Dasigi et al., 2021; Huang et al., 2021; Li et al., 2024b; Kuratov et al., 2024). While these "needle-in-a-haystack" tasks test a model's ability to identify and extract relevant information, our benchmark introduces a distinct challenge of preference following, where models need to infer from implicit preferences and dynamically apply this understanding across conversation contexts rather than simply retrieving explicit preferences. **Instruction Following.** Fine-tuning on human-annotated instruction-response pairs has enhanced LLMs' instruction-following capabilities, as seen in work like InstructGPT (Ouyang et al., 2022). These models perform a broad range of tasks, including summarization, translation, and problem-solving (Zhong et al., 2024; Zhou et al., 2023). While existing benchmarks focus on executing discrete instructions, our work extends this paradigm by emphasizing the inference and adherence to user preferences across multiple conversation turns. Importantly, we also assess *Personalization Proactiveness*, which is the initiative and ability to know when and where to apply user preferences, moving beyond discrete task execution. Due to space constraints, we provide a more comprehensive discussion of Related Work in Section A.2.

## 5 CONCLUSION

In this work, we present PREFEVAL, a comprehensive benchmark addressing a critical gap in evaluating large language models' ability to follow user preferences in multi-session conversational settings. Our benchmark considers comprehensive aspects, including explicit and implicit preferences, both generation and classification tasks, and employs LLM-based and automatic evaluation methods. Through rigorous testing of 10 state-of-the-art LLMs across 20 diverse topics and various conversation lengths up to 100k, we demonstrate that preference following remains a significant challenge. Our findings reveal that even advanced models struggle to maintain adherence with user preferences in conversational setting, with accuracy dropping below 10% in default settings for conversations exceeding 10 turns. These models struggle to proactively recall and incorporate user preferences stated earlier in conversations without explicit prompting. Implicit preferences create further difficulties for LLMs to infer user preferences. While prompting techniques, such as reminders, show promise in mitigating this performance drop, substantial room for improvement still remains. PREFEVAL not only highlights the current limitations of LLMs in personalized interaction but also provides a valuable resource for researchers and developers to evaluate and enhance the personalization capabilities of conversational AI systems.

# 6 ACKNOWLEDGMENT

M. Hong's work is partially supported by the NSF grant CIF-2414372.

# 7 REPRODUCIBILITY STATEMENT

Our work presents a comprehensive benchmark that includes a manually curated dataset to evaluate current open-source and proprietary models (see Section 3.1), with detailed descriptions of the model versions provided in Table 3. We plan to release our benchmark in the future, enabling others to reproduce our results. Additionally, we will make available the contextual turns from the Lmsys-1M-dataset used to construct inter-turn distractors. To ensure reproducibility, we also provide the prompts used for LLM-based evaluators (Claude 3 Sonnet) in Section A.12, which are critical for obtaining performance results, as well as the detailed method descriptions and their prompts used in our experiments (Section A.4). These resources will facilitate further research and allow for replication of our work.

# 8 ETHICS STATEMENT

In this paper, we introduce PREFEVAL, a benchmark designed to evaluate large language models' ability to infer, memorize, and adhere to user preferences in long-context multi-session conversational settings. Our research prioritizes responsible and ethical practices, particularly concerning data privacy, data quality in terms of ethics, bias mitigation, and research integrity. Our dataset consists of manually curated preference-query pairs spanning 20 topics, generated with assistance from AI language models. In the construction process, we have invested significant effort in manually rating and filtering these pairs to ensure quality and relevance, removing any potentially unethical preference instances. We also examined inter-conversation data from the LMSYS dataset, which contains anonymized interactions between users and language models, and removed problematic conversations from our experiments. Throughout this process, we maintained strict privacy standards, ensuring no sensitive or personal information was collected or used. In the benchmarking process, we continuously examined the LLM's output to ensure no preference pair led to unethical responses, and we have detailed the API versions of the LLMs we benchmarked for reproducibility. In terms of future deployment, the enhancement of an LLM's ability to remember and follow user preferences might raise privacy considerations. We advocate for responsible deployment practices that protect user data. We acknowledge that LLMs may inherit biases from their training data, potentially leading to unfair or discriminatory outputs. To address this concern, we aimed to optimize the diversity of the topics in our dataset to minimize potential biases. All results presented accurately represent our findings, supported by detailed documentation of our methodologies, such as the prompts we used. We conducted all experiments using either publicly available models or through documented commercial API access. To promote reproducibility and advance research in this field, we will make our benchmark dataset publicly available.

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

# A  APPENDIX

## A.1  LIMITATION

Our benchmark aims to evaluate preference following rather than verifying the factual accuracy of the recommendations. While the LLM's recommendations may contain inaccurate information, fact-checking is a separate dimension beyond the scope of this work. Additionally, although we have conducted extensive human filtering on our synthetic preference-query pairs, incorporating real user preferences in future work would help capture more nuanced aspects of user interactions.

For implicit preference elicitation, we designed the available options such that only one option adheres to or violates the target preference. While this approach yields a high probability of inferring the user's preference from their choice, it does not guarantee 100% accurate inference from the multiple-choice selection. However, given the early turns' results of a relatively high preference-following accuracy of the strongest baseline is around 90%, we believe the errors should be within an acceptable range.

## A.2  DETAILED RELATED WORKS

**LLM Personalization and Benchmarks.** Prior to the era of LLMs, personalization for language models mainly focused on personalized dialogue systems. These systems conditioned agents on user profiles, such as Reddit posting histories, to generate more engaging chit-chats that mimicked users' personalities or styles (Zhang et al., 2018; Mazaré et al., 2018; Wu et al., 2021; Zhong et al., 2022). Other personalization tasks included news headline generation (Ao et al., 2021) and review generation (Li & Tuzhilin, 2019). With the rise of LLMs, the scope of personalization tasks has broadened. Recent works aim to personalize the LLMs themselves to embody particular personas and mimic speaking styles (Vincent et al., 2023). In recent work on personalization benchmarks, Salemi et al. (2023) introduced the LAMP benchmark, which tests LLMs' ability to produce personalized output in specific tasks such as movie tagging and email subject generation and emphasizes explicit user profile conditioning through retrieval-augmented techniques. The RPBench-Auto benchmark (Boson AI, 2024) focuses on character-based and scene-based role-playing tasks, evaluating models' abilities to maintain consistent personas across 80 unique characters in free-form conversations and structured narrative scenarios. TIMECHARA benchmark (Ahn et al., 2024) specifically addresses point-in-time character representation, examining how well models maintain temporal consistency in narratives without revealing future events or contradicting established character timelines. RoleLLM benchmark (Wang et al., 2024b) introduces a framework for fine-grained role-playing across 100 diverse characters, emphasizing speaking style imitation and role-specific knowledge capture through systematic instruction tuning. Beyond benchmarking efforts, researchers have explored various personalization methods, Bao et al. (2023) worked on fine-tuning LLMs for recommending items using users' past interactions. Jang et al. (2023) aimed to align LLMs with multi-preferences that can be combined post-training through parameter merging. Zhao et al. (2023) considered few-shot adaptation of LLMs to cater to human group preferences across demographics. However, these recent works mostly consider preferences about LLM response stylistic attributes such as conciseness or informativeness (Lee et al., 2024; Li et al., 2024a) or focus on single-turn tasks like email title generation. Our work differs from these benchmarks by focusing on adherence to more lifestyle-oriented, day-to-day user preferences and we extend to long-context, multi-turn conversations.

**Long Context LLM and Benchmarks.** Recent LLMs such as Gemini have extended the context window to millions of tokens (Reid et al., 2024), enabling researchers to expand from few-shot to many-shot settings (Agarwal et al., 2024; Bertsch et al., 2024). To keep pace with increasing context length capabilities, new datasets and benchmarks have been proposed to evaluate long-context reasoning abilities, primarily focusing on question-answering and summarization tasks (Zhang et al., 2024; Wang et al., 2024a; An et al., 2023; Li et al., 2023b). For instance, L-eval (An et al., 2023) and LongBench (Bai et al., 2023) are among the first to aggregate existing benchmarks such as Kočiský et al. (2018); Dasigi et al. (2021); Huang et al. (2021) into long-context benchmarks, spanning tasks like question-answering, summarization, retrieval, and coding. Kuratov et al. (2024) designed a benchmark to assess LLMs' fact reasoning abilities across extremely long documents. LongICLBench (Li et al., 2024b) evaluates long in-context learning in extreme-label classification tasks, where the model needs to comprehend the entire context to understand the label space. While

these "needle-in-a-haystack" tasks test a model's ability to identify and extract relevant information, our benchmark introduces a distinct challenge of preference following, where models need to infer from implicit preferences and dynamically apply this understanding across conversation contexts rather than simply retrieving explicit preferences. Our benchmark evaluates LLMs' long-context retrieval capabilities in a more realistic and practical setting by assessing preference-following across multi-turn conversations. It demands a comprehensive understanding of the conversational flow, enabling models to accurately infer user preferences as they evolve over dialogue and to know when and where to apply these preferences in responses.

**Instruction Following.** Recent LLMs are fine-tuned on human-annotated instruction-response pairs to enhance their instruction-following abilities. Works such as InstructGPT (Ouyang et al., 2022) have demonstrated this approach's effectiveness in enabling models to understand and deliver on tasks specified by humans. These instructions encompass a wide range of tasks and require a composition of capabilities (Zhong et al., 2024; Zhou et al., 2023), including answering questions concisely, summarizing long texts, translating between languages, explaining complex topics at various educational levels, and solving mathematical or logical problems step-by-step (Radford et al., 2019; Gupta et al., 2023). Our work extends this concept by considering another dimension of instruction following: the ability to infer and adhere to user preferences across multiple turns of conversation, rather than focusing solely on executing discrete tasks or queries.

## A.3 MODEL VERSION

With PREFEVAL, we have evaluated the following large language models in our experiments with their versions in Table 3.

Table 3: The LLMs benchmarked and their respective versions used for evaluation.

| Model Name | Version |
|---|---|
| Claude 3 Sonnet | anthropic.claude-3-sonnet-20240229-v1:0 |
| Claude 3 Haiku | anthropic.claude-3-haiku-20240307-v1:0 |
| Claude-3.5-Sonnet | anthropic.claude-3-5-sonnet-20240620-v1:0 |
| Mistral 7b Instruct | mistral.mistral-7b-instruct-v0:2 |
| Mistral 8x7b Instruct | mistral.mixtral-8x7b-instruct-v0:1 |
| LLaMA 3 8b Instruct | meta.llama3-8b-instruct-v1:0 |
| LLaMA 3 70b Instruct | meta.llama3-70b-instruct-v1:0 |
| GPT-4o | GPT-4o-2024-08-06 |
| o1-preview | o1-preview-2024-09-12 |
| Gemini-1.5-Pro | Gemini-1.5-Pro-latest as of Sep 30th, 2024. |

## A.4 METHODS DESCRIPTION

We extensively evaluate a variety of state-of-the-art large language models and investigate five methods for each model:

**Zero-shot:** The default case, where the LLM directly answers the user's query without any additional prompting.

**Reminder:** Before answering the question, the LLM is provided with a reminder sentence to consider the user's previously stated preference in its response. The reminder used is:

> "In your response, please ensure that you take into account our earlier discussion, and provide an answer that is consistent with my preference."

This reminder is added right after the question and before the LLM's response.

**Self-Critic:** The LLM first generates an initial *zero-shot* response to the question, then produces a critique assessing whether it has followed the user's preference, and finally generates a revised response incorporating the critique. In total, three calls to the LLM are made. This self-critic process is akin to *Intrinsic Self-Correction*, as termed in Huang et al..

The critique and revision request prompts are as follows:

---

**Critique Request:** Review your previous response to the user's question in the last conversation turn. Check if the response adheres to or violates any user preferences stated earlier in the conversation that relate to this query. Provide a critique on how well those preferences were followed in 2 sentences. Answer in this format:

`Critic:` [Your Critique Here]

**Revision Request:** Based on your critique, please rewrite your previous response to align more closely with the user's earlier stated preferences. Answer the question again:

`Response:` [Revised Response Here]

---

**Few-Shot Chain-of-Thought (CoT):** The LLM is given several few-shot examples (in our experiments, we used 5-shot) of chain-of-thoughts that demonstrate how to follow the user's preference before answering the question. The chain of thought prompt is as follows:

---

**When answering a user's question, a good assistant should carefully consider the user's stated preferences and tailor the response accordingly.**

Example 1: User's preference: "I have an intense phobia of bugs and insects. What outdoor activities would you recommend for the summer?" **Good assistant response:** "Considering your phobia of bugs and insects, I'll focus on recommending activities that minimize encounters with insects. Let's explore outdoor options with fewer bugs, like beaches, pools, urban areas, well-maintained park trails, and activities during cooler times of day. I'll avoid suggesting things like camping, off-trail hiking, picnicking, or activities in dense forests. Here are some summer outdoor activities you might enjoy: spending a day at the beach swimming and playing sports, exploring outdoor malls or pedestrian areas in the city, hiking on maintained trails in parks (avoiding dense woods), attending evening outdoor concerts or movies, and enjoying water activities like kayaking or paddle boating on lakes or rivers. These options should help you enjoy the outdoors while minimizing encounters with insects."

Example 2: User's preference: "I'm a vegan looking for new dinner recipes. Any ideas?" **Good assistant response:** "As a vegan, you'll need plant-based recipes without animal products. I'll focus on nutrient-rich, diverse ingredients to ensure balanced meals. Consider: quinoa and black bean burrito bowls, lentil and vegetable curry, roasted vegetable and hummus wrap, or zucchini noodles with avocado pesto. These options offer protein, fiber, and essential nutrients while adhering to vegan principles."

...

Example 5: User's preference: "I'm sensitive to caffeine but need an energy boost in the mornings. Any suggestions?" **Good assistant response:** "Given your caffeine sensitivity, we'll explore natural, caffeine-free energy boosters. Focus on lifestyle changes and alternatives to caffeinated drinks. Try: starting your day with a brisk walk or light exercise, drinking a glass of water with lemon, eating a protein-rich breakfast, or trying herbal teas like peppermint or ginger. Avoid coffee, energy drinks, and black tea. These alternatives can help increase alertness without the jitters or crashes associated with caffeine."

**Now, please answer the following question while considering my preferences (not the user preferences in the examples above), which I have stated either explicitly or implicitly in our previous conversation:**

---

**Retrieval-Augmented Generation (RAG):** We employ a sentence embedding model, which is *'princeton-nlp/sup-simcse-roberta-large'* (Gao et al., 2021), to retrieve the most relevant conversa-

tion exchanges based on similarity to the current query. The top five most relevant exchanges are then presented to the LLM as contextual information to guide its response.

The prompt used in this method is structured as follows, here we show RAG with top-5 retrieved exchanges:

```
Before answering my question, please consider the following context
from our previous conversations.  These are the 5 most relevant
exchanges that we had previously, which may contain information about
my preferences or prior discussions related to my query:
```

#Start of Context#
exchange 1. [Most relevant exchange 1]
exchange 2. [Most relevant exchange 2]
exchange 3. [Most relevant exchange 3]
exchange 4. [Most relevant exchange 4]
exchange 5. [Most relevant exchange 5]
#End of Context#

```
Please use this context to inform your answer and adhere to any
preferences I've expressed that are relevant to the current query.
Note that not all contexts are useful for answering my question
and there may be no context that is useful.  Now, please address my
question:
```

### A.5 CLASSIFICATION TASK CORRELATION PLOT

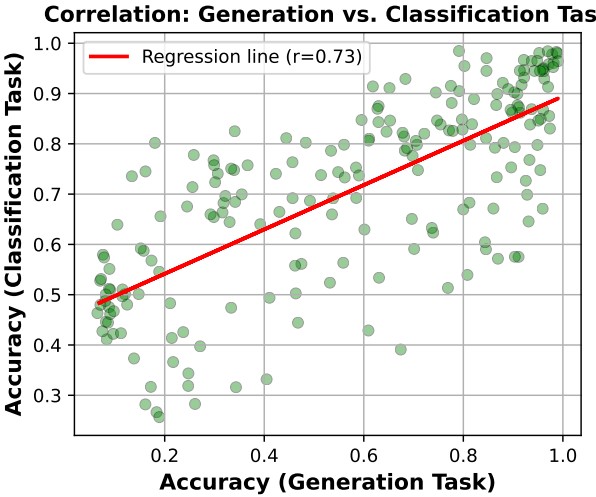

Figure 11: Correlation analysis between LLM-based preference following accuracy in generation tasks and classification accuracy in classification MCQ tasks across all models and methods. Across 6 models, 5 methods and 12 turns, each point on the scatter plot is averaged over 20 topics. A correlation coefficient of 0.73 suggests a strong alignment between the two evaluation approaches.

## A.6 PRACTICAL GUIDE TO USING PREFEVAL.

While our benchmark enables comprehensive evaluation across multiple dimensions (including various baselines, conversation turns, preference forms and topics, and tasks), benchmarking on complete setups is computationally intensive. For practical use, we provide guidance based on available resources. If an evaluator model like the Claude 3 Sonnet is available, then the generation task can be used; otherwise, one can choose a local LLM as the evaluator or opt for our classification task, which does not require LLM-based evaluators but still has strong correlation with generation task performance (see Sec 3.4). For initial testing, we recommend starting with a subset of topics and conversation lengths using explicit preference forms. As an example, our repository (https://github.com/amazon-science/PrefEval) includes a leaderboard comparing various LLMs on the "travel restaurant" topic at both 10 and 300 turns, assessing both short-turn and long-turn preference-following capabilities. With additional computational resources, one can further use generation task evaluators for detailed error-type analysis, and test implicit preference forms to evaluate more advanced preference-following capabilities.

## A.7 PROPRIETARY LLMS PERFORMANCE COMPARISON

The performance comparison shown in Table 4 shows that current SoTA proprietary LLMs struggle to proactively follow user preferences in short-turn, zero-shot settings, where no explicit prompting is provided. Only when using the Reminder method, which explicitly reinforces the need to adhere to preferences, do these models show improvement; however, accuracy still deteriorates with longer context lengths. Note that the o-series models' results may not be directly comparable to other models as they may require additional test-time computation with a "thinking" phase.

Table 4: Comparison of preference-following accuracy across SoTA LLMs evaluated at two context lengths with two methods: Zero-shot and Reminder (best prompting method), evaluated at two context lengths, on the *travel restaurant* topic and on generation task.

|  | 10 Turns / ~3k tokens | | 300 Turns / ~103k tokens | |
| --- | --- | --- | --- | --- |
|  | **Zero-shot** | **Reminder** | **Zero-shot** | **Reminder** |
| Claude-3.5-Sonnet | 0.07 | 0.45 | 0.02 | 0.02 |
| Gemini-1.5-Pro | 0.07 | 0.91 | 0.09 | 0.05 |
| GPT-4o | 0.07 | 0.98 | 0.05 | 0.23 |
| GPT-4.1 | 0.50 | 0.98 | 0.11 | 0.70 |
| o1-preview* | 0.50 | 0.98 | **0.14** | **0.98** |
| o4-mini* | 0.23 | 0.98 | 0.02 | 0.54 |
| o3* | **0.71** | **1.0** | 0.05 | 0.43 |

## A.8 RAG METHOD ABLATION AND ANALYSIS

To examine how RAG (Retrieval-Augmented Generation) improves performance, we compare the RAG sentence transformer's ground truth retrieval accuracy in explicit preference settings. We consider it accurate if the explicit preference stated in a turn is extracted by the sentence transformer and sent to the LLM as a reference. As shown in Figure 12, we find that when k=5, the performance is among the best, similar to when k=10. Although when k=10 the ground truth retrieval accuracy improves, the preference following performance does not reflect this improvement. This may suggest that for RAG, providing more turns of exchanges as reference might serve as another form of distraction, potentially harming performance. Based on these findings, we select k=5 to report this method's results in the main paper.

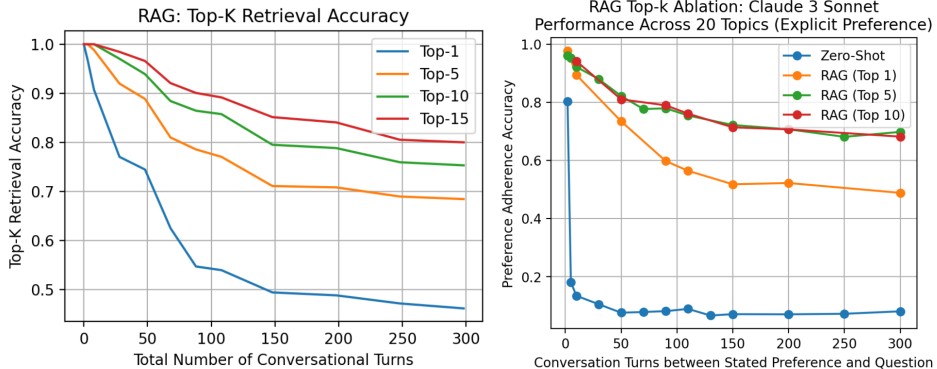

Figure 12: Comparison of RAG retrieval accuracy versus RAG method's performance in preference following across different Top-K values.

## A.9 CROSS-TOPIC PERFORMANCES

We present an extensive benchmark study conducted across 20 diverse topics, on 6 Large Language Models at three fixed conversation lengths: 10, 70, and 300 turns. These correspond to approximate context lengths of 3k, 23k, and 100k tokens, respectively, spanning from the user's initial stated preference to their final query. The user's stated preference is positioned at the beginning of each conversation. We show results across three preference forms and evaluates five different methods. The detailed results are presented in Tables 5, 6, and 7 for explicit preferences; Tables 8, 9, and 10 for implicit choice-based preferences; and Tables 11, 12, and 13 for implicit persona-based preferences. This comprehensive set of results allows for a thorough examination of LLM performance across various topics, conversation lengths, and preference revelation forms.

Table 5: Preference adherence accuracy across language models and two baselines when the number of conversation turns between the stated preference and the query is 10 (token length $\sim$ 3k), across 20 topics in the **explicit preference setting.**

| | Travel Restaurant | Travel Hotel | Travel Activities | Travel Transportation | Entertain Music&Book | Entertain Sports | Entertain Shows | Entertain Games | Lifestyle Dietary | Lifestyle Health |
|---|---|---|---|---|---|---|---|---|---|---|
| Claude 3 Sonnet + Zero-Shot | 5.36 | 16.67 | **5.17** | 10.87 | 10.71 | **7.69** | 9.68 | 17.65 | 3.51 | 18.37 |
| Claude 3 Haiku + Zero-Shot | 5.36 | 18.52 | **5.17** | 10.87 | **26.79** | 5.77 | **16.13** | **23.53** | 3.51 | 22.45 |
| Llama3 8B Instruct + Zero-Shot | 0.00 | 1.85 | 1.72 | 17.39 | 8.93 | 1.92 | 4.84 | 9.80 | 1.75 | 14.29 |
| Llama3 70B Instruct + Zero-Shot | **10.71** | 9.26 | 3.45 | 17.39 | 16.07 | 0.00 | 11.29 | 11.76 | 5.26 | 26.53 |
| Mistral 7b + Zero-Shot | 3.57 | **31.48** | 1.72 | **19.57** | 16.07 | **7.69** | 6.45 | 17.65 | **8.77** | 26.53 |
| Mistral 8x7b + Zero-Shot | 8.93 | 11.11 | **5.17** | 13.04 | 16.07 | 3.85 | 14.52 | 17.65 | 3.51 | **28.57** |
| Claude 3 Sonnet + Reminder | **96.43** | **100.00** | **100.00** | **100.00** | **100.00** | **100.00** | 98.39 | **100.00** | 98.25 | **100.00** |
| Claude 3 Haiku + Reminder | 67.86 | 92.59 | 79.31 | 97.83 | 96.43 | 94.23 | 88.71 | **100.00** | 59.65 | 93.88 |
| Llama3 8B Instruct + Reminder | 57.14 | 88.89 | 87.93 | 97.83 | 92.86 | 90.38 | 85.48 | 98.04 | 36.84 | 85.71 |
| Llama3 70B Instruct + Reminder | 37.50 | 24.07 | 29.31 | 50.00 | 41.07 | 38.46 | 40.32 | 66.67 | 54.39 | 67.35 |
| Mistral 7b + Reminder | 75.00 | 96.30 | 82.76 | 95.65 | 91.07 | 94.23 | 88.71 | 98.04 | 57.89 | 93.88 |
| Mistral 8x7b + Reminder | 83.93 | 98.15 | 89.66 | 97.83 | 94.64 | 88.46 | 90.32 | 98.04 | 85.96 | 95.92 |
| Claude 3 Sonnet + RAG (Top 5) | 64.15 | 88.46 | **87.93** | 93.48 | **96.43** | 90.38 | 95.00 | **100.00** | 63.16 | 91.11 |
| Claude 3 Haiku + RAG (Top 5) | 52.83 | 71.15 | 68.97 | 91.30 | 89.29 | 86.54 | 91.67 | 90.20 | 54.39 | 93.33 |
| Llama3 8B Instruct + RAG (Top 5) | 58.49 | 86.54 | 77.59 | 89.13 | **96.43** | 86.54 | 90.00 | 96.08 | 49.12 | 91.11 |
| Llama3 70B Instruct + RAG (Top 5) | 58.49 | 76.92 | 81.03 | 89.13 | 82.14 | 84.62 | 81.67 | 84.31 | 68.42 | 82.22 |
| Mistral 7b + RAG (Top 5) | 62.26 | 96.15 | 79.31 | **95.65** | 92.86 | 84.62 | 86.67 | 96.08 | 70.18 | 93.33 |
| Mistral 8x7b + RAG (Top 5) | **67.92** | 96.15 | 84.48 | 93.48 | 89.29 | 88.46 | 85.00 | 90.20 | **77.19** | **95.56** |
| Claude 3 Sonnet + Self-Critic | 78.57 | 92.59 | **100.00** | 93.48 | **98.21** | **96.15** | **93.55** | 90.20 | **87.72** | 93.88 |
| Claude 3 Haiku + Self-Critic | 73.21 | 92.59 | 91.38 | 93.48 | 92.86 | 90.38 | 85.48 | 92.16 | 63.16 | 91.84 |
| Llama3 8B Instruct + Self-Critic | 46.43 | 87.04 | 62.07 | 71.74 | 83.93 | 59.62 | 75.81 | 76.47 | 63.16 | 79.59 |
| Llama3 70B Instruct + Self-Critic | **87.50** | 92.59 | 84.48 | **95.65** | 85.71 | 76.92 | 83.87 | 88.24 | 82.46 | 89.80 |
| Mistral 7b + Self-Critic | 67.86 | **98.15** | 98.28 | 91.30 | 85.71 | 86.54 | 90.32 | **96.08** | 59.65 | **95.92** |
| Mistral 8x7b + Self-Critic | 58.93 | 87.04 | 74.14 | 84.78 | 78.57 | 80.77 | 66.13 | 90.20 | 70.18 | 85.71 |
| Claude 3 Sonnet + CoT | 41.07 | 72.22 | **87.93** | 78.26 | 85.71 | **80.77** | 82.26 | 92.16 | 31.58 | **89.80** |
| Claude 3 Haiku + CoT | 58.93 | 85.19 | 50.00 | 69.57 | 92.86 | 57.69 | 87.10 | 90.20 | 50.88 | 69.39 |
| Llama3 8B Instruct + CoT | 17.86 | 22.22 | 36.21 | 67.39 | 42.86 | 57.69 | 24.19 | 21.57 | 36.84 | 75.51 |
| Llama3 70B Instruct + CoT | 58.93 | 25.93 | 51.72 | 41.30 | 50.00 | 48.08 | 37.10 | 45.10 | 56.14 | 77.55 |
| Mistral 7b + CoT | 32.14 | 83.33 | 41.38 | 58.70 | 62.50 | 28.85 | 46.77 | 62.75 | 22.81 | 57.14 |
| Mistral 8x7b + CoT | **64.29** | **88.89** | **87.93** | **93.48** | **94.64** | 69.23 | 90.32 | **98.04** | **71.93** | 89.80 |

| | Lifestyle Fitness | Lifestyle Beauty | Shop Fashion | Shop Home | Shop Motors | Shop Technology | Education Learning Styles | Education Resources | Professional Work Style | Pet Ownership | Average |
|---|---|---|---|---|---|---|---|---|---|---|---|
| Claude 3 Sonnet + Zero-Shot | 5.77 | 11.32 | 5.56 | 14.29 | **20.00** | 14.29 | 19.35 | **37.04** | 13.89 | 20.93 | 13.41 |
| Claude 3 Haiku + Zero-Shot | 9.62 | 11.32 | 11.11 | 16.07 | 17.78 | 5.71 | **38.71** | 33.33 | 8.33 | 25.58 | 15.78 |
| Llama3 8B Instruct + Zero-Shot | 5.77 | 7.55 | 5.56 | 14.29 | 17.78 | 12.90 | 14.81 | 14.81 | 8.33 | 32.56 | 9.67 |
| Llama3 70B Instruct + Zero-Shot | 5.77 | 11.32 | 7.41 | 10.71 | 11.11 | 11.43 | 16.13 | 22.22 | 11.11 | 30.23 | 12.46 |
| Mistral 7b + Zero-Shot | **11.54** | **22.64** | **14.81** | **28.57** | 17.78 | **17.14** | 25.81 | 18.52 | 11.11 | 39.53 | **17.35** |
| Mistral 8x7b + Zero-Shot | 7.69 | 11.32 | 12.96 | 14.29 | 17.78 | 14.29 | 19.35 | 22.22 | **19.44** | **41.86** | 15.18 |
| Claude 3 Sonnet + Reminder | **100.00** | **100.00** | **100.00** | **100.00** | 95.56 | 97.14 | **100.00** | **100.00** | 97.22 | **97.67** | **99.03** |
| Claude 3 Haiku + Reminder | 96.15 | **100.00** | 96.30 | **100.00** | 93.33 | **100.00** | 96.77 | 98.15 | 94.44 | 88.37 | 91.70 |
| Llama3 8B Instruct + Reminder | 84.62 | 88.68 | 88.89 | 89.29 | 82.22 | 80.00 | 80.65 | 92.59 | 94.44 | 88.37 | 84.54 |
| Llama3 70B Instruct + Reminder | 61.54 | 37.74 | 31.48 | 42.86 | 26.67 | 34.29 | 54.84 | 85.19 | 22.22 | 67.44 | 45.67 |
| Mistral 7b + Reminder | 98.08 | 92.45 | 98.15 | 89.29 | 86.67 | 82.86 | 87.10 | 96.30 | 97.22 | 90.70 | 89.62 |
| Mistral 8x7b + Reminder | 96.15 | 94.34 | 94.44 | 94.64 | 91.11 | 97.14 | 90.32 | 96.30 | **100.00** | 90.70 | 93.40 |
| Claude 3 Sonnet + RAG (Top 5) | 92.31 | 98.11 | 96.30 | **100.00** | 97.78 | 96.88 | **100.00** | **100.00** | 97.22 | 94.29 | **92.15** |
| Claude 3 Haiku + RAG (Top 5) | 94.23 | 90.57 | 88.89 | 91.07 | 82.22 | 87.50 | 96.15 | 92.59 | 83.33 | 97.14 | 84.67 |
| Llama3 8B Instruct + RAG (Top 5) | **98.08** | 92.45 | 85.19 | 96.43 | 93.33 | 90.62 | **100.00** | 92.59 | **97.22** | 94.29 | 88.06 |
| Llama3 70B Instruct + RAG (Top 5) | 88.46 | 92.45 | 83.33 | 96.43 | 73.33 | 96.88 | 92.31 | 94.44 | 86.11 | **100.00** | 84.64 |
| Mistral 7b + RAG (Top 5) | 90.38 | **100.00** | **98.15** | 98.21 | 95.56 | **100.00** | 96.15 | 96.30 | 91.67 | 97.14 | 91.03 |
| Mistral 8x7b + RAG (Top 5) | 90.38 | 98.11 | 96.30 | 98.21 | 93.33 | **100.00** | 92.59 | 96.30 | | 88.57 | 90.57 |
| Claude 3 Sonnet + Self-Critic | 96.15 | 90.57 | 94.44 | **98.21** | 88.89 | 94.29 | 96.77 | 94.44 | 94.44 | 93.02 | **93.28** |
| Claude 3 Haiku + Self-Critic | **98.08** | 96.23 | **98.15** | 98.21 | 88.89 | 91.43 | **100.00** | 96.30 | 97.22 | 90.70 | 91.09 |
| Llama3 8B Instruct + Self-Critic | 80.77 | 84.91 | 77.78 | 85.71 | 71.11 | 68.57 | 67.74 | 72.22 | 80.56 | 83.72 | 73.95 |
| Llama3 70B Instruct + Self-Critic | 92.31 | 88.68 | 87.04 | 82.14 | 84.44 | 91.43 | 93.55 | 81.48 | 91.67 | 79.07 | 86.95 |
| Mistral 7b + Self-Critic | 86.54 | **98.11** | 87.04 | 96.43 | 88.89 | 91.43 | **100.00** | 96.30 | **97.22** | **95.35** | 90.36 |
| Mistral 8x7b + Self-Critic | 88.46 | 88.68 | 77.78 | 85.71 | 75.56 | 88.57 | 80.65 | 94.44 | 83.33 | 76.74 | 80.82 |
| Claude 3 Sonnet + CoT | 84.62 | 94.34 | **90.74** | 91.07 | 77.78 | 80.00 | 90.32 | 88.89 | 55.56 | **88.37** | 79.17 |
| Claude 3 Haiku + CoT | 69.23 | 83.02 | 81.48 | 89.29 | 64.44 | 80.00 | **100.00** | 96.30 | 83.33 | 83.72 | 77.13 |
| Llama3 8B Instruct + CoT | 69.23 | 60.38 | 53.70 | 60.71 | 20.00 | 37.14 | 61.29 | 61.11 | 33.33 | 67.44 | 46.33 |
| Llama3 70B Instruct + CoT | 59.62 | 35.85 | 35.19 | 53.57 | 13.33 | 40.00 | 58.06 | 68.52 | 55.56 | 72.09 | 49.18 |
| Mistral 7b + CoT | 55.77 | 81.13 | 51.85 | 50.00 | 53.33 | 40.00 | 67.74 | 83.33 | 83.33 | 60.47 | 56.17 |
| Mistral 8x7b + CoT | **90.38** | **96.23** | 81.48 | 82.14 | 77.78 | 82.86 | 90.32 | **96.30** | **97.22** | 83.72 | **86.35** |

Table 6: Preference adherence accuracy across language models and two baselines when the number of conversation turns between the stated preference and the query is 70 (token length ∼ 23k), across 20 topics in the **explicit preference setting.**

| | Travel Restaurant | Travel Hotel | Travel Activities | Travel Transportation | Entertain Music&Book | Entertain Sports | Entertain Shows | Entertain Games | Lifestyle Dietary | Lifestyle Health |
|---|---|---|---|---|---|---|---|---|---|---|
| Claude 3 Sonnet + Zero-Shot | 3.57 | 1.85 | 1.72 | 10.87 | 7.14 | 0.00 | 4.84 | **11.76** | **3.51** | 12.24 |
| Claude 3 Haiku + Zero-Shot | 1.79 | 3.70 | **3.45** | 6.52 | **10.71** | 0.00 | **8.06** | 5.88 | 1.75 | 12.24 |
| Mistral 7b + Zero-Shot | 0.00 | 7.41 | **3.45** | 8.70 | 8.93 | **1.92** | 6.45 | 5.88 | 1.75 | 24.49 |
| Mistral 8x7b + Zero-Shot | **8.93** | 3.70 | 1.72 | **15.22** | **10.71** | **1.92** | 6.45 | 5.88 | 0.00 | 22.45 |
| Claude 3 Sonnet + Reminder | 58.93 | **92.59** | **87.93** | 84.78 | **87.50** | **94.23** | 80.65 | **96.08** | **71.93** | **89.80** |
| Claude 3 Haiku + Reminder | 8.93 | 46.30 | 22.41 | 60.87 | 58.93 | 53.85 | 41.94 | 72.55 | 14.04 | 63.27 |
| Mistral 7b + Reminder | 33.93 | 75.93 | 18.97 | 60.87 | 44.64 | 38.46 | 27.42 | 56.86 | 17.54 | 63.27 |
| Mistral 8x7b + Reminder | **71.43** | 90.74 | 63.79 | **97.83** | 85.71 | 69.23 | **82.26** | 92.16 | 50.88 | 85.71 |
| Claude 3 Sonnet + RAG (Top 5) | 47.17 | 88.46 | **48.28** | 73.91 | **91.07** | 57.69 | **93.33** | 82.35 | 36.84 | **86.67** |
| Claude 3 Haiku + RAG (Top 5) | 28.30 | 59.62 | 37.93 | 60.87 | 62.50 | **59.62** | 63.33 | 72.55 | 31.58 | 82.22 |
| Mistral 7b + RAG (Top 5) | **58.49** | 96.15 | 43.10 | **84.78** | 82.14 | 51.92 | 85.00 | 80.39 | 40.35 | 80.00 |
| Mistral 8x7b + RAG (Top 5) | 52.83 | 90.38 | 44.83 | 73.91 | 75.00 | 44.23 | 75.00 | 78.43 | **45.61** | 80.00 |
| Claude 3 Sonnet + Self-Critic | 44.64 | 59.26 | **87.93** | 76.09 | **76.79** | 80.77 | 58.06 | 56.86 | 38.60 | 69.39 |
| Claude 3 Haiku + Self-Critic | 39.29 | 48.15 | 22.41 | 45.65 | 37.50 | 53.85 | 35.48 | 23.53 | 21.05 | 59.18 |
| Mistral 7b + Self-Critic | 17.86 | 25.93 | 15.52 | 17.39 | 25.00 | 19.23 | 12.90 | 27.45 | 28.07 | 26.53 |
| Mistral 8x7b + Self-Critic | 17.86 | 57.41 | 18.97 | 43.48 | 50.00 | 28.85 | 43.55 | 45.10 | 10.53 | 48.98 |
| Claude 3 Sonnet + CoT | 19.64 | 22.22 | 20.69 | 30.43 | 55.36 | 28.85 | 33.87 | 33.33 | 10.53 | 38.78 |
| Claude 3 Haiku + CoT | 26.79 | 38.89 | 20.69 | 34.78 | 50.00 | 34.62 | **61.29** | 47.06 | 24.56 | 38.78 |
| Mistral 7b + CoT | 26.79 | **77.78** | 41.38 | 52.17 | 44.64 | 30.77 | 38.71 | 60.78 | 22.81 | 46.94 |
| Mistral 8x7b + CoT | **57.14** | 72.22 | **65.52** | 63.04 | **73.21** | 46.15 | 58.06 | 52.94 | **54.39** | 69.39 |

| | Lifestyle Fitness | Lifestyle Beauty | Shop Fashion | Shop Home | Shop Motors | Shop Technology | Education Learning Styles | Education Resources | Professional Work Style | Pet Ownership | Average |
|---|---|---|---|---|---|---|---|---|---|---|---|
| Claude 3 Sonnet + Zero-Shot | 7.69 | 0.00 | **3.70** | 5.36 | 15.56 | 5.71 | 9.68 | 18.52 | **16.67** | 16.28 | 7.83 |
| Claude 3 Haiku + Zero-Shot | 1.92 | 5.66 | 1.85 | **8.93** | 11.11 | 0.00 | **22.58** | 25.93 | 13.89 | 30.23 | 8.81 |
| Mistral 7b + Zero-Shot | 9.62 | **11.32** | **3.70** | 7.14 | 13.33 | 8.57 | **22.58** | 18.52 | 8.33 | 23.26 | 9.77 |
| Mistral 8x7b + Zero-Shot | 13.46 | 9.43 | **3.70** | 8.93 | 22.22 | 11.43 | 19.35 | 18.52 | 11.11 | **34.88** | **11.50** |
| Claude 3 Sonnet + Reminder | **96.15** | 98.11 | 98.15 | 91.07 | 88.89 | 85.71 | 77.42 | 96.30 | 83.33 | 76.74 | 86.81 |
| Claude 3 Haiku + Reminder | 67.31 | 39.62 | 31.48 | 48.21 | 24.44 | 40.00 | 80.65 | 79.63 | 47.22 | 65.12 | 48.34 |
| Mistral 7b + Reminder | 65.38 | 62.26 | 38.89 | 39.29 | 31.11 | 20.00 | 67.74 | 64.81 | 36.11 | 60.47 | 46.20 |
| Mistral 8x7b + Reminder | 80.77 | 84.91 | 79.63 | 76.79 | 71.11 | 65.71 | **83.87** | 94.44 | **88.89** | 83.72 | 79.98 |
| Claude 3 Sonnet + RAG (Top 5) | 88.46 | 88.68 | 81.48 | **96.43** | 88.89 | 62.50 | **76.92** | 88.89 | 75.00 | 100.00 | 77.65 |
| Claude 3 Haiku + RAG (Top 5) | 82.69 | 58.49 | 66.67 | 85.71 | 80.00 | 43.75 | 73.08 | 72.22 | 58.33 | 80.00 | 62.97 |
| Mistral 7b + RAG (Top 5) | 84.62 | 88.68 | **87.04** | 82.14 | 68.89 | 62.50 | 57.69 | 75.93 | 63.89 | 85.71 | 72.97 |
| Mistral 8x7b + RAG (Top 5) | 76.92 | **92.45** | 83.33 | 87.50 | 73.33 | **68.75** | 57.69 | 66.67 | 69.44 | 74.29 | 70.53 |
| Claude 3 Sonnet + Self-Critic | **71.15** | 62.26 | **68.52** | 58.93 | 66.67 | 42.86 | 51.61 | 94.44 | 50.00 | 46.51 | 63.07 |
| Claude 3 Haiku + Self-Critic | 65.38 | 43.40 | 46.30 | 48.21 | 28.89 | 5.71 | **64.52** | 48.15 | 33.33 | 39.53 | 40.48 |
| Mistral 7b + Self-Critic | 25.00 | 24.53 | 16.67 | 10.71 | 24.44 | 11.43 | 12.90 | 24.07 | 16.67 | 39.53 | 21.09 |
| Mistral 8x7b + Self-Critic | 53.85 | **64.15** | 44.44 | 46.43 | 40.00 | 31.43 | 32.26 | 55.56 | 27.78 | **60.47** | 41.05 |
| Claude 3 Sonnet + CoT | 25.00 | 49.06 | 31.48 | **50.00** | 15.56 | 25.71 | 45.16 | **79.63** | 19.44 | 46.51 | 34.06 |
| Claude 3 Haiku + CoT | 28.85 | 41.51 | 25.93 | 23.21 | 40.00 | 17.14 | 6.45 | 29.63 | 30.56 | **58.14** | 33.94 |
| Mistral 7b + CoT | **48.08** | 56.60 | 35.19 | 39.29 | **48.89** | 22.86 | 38.71 | 70.37 | **66.67** | 55.81 | 46.26 |
| Mistral 8x7b + CoT | 44.23 | **79.25** | **46.30** | 50.00 | 37.78 | **34.29** | **61.29** | 79.63 | 66.67 | 58.14 | **58.48** |

Table 7: Preference adherence accuracy across language models and two baselines when the number of conversation turns between the stated preference and the query is 300 (token length $\sim$ 100k), across 20 topics in the **explicit preference setting.**

| | Travel Restaurant | Travel Hotel | Travel Activities | Travel Transportation | Entertain Music&Book | Entertain Sports | Entertain Shows | Entertain Games | Lifestyle Dietary | Lifestyle Health |
|---|---|---|---|---|---|---|---|---|---|---|
| Claude 3 Sonnet + Zero-Shot | **3.57** | **9.26** | **1.72** | 4.35 | 3.57 | **1.92** | 6.45 | 13.73 | **5.26** | 6.12 |
| Claude 3 Haiku + Zero-Shot | 1.79 | 1.85 | **1.72** | 8.70 | **7.14** | 0.00 | **9.68** | 9.80 | 0.00 | **8.16** |
| Claude 3 Sonnet + Reminder | 35.71 | 42.59 | 32.76 | 63.04 | 62.50 | 36.54 | 53.23 | 64.71 | 45.61 | 65.31 |
| Claude 3 Haiku + Reminder | 1.79 | 12.96 | 12.07 | 39.13 | 37.50 | 30.77 | 30.65 | 41.18 | 14.04 | 44.90 |
| Claude 3 Sonnet + RAG (Top 5) | 37.74 | 88.46 | 39.66 | 60.87 | 78.57 | 48.08 | 90.00 | 78.43 | 26.32 | 77.78 |
| Claude 3 Haiku + RAG (Top 5) | 18.87 | 34.62 | 17.24 | 50.00 | 60.71 | 40.38 | 66.67 | 68.63 | 19.30 | 66.67 |
| Claude 3 Sonnet + Self-Critic | 7.14 | 9.26 | **15.52** | **15.22** | **21.43** | 15.38 | **20.97** | 15.69 | 10.53 | 30.61 |
| Claude 3 Haiku + Self-Critic | 23.21 | 12.96 | 13.79 | 8.70 | 10.71 | **30.77** | 16.13 | 13.73 | **10.53** | 34.69 |
| Claude 3 Sonnet + CoT | 14.29 | 18.52 | 10.34 | **23.91** | 19.64 | 21.15 | 20.97 | 29.41 | 8.77 | 44.90 |
| Claude 3 Haiku + CoT | **21.43** | 16.67 | **22.41** | 8.70 | **28.57** | 19.23 | **35.48** | 33.33 | 21.05 | 30.61 |

| | Lifestyle Fitness | Lifestyle Beauty | Shop Fashion | Shop Home | Shop Motors | Shop Technology | Education Learning Styles | Education Resources | Professional Work Style | Pet Ownership | Average |
|---|---|---|---|---|---|---|---|---|---|---|---|
| Claude 3 Sonnet + Zero-Shot | **3.85** | 3.77 | **1.85** | 7.14 | **15.56** | 2.86 | **16.13** | 22.22 | **11.11** | 20.93 | **8.07** |
| Claude 3 Haiku + Zero-Shot | 0.00 | **5.66** | 0.00 | 5.36 | 13.33 | **5.71** | 16.13 | 20.37 | **11.11** | 23.26 | 7.49 |
| Claude 3 Sonnet + Reminder | 63.46 | 81.13 | 62.96 | 46.43 | 40.00 | 51.43 | 54.84 | 87.04 | 61.11 | 46.51 | 54.85 |
| Claude 3 Haiku + Reminder | 51.92 | 41.51 | 22.22 | 42.86 | 15.56 | 22.86 | 35.48 | 51.85 | 36.11 | **51.16** | 31.83 |
| Claude 3 Sonnet + RAG (Top 5) | 76.92 | 90.57 | 83.33 | 85.71 | 84.44 | 46.88 | 69.23 | 77.78 | 72.22 | 82.86 | 69.79 |
| Claude 3 Haiku + RAG (Top 5) | 69.23 | 58.49 | 62.96 | 67.86 | 62.22 | 31.25 | 57.69 | 59.26 | 36.11 | 80.00 | 51.41 |
| Claude 3 Sonnet + Self-Critic | 23.08 | **18.87** | **16.67** | 12.50 | 20.00 | 11.43 | 32.26 | 98.15 | 25.00 | 11.63 | 21.57 |
| Claude 3 Haiku + Self-Critic | **28.85** | **18.87** | 12.96 | **19.64** | 8.89 | 5.71 | 22.58 | 38.89 | 16.67 | **30.23** | 18.93 |
| Claude 3 Sonnet + CoT | 21.15 | 22.64 | 12.96 | 16.07 | 13.33 | 25.71 | 61.29 | 59.26 | **27.78** | 39.53 | 25.58 |
| Claude 3 Haiku + CoT | **25.00** | **24.53** | **20.37** | **25.00** | 15.56 | **45.71** | **70.97** | **70.37** | 25.00 | 37.21 | **29.86** |

Table 8: Preference adherence accuracy across language models and two baselines when the number of conversation turns between the stated preference and the query is 10 (token length ∼ 3k), across 20 topics. The preference type considered here is **implicit preference: choice-based dialogue.**

| | Travel Restaurant | Travel Hotel | Travel Activities | Travel Transportation | Entertain Music&Book | Entertain Sports | Entertain Shows | Entertain Games | Lifestyle Dietary | Lifestyle Health |
|---|---|---|---|---|---|---|---|---|---|---|
| Claude 3 Sonnet + Zero-Shot | 0.00 | 9.62 | 0.00 | 8.70 | 8.93 | 0.00 | 6.67 | 15.69 | 1.75 | 6.67 |
| Claude 3 Haiku + Zero-Shot | 0.00 | 3.85 | 3.45 | 10.87 | 3.57 | 1.92 | 3.33 | 9.80 | 1.75 | 8.89 |
| Llama3 8B Instruct + Zero-Shot | 3.77 | 3.85 | 3.45 | 15.22 | 7.14 | 5.77 | 1.67 | 1.96 | 1.75 | 15.56 |
| Llama3 70B Instruct + Zero-Shot | 5.66 | 3.85 | 3.45 | 17.39 | 10.71 | 0.00 | 5.00 | 9.80 | 0.00 | 13.33 |
| Mistral 7b + Zero-Shot | 0.00 | 5.77 | 1.72 | 10.87 | 3.57 | 0.00 | 5.00 | 15.69 | 3.51 | 11.11 |
| Mistral 8x7b + Zero-Shot | 0.00 | 3.85 | 0.00 | 4.35 | 7.14 | 1.92 | 0.00 | 9.80 | 1.75 | 20.00 |
| Claude 3 Sonnet + Reminder | 73.58 | 100.00 | 89.66 | 95.65 | 91.07 | 90.38 | 83.33 | 98.04 | 63.16 | 77.78 |
| Claude 3 Haiku + Reminder | 56.60 | 82.69 | 87.93 | 95.65 | 91.07 | 80.77 | 76.67 | 96.08 | 35.09 | 68.89 |
| Llama3 8B Instruct + Reminder | 22.64 | 82.69 | 51.72 | 89.13 | 80.36 | 55.77 | 58.33 | 82.35 | 12.28 | 68.89 |
| Llama3 70B Instruct + Reminder | 28.30 | 40.38 | 44.83 | 52.17 | 50.00 | 55.77 | 51.67 | 54.90 | 43.86 | 57.78 |
| Mistral 7b + Reminder | 49.06 | 80.77 | 55.17 | 73.91 | 67.86 | 63.46 | 55.00 | 72.55 | 22.81 | 64.44 |
| Mistral 8x7b + Reminder | 60.38 | 90.38 | 74.14 | 86.96 | 80.36 | 67.31 | 58.33 | 80.39 | 38.60 | 73.33 |
| Claude 3 Sonnet + Self-Critic | 16.98 | 28.85 | 32.76 | 67.39 | 53.57 | 40.38 | 40.00 | 41.18 | 15.79 | 46.67 |
| Claude 3 Haiku + Self-Critic | 22.64 | 34.62 | 20.69 | 34.78 | 41.07 | 44.23 | 21.67 | 29.41 | 15.79 | 57.78 |
| Llama3 8B Instruct + Self-Critic | 20.75 | 42.31 | 31.03 | 47.83 | 51.79 | 32.69 | 40.00 | 47.06 | 15.79 | 55.56 |
| Llama3 70B Instruct + Self-Critic | 15.09 | 46.15 | 32.76 | 45.65 | 48.21 | 32.69 | 41.67 | 37.25 | 22.81 | 48.89 |
| Mistral 7b + Self-Critic | 47.17 | 61.54 | 41.38 | 67.39 | 58.93 | 26.92 | 40.00 | 58.82 | 28.07 | 57.78 |
| Mistral 8x7b + Self-Critic | 22.64 | 25.00 | 25.86 | 21.74 | 35.71 | 26.92 | 26.67 | 33.33 | 15.79 | 42.22 |
| Claude 3 Sonnet + CoT | 11.32 | 46.15 | 31.03 | 67.39 | 60.71 | 42.31 | 50.00 | 58.82 | 21.05 | 66.67 |
| Claude 3 Haiku + CoT | 26.42 | 53.85 | 22.41 | 34.78 | 76.79 | 25.00 | 65.00 | 62.75 | 28.07 | 48.89 |
| Llama3 8B Instruct + CoT | 1.89 | 1.92 | 3.45 | 28.26 | 12.50 | 13.46 | 6.67 | 11.76 | 10.53 | 44.44 |
| Llama3 70B Instruct + CoT | 28.30 | 46.15 | 41.38 | 56.52 | 42.86 | 46.15 | 53.33 | 60.78 | 22.81 | 57.78 |
| Mistral 7b + CoT | 22.64 | 65.38 | 24.14 | 47.83 | 39.29 | 13.46 | 26.67 | 52.94 | 15.79 | 37.78 |
| Mistral 8x7b + CoT | 49.06 | 80.77 | 74.14 | 76.09 | 75.00 | 51.92 | 63.33 | 78.43 | 57.89 | 73.33 |
| Claude 3 Sonnet + RAG (top-5) | 75.47 | 100.00 | 89.66 | 97.83 | 91.07 | 94.23 | 85.00 | 100.00 | 64.91 | 77.78 |
| Claude 3 Haiku + RAG (top-5) | 49.06 | 73.08 | 86.21 | 89.13 | 85.71 | 73.08 | 71.67 | 80.39 | 45.61 | 77.78 |
| Llama3 8B Instruct + RAG (top-5) | 52.83 | 86.54 | 79.31 | 97.83 | 89.29 | 88.46 | 91.67 | 92.16 | 43.86 | 84.44 |
| Llama3 70B Instruct + RAG (top-5) | 45.28 | 71.15 | 72.41 | 93.48 | 75.00 | 82.69 | 73.33 | 88.24 | 43.86 | 73.33 |
| Mistral 7b + RAG (top-5) | 58.49 | 82.69 | 65.52 | 91.30 | 69.64 | 71.15 | 61.67 | 82.35 | 33.33 | 73.33 |
| Mistral 8x7b + RAG (top-5) | 58.49 | 75.00 | 62.07 | 91.30 | 78.57 | 73.08 | 56.67 | 82.35 | 36.84 | 66.67 |

| | Lifestyle Fitness | Lifestyle Beauty | Shop Fashion | Shop Home | Shop Motors | Shop Technology | Education Learning Styles | Education Resources | Professional Work Style | Pet Ownership | Average |
|---|---|---|---|---|---|---|---|---|---|---|---|
| Claude 3 Sonnet + Zero-Shot | 3.85 | 5.66 | 3.70 | 10.71 | 8.89 | 3.12 | 11.54 | 14.81 | 5.56 | 20.00 | 7.29 |
| Claude 3 Haiku + Zero-Shot | 5.77 | 1.89 | 3.70 | 7.14 | 13.33 | 6.25 | 19.23 | 18.52 | 2.78 | 20.00 | 7.30 |
| Llama3 8B Instruct + Zero-Shot | 0.00 | 3.77 | 0.00 | 8.93 | 8.89 | 6.25 | 7.69 | 12.96 | 2.78 | 17.14 | 6.43 |
| Llama3 70B Instruct + Zero-Shot | 5.77 | 5.66 | 1.85 | 7.14 | 11.11 | 3.12 | 7.69 | 14.81 | 11.11 | 22.86 | 8.02 |
| Mistral 7b + Zero-Shot | 13.46 | 9.43 | 7.41 | 10.71 | 4.44 | 3.12 | 11.54 | 16.67 | 5.56 | 20.00 | 7.98 |
| Mistral 8x7b + Zero-Shot | 7.69 | 3.77 | 5.56 | 10.71 | 17.78 | 0.00 | 7.69 | 7.41 | 8.33 | 17.14 | 6.75 |
| Claude 3 Sonnet + Reminder | 92.31 | 94.34 | 90.74 | 96.43 | 95.56 | 96.88 | 96.15 | 94.44 | 91.67 | 100.00 | 90.56 |
| Claude 3 Haiku + Reminder | 78.85 | 86.79 | 79.63 | 80.36 | 82.22 | 65.62 | 88.46 | 92.59 | 83.33 | 91.43 | 80.04 |
| Llama3 8B Instruct + Reminder | 63.46 | 71.70 | 64.81 | 75.00 | 75.56 | 65.62 | 84.62 | 87.04 | 77.78 | 77.14 | 67.34 |
| Llama3 70B Instruct + Reminder | 55.77 | 37.74 | 29.63 | 53.57 | 48.89 | 46.88 | 65.38 | 61.11 | 38.89 | 74.29 | 49.59 |
| Mistral 7b + Reminder | 65.38 | 83.02 | 66.67 | 80.36 | 80.00 | 59.38 | 92.31 | 88.89 | 77.78 | 74.29 | 68.65 |
| Mistral 8x7b + Reminder | 69.23 | 83.02 | 70.37 | 76.79 | 73.33 | 75.00 | 88.46 | 88.89 | 86.11 | 91.43 | 75.64 |
| Claude 3 Sonnet + Self-Critic | 57.69 | 41.51 | 50.00 | 57.14 | 46.67 | 21.88 | 69.23 | 44.44 | 33.33 | 60.00 | 43.27 |
| Claude 3 Haiku + Self-Critic | 51.92 | 54.72 | 46.30 | 58.93 | 51.11 | 25.00 | 50.00 | 51.85 | 41.67 | 57.14 | 40.57 |
| Llama3 8B Instruct + Self-Critic | 42.31 | 54.72 | 62.96 | 69.64 | 51.11 | 59.38 | 57.69 | 53.70 | 47.22 | 82.86 | 48.32 |
| Llama3 70B Instruct + Self-Critic | 40.38 | 37.74 | 35.19 | 60.71 | 48.89 | 37.50 | 46.15 | 50.00 | 47.22 | 65.71 | 42.03 |
| Mistral 7b + Self-Critic | 63.46 | 54.72 | 48.15 | 60.71 | 64.44 | 59.38 | 61.54 | 68.52 | 50.00 | 65.71 | 54.23 |
| Mistral 8x7b + Self-Critic | 34.62 | 33.96 | 40.74 | 37.50 | 46.67 | 40.62 | 61.54 | 57.41 | 50.00 | 48.57 | 36.38 |
| Claude 3 Sonnet + CoT | 55.77 | 67.92 | 68.52 | 73.21 | 73.33 | 40.62 | 53.85 | 83.33 | 61.11 | 77.14 | 55.51 |
| Claude 3 Haiku + CoT | 36.54 | 69.81 | 53.70 | 69.64 | 57.78 | 43.75 | 88.46 | 79.63 | 52.78 | 71.43 | 53.37 |
| Llama3 8B Instruct + CoT | 38.46 | 28.30 | 27.78 | 51.79 | 17.78 | 12.50 | 38.46 | 33.33 | 13.89 | 48.57 | 22.29 |
| Llama3 70B Instruct + CoT | 59.62 | 33.96 | 44.44 | 60.71 | 44.44 | 40.62 | 69.23 | 66.67 | 55.56 | 62.86 | 49.71 |
| Mistral 7b + CoT | 38.46 | 58.49 | 35.19 | 39.29 | 51.11 | 31.25 | 50.00 | 59.26 | 69.44 | 57.14 | 41.78 |
| Mistral 8x7b + CoT | 61.54 | 84.91 | 61.11 | 67.86 | 62.22 | 53.12 | 88.46 | 87.04 | 77.78 | 74.29 | 69.91 |
| Claude 3 Sonnet + RAG (top-5) | 82.69 | 94.34 | 94.44 | 96.43 | 95.56 | 81.25 | 100.00 | 94.44 | 100.00 | 97.14 | 90.61 |
| Claude 3 Haiku + RAG (top-5) | 75.00 | 69.81 | 77.78 | 91.07 | 88.89 | 84.38 | 84.62 | 92.59 | 86.11 | 94.29 | 78.81 |
| Llama3 8B Instruct + RAG (top-5) | 78.85 | 79.25 | 83.33 | 91.07 | 88.89 | 87.50 | 100.00 | 92.59 | 88.89 | 94.29 | 84.55 |
| Llama3 70B Instruct + RAG (top-5) | 71.15 | 84.91 | 70.37 | 87.50 | 66.67 | 62.50 | 88.46 | 83.33 | 86.11 | 91.43 | 75.56 |
| Mistral 7b + RAG (top-5) | 69.23 | 94.34 | 81.48 | 87.50 | 84.44 | 71.88 | 96.15 | 92.59 | 97.22 | 71.43 | 76.79 |
| Mistral 8x7b + RAG (top-5) | 69.23 | 83.02 | 79.63 | 85.71 | 75.56 | 78.12 | 88.46 | 88.89 | 86.11 | 88.57 | 75.22 |

Table 9: Preference adherence accuracy across language models and two baselines when the number of conversation turns between the stated preference and the query is 70 (token length ∼ 23k), across 20 topics. The preference type considered here is **implicit preference: choice-based dialogue.**

| | Travel Restaurant | Travel Hotel | Travel Activities | Travel Transportation | Entertain Music&Book | Entertain Sports | Entertain Shows | Entertain Games | Lifestyle Dietary | Lifestyle Health |
|---|---|---|---|---|---|---|---|---|---|---|
| Claude 3 Sonnet + Zero-Shot | 0.00 | **7.69** | 0.00 | **10.87** | **8.93** | 3.85 | 5.00 | **11.76** | 3.51 | 4.44 |
| Claude 3 Haiku + Zero-Shot | 0.00 | 5.77 | **5.17** | 2.17 | 1.79 | 0.00 | 5.00 | 9.80 | 1.75 | 11.11 |
| Mistral 7b + Zero-Shot | 0.00 | 3.85 | 1.72 | **10.87** | 3.57 | 0.00 | **6.67** | 5.88 | **5.26** | **13.33** |
| Mistral 8x7b + Zero-Shot | **1.89** | 1.92 | 1.72 | **10.87** | 7.14 | 0.00 | **6.67** | 5.88 | 1.75 | **13.33** |
| Claude 3 Sonnet + Reminder | 43.40 | 86.54 | 72.41 | 86.96 | **78.57** | **75.00** | 43.33 | **84.31** | **52.63** | 77.78 |
| Claude 3 Haiku + Reminder | 1.89 | 42.31 | 31.03 | 60.87 | 53.57 | 46.15 | 36.67 | 52.94 | 17.54 | 55.56 |
| Mistral 7b + Reminder | 18.87 | 50.00 | 25.86 | 60.87 | 33.93 | 25.00 | 21.67 | 52.94 | 12.28 | 55.56 |
| Mistral 8x7b + Reminder | **56.60** | 82.69 | 48.28 | **89.13** | 69.64 | 53.85 | **63.33** | 78.43 | 35.09 | **80.00** |
| Claude 3 Sonnet + Self-Critic | 1.89 | 13.46 | 5.17 | **21.74** | 16.07 | 13.46 | 5.00 | 7.84 | 8.77 | 8.89 |
| Claude 3 Haiku + Self-Critic | **22.64** | **23.08** | **25.86** | **21.74** | 26.79 | 13.46 | **18.33** | 19.61 | 12.28 | **28.89** |
| Mistral 7b + Self-Critic | 15.09 | 11.54 | 22.41 | 13.04 | **30.36** | 15.38 | **18.33** | **33.33** | **19.30** | 11.11 |
| Mistral 8x7b + Self-Critic | 7.55 | 13.46 | 12.07 | **21.74** | 19.64 | 5.77 | **18.33** | 25.49 | 1.75 | 24.44 |
| Claude 3 Sonnet + CoT | 5.66 | 19.23 | 15.52 | 13.04 | 25.00 | 19.23 | 15.00 | 31.37 | 10.53 | 26.67 |
| Claude 3 Haiku + CoT | 33.96 | 13.46 | 17.24 | 8.70 | 21.43 | 9.62 | 23.33 | 23.53 | 24.56 | 35.56 |
| Mistral 7b + CoT | 26.42 | 65.38 | 43.10 | 41.30 | 39.29 | 26.92 | 36.67 | **58.82** | 19.30 | **46.67** |
| Mistral 8x7b + CoT | **52.83** | 42.31 | **58.62** | **47.83** | **55.36** | **42.31** | **41.67** | 33.33 | **38.60** | 44.44 |
| Claude 3 Sonnet + RAG (top-5) | **60.38** | 86.54 | 68.97 | **95.65** | **85.71** | **76.92** | **78.33** | 80.39 | **52.63** | **73.33** |
| Claude 3 Haiku + RAG (top-5) | 41.51 | 82.69 | 60.34 | 91.30 | 82.14 | 67.31 | **78.33** | **82.35** | 24.56 | 66.67 |
| Mistral 7b + RAG (top-5) | 43.40 | 82.69 | 44.83 | 82.61 | 64.29 | 55.77 | 58.33 | 52.94 | 29.82 | 62.22 |
| Mistral 8x7b + RAG (top-5) | 43.40 | 71.15 | 27.59 | 67.39 | 64.29 | 46.15 | 55.00 | 49.02 | 22.81 | 57.78 |

| | Lifestyle Fitness | Lifestyle Beauty | Shop Fashion | Shop Home | Shop Motors | Shop Technology | Education Learning Styles | Education Resources | Professional Work Style | Pet Ownership | Average |
|---|---|---|---|---|---|---|---|---|---|---|---|
| Claude 3 Sonnet + Zero-Shot | 1.92 | 3.77 | 0.00 | 5.36 | 8.89 | 0.00 | 15.38 | **24.07** | 0.00 | 11.43 | 6.34 |
| Claude 3 Haiku + Zero-Shot | 0.00 | 3.77 | 0.00 | 7.14 | 4.44 | 0.00 | **19.23** | 22.22 | 0.00 | 14.29 | 5.68 |
| Mistral 7b + Zero-Shot | 11.54 | **7.55** | **3.70** | 5.36 | **15.56** | **6.25** | **19.23** | 14.81 | 2.78 | 14.29 | **7.61** |
| Mistral 8x7b + Zero-Shot | 7.69 | 5.66 | 1.85 | **8.93** | 6.67 | **6.25** | 15.38 | 9.26 | **5.56** | **20.00** | 6.92 |
| Claude 3 Sonnet + Reminder | **96.15** | **90.57** | **81.48** | **82.14** | **82.22** | **62.50** | **84.62** | **96.30** | **83.33** | 77.14 | **76.87** |
| Claude 3 Haiku + Reminder | 48.08 | 45.28 | 50.00 | 55.36 | 57.78 | 34.38 | 50.00 | 57.41 | 55.56 | **85.71** | 46.90 |
| Mistral 7b + Reminder | 57.69 | 50.94 | 38.89 | 46.43 | 31.11 | 25.00 | 57.69 | 59.26 | 44.44 | 65.71 | 41.71 |
| Mistral 8x7b + Reminder | 76.92 | 84.91 | 62.96 | 73.21 | 57.78 | 59.38 | 76.92 | 85.19 | 66.67 | 74.29 | 68.76 |
| Claude 3 Sonnet + Self-Critic | 9.62 | 18.87 | 22.22 | 23.21 | 15.56 | 6.25 | 30.77 | 22.22 | 11.11 | 25.71 | 14.39 |
| Claude 3 Haiku + Self-Critic | 26.92 | 30.19 | 29.63 | 37.50 | 20.00 | 12.50 | 34.62 | 38.89 | 27.78 | 28.57 | 24.96 |
| Mistral 7b + Self-Critic | 13.46 | 22.64 | 24.07 | 17.86 | 28.89 | 21.88 | 11.54 | 25.93 | 13.89 | 28.57 | 19.93 |
| Mistral 8x7b + Self-Critic | 23.08 | 22.64 | 24.07 | 28.57 | 24.44 | **25.00** | 42.31 | 35.19 | 19.44 | **37.14** | 21.61 |
| Claude 3 Sonnet + CoT | 21.15 | 26.42 | 16.67 | 23.21 | 8.89 | 25.00 | **50.00** | 53.70 | 22.22 | 45.71 | 23.71 |
| Claude 3 Haiku + CoT | 25.00 | 26.42 | 29.63 | 25.00 | 24.44 | **31.25** | 19.23 | 22.22 | 19.44 | 51.43 | 24.27 |
| Mistral 7b + CoT | **42.31** | 52.83 | **46.30** | 32.14 | 33.33 | 21.88 | 23.08 | **61.11** | **66.67** | 51.43 | 41.75 |
| Mistral 8x7b + CoT | 32.69 | **64.15** | 40.74 | **51.79** | 33.33 | 31.25 | 50.00 | 59.26 | 55.56 | **57.14** | **46.66** |
| Claude 3 Sonnet + RAG (top-5) | **82.69** | **88.68** | 75.93 | **92.86** | **84.44** | **53.12** | **88.46** | **92.59** | **88.89** | **85.71** | **79.61** |
| Claude 3 Haiku + RAG (top-5) | 80.77 | 64.15 | **77.78** | **92.86** | 82.22 | 50.00 | 80.77 | 77.78 | 58.33 | **85.71** | 71.38 |
| Mistral 7b + RAG (top-5) | 48.08 | 67.92 | 55.56 | 73.21 | 77.78 | 46.88 | 53.85 | 68.52 | 66.67 | 82.86 | 60.91 |
| Mistral 8x7b + RAG (top-5) | 48.08 | 66.04 | 59.26 | 64.29 | 57.78 | 37.50 | 46.15 | 55.56 | 44.44 | 77.14 | 53.04 |

Table 10: Preference adherence accuracy across language models and two baselines when the number of conversation turns between the stated preference and the query is 300 (token length $\sim$ 100k), across 20 topics. The preference type considered here is **implicit preference: choice-based dialogue.**

| | Travel Restaurant | Travel Hotel | Travel Activities | Travel Transportation | Entertain Music&Book | Entertain Sports | Entertain Shows | Entertain Games | Lifestyle Dietary | Lifestyle Health |
|---|---|---|---|---|---|---|---|---|---|---|
| Claude 3 Sonnet + Zero-Shot | **3.77** | **5.77** | 0.00 | **8.70** | 1.79 | **0.00** | 5.00 | 11.76 | **3.51** | 6.67 |
| Claude 3 Haiku + Zero-Shot | 0.00 | 1.92 | **1.72** | 4.35 | **7.14** | **0.00** | 8.33 | 7.84 | 1.75 | **8.89** |
| Claude 3 Sonnet + Reminder | 35.85 | 46.15 | 29.31 | 56.52 | 51.79 | 36.54 | 56.67 | 60.78 | 49.12 | 66.67 |
| Claude 3 Haiku + Reminder | 9.43 | 28.85 | 18.97 | 54.35 | 26.79 | **40.38** | 23.33 | 47.06 | 12.28 | 42.22 |
| Claude 3 Sonnet + Self-Critic | 3.77 | 7.69 | 6.90 | 10.87 | 8.93 | 7.69 | **6.67** | 13.73 | 5.26 | 13.33 |
| Claude 3 Haiku + Self-Critic | 18.87 | 15.38 | 10.34 | **15.22** | 10.71 | 19.23 | 5.00 | **15.69** | 14.04 | 31.11 |
| Claude 3 Sonnet + CoT | 3.77 | 19.23 | 6.90 | **17.39** | 8.93 | 21.15 | 10.00 | 31.37 | 10.53 | 37.78 |
| Claude 3 Haiku + CoT | 28.30 | 25.00 | 13.79 | 6.52 | 32.14 | 17.31 | 43.33 | 43.14 | 17.54 | 46.67 |
| Claude 3 Sonnet + RAG (top-5) | 43.40 | 86.54 | 51.72 | 73.91 | 73.21 | 59.62 | 65.00 | 39.22 | 31.58 | 62.22 |
| Claude 3 Haiku + RAG (top-5) | 30.19 | 53.85 | 46.55 | 69.57 | **78.57** | 51.92 | 55.00 | 50.98 | 29.82 | 57.78 |

| | Lifestyle Fitness | Lifestyle Beauty | Shop Fashion | Shop Home | Shop Motors | Shop Technology | Education Learning Styles | Education Resources | Professional Work Style | Pet Ownership | Average |
|---|---|---|---|---|---|---|---|---|---|---|---|
| Claude 3 Sonnet + Zero-Shot | **3.85** | **5.66** | 1.85 | **10.71** | 6.67 | **0.00** | 15.38 | 14.81 | 8.33 | 14.29 | **6.43** |
| Claude 3 Haiku + Zero-Shot | 0.00 | 1.89 | **1.85** | 8.93 | 2.22 | **0.00** | **19.23** | 12.96 | 5.56 | 14.29 | 5.44 |
| Claude 3 Sonnet + Reminder | 65.38 | 81.13 | 59.26 | 51.79 | 42.22 | 50.00 | 69.23 | 77.78 | 58.33 | 45.71 | **54.51** |
| Claude 3 Haiku + Reminder | 42.31 | 37.74 | 31.48 | 50.00 | 35.56 | 37.50 | 50.00 | 44.44 | 41.67 | **68.57** | 37.15 |
| Claude 3 Sonnet + Self-Critic | 9.62 | 3.77 | 7.41 | 14.29 | 11.11 | 6.25 | **30.77** | 20.37 | 5.56 | 14.29 | 10.41 |
| Claude 3 Haiku + Self-Critic | **36.54** | 26.42 | 18.52 | 26.79 | 13.33 | 12.50 | 30.77 | 42.59 | 11.11 | 28.57 | 20.14 |
| Claude 3 Sonnet + CoT | 23.08 | 22.64 | 14.81 | 16.07 | **11.11** | 12.50 | 61.54 | 59.26 | **30.56** | 31.43 | 22.50 |
| Claude 3 Haiku + CoT | 30.77 | 26.42 | 18.52 | 32.14 | 8.89 | 40.62 | 80.77 | 75.93 | 19.44 | 40.00 | 32.36 |
| Claude 3 Sonnet + RAG (top-5) | 65.38 | **84.91** | 64.81 | 85.71 | 73.33 | 50.00 | 69.23 | 72.22 | 66.67 | 91.43 | 65.51 |
| Claude 3 Haiku + RAG (top-5) | **67.31** | 56.60 | 62.96 | 82.14 | **75.56** | 34.38 | 65.38 | 62.96 | 63.89 | 82.86 | 58.91 |

Table 11: Preference adherence accuracy across language models and two baselines when the number of conversation turns between the stated preference and the query is 10 (token length ∼ 3k), across 20 topics. The preference type considered here is **implicit preference: persona-driven dialogue.**

| | Travel Restaurant | Travel Hotel | Travel Activities | Travel Transportation | Entertain Music&Book | Entertain Sports | Entertain Shows | Entertain Games | Lifestyle Dietary | Lifestyle Health |
|---|---|---|---|---|---|---|---|---|---|---|
| Claude 3 Sonnet + Zero-Shot | 1.89 | **9.62** | 1.72 | 6.52 | 3.57 | **5.77** | **10.00** | 3.92 | 1.75 | 4.44 |
| Claude 3 Haiku + Zero-Shot | 0.00 | 5.77 | 0.00 | 6.52 | 5.36 | 0.00 | 5.00 | 7.84 | 1.75 | 6.67 |
| Llama3 8B Instruct + Zero-Shot | 0.00 | 0.00 | 0.00 | 15.22 | 7.14 | 0.00 | 3.33 | 3.92 | 1.75 | 15.56 |
| Llama3 70B Instruct + Zero-Shot | **3.77** | 5.77 | **3.45** | 10.87 | **10.71** | 3.85 | 3.33 | 1.96 | 0.00 | 13.33 |
| Mistral 7b + Zero-Shot | **3.77** | 7.69 | 0.00 | **19.57** | 7.14 | 3.85 | 8.33 | 5.88 | **5.26** | **22.22** |
| Mistral 8x7b + Zero-Shot | 0.00 | 1.92 | 0.00 | 13.04 | 5.36 | 3.85 | 5.00 | 3.92 | 3.51 | 15.56 |
| Claude 3 Sonnet + Reminder | **84.91** | **98.08** | **84.48** | **95.65** | **98.21** | 90.38 | **95.00** | 92.16 | 87.72 | 88.89 |
| Claude 3 Haiku + Reminder | 32.08 | 69.23 | 39.66 | 73.91 | 64.29 | 63.46 | 71.67 | 78.43 | 28.07 | 60.00 |
| Llama3 8B Instruct + Reminder | 32.08 | 59.62 | 34.48 | 67.39 | 57.14 | 53.85 | 60.00 | 50.98 | 28.07 | 68.89 |
| Llama3 70B Instruct + Reminder | 30.19 | 3.85 | 18.97 | 23.91 | 14.29 | 9.62 | 15.00 | 27.45 | 42.11 | 40.00 |
| Mistral 7b + Reminder | 54.72 | 82.69 | 58.62 | 80.43 | 71.43 | 71.15 | 61.67 | 88.24 | 38.60 | 73.33 |
| Mistral 8x7b + Reminder | 67.92 | 90.38 | 74.14 | 86.96 | 87.50 | 80.77 | 83.33 | 82.35 | 78.95 | **91.11** |
| Claude 3 Sonnet + CoT | 20.75 | 17.31 | 37.93 | 36.96 | 42.86 | 48.08 | 38.33 | 43.14 | 29.82 | 64.44 |
| Claude 3 Haiku + CoT | 24.53 | 30.77 | 20.69 | 26.09 | 53.57 | 17.31 | 45.00 | 45.10 | 33.33 | 42.22 |
| Llama3 8B Instruct + CoT | 9.43 | 1.92 | 8.62 | 10.87 | 5.36 | 11.54 | 5.00 | 5.88 | 10.53 | 31.11 |
| Llama3 70B Instruct + CoT | 33.96 | 25.00 | 46.55 | 30.43 | 17.86 | 32.69 | 33.33 | 23.53 | 36.84 | 53.33 |
| Mistral 7b + CoT | 26.42 | 73.08 | 29.31 | 58.70 | 42.86 | 19.23 | 40.00 | 49.02 | 10.53 | 46.67 |
| Mistral 8x7b + CoT | **58.49** | 90.38 | 70.69 | 93.48 | 76.79 | 57.69 | 73.33 | 84.31 | 64.91 | 75.56 |
| Claude 3 Sonnet + Self-Critic | 28.30 | 44.23 | 41.38 | 58.70 | 51.79 | 61.54 | 45.00 | 33.33 | **40.35** | 46.67 |
| Claude 3 Haiku + Self-Critic | 26.42 | 34.62 | 27.59 | 32.61 | 39.29 | 44.23 | 38.33 | 37.25 | 31.58 | 64.44 |
| Llama3 8B Instruct + Self-Critic | 26.42 | 28.85 | 31.03 | 43.48 | 35.71 | 28.85 | 30.00 | 35.29 | 21.05 | 33.33 |
| Llama3 70B Instruct + Self-Critic | 16.98 | 15.38 | 32.76 | 13.04 | 12.50 | 25.00 | 15.00 | 21.57 | 17.54 | 35.56 |
| Mistral 7b + Self-Critic | **45.28** | **61.54** | **48.28** | **73.91** | **62.50** | 67.31 | **46.67** | 58.82 | **40.35** | **80.00** |
| Mistral 8x7b + Self-Critic | 30.19 | 46.15 | 46.55 | 54.35 | 51.79 | 42.31 | 43.33 | 41.18 | 38.60 | 64.44 |
| Claude 3 Sonnet + RAG (top-5) | 47.17 | 75.00 | **60.34** | 82.61 | **91.07** | 65.38 | **83.33** | 82.35 | 52.63 | 86.67 |
| Claude 3 Haiku + RAG (top-5) | 18.87 | 30.77 | 18.97 | 47.83 | 53.57 | 55.77 | 43.33 | 54.90 | 40.35 | 73.33 |
| Llama3 8B Instruct + RAG (top-5) | 47.17 | 71.15 | 56.90 | 73.91 | 82.14 | **73.08** | 81.67 | 80.39 | 49.12 | 80.00 |
| Llama3 70B Instruct + RAG (top-5) | 35.85 | 53.85 | 34.48 | 63.04 | 55.36 | 67.31 | 70.00 | 68.63 | 35.09 | 82.22 |
| Mistral 7b + RAG (top-5) | 39.62 | **88.46** | **60.34** | **84.78** | 71.43 | 63.46 | **83.33** | 76.47 | 38.60 | 80.00 |
| Mistral 8x7b + RAG (top-5) | **52.83** | 78.85 | 55.17 | **84.78** | 69.64 | 61.54 | 71.67 | 78.43 | **59.65** | 80.00 |

| | Lifestyle Fitness | Lifestyle Beauty | Shop Fashion | Shop Home | Shop Motors | Shop Technology | Education Learning Styles | Education Resources | Professional Work Style | Pet Ownership | Average |
|---|---|---|---|---|---|---|---|---|---|---|---|
| Claude 3 Sonnet + Zero-Shot | 3.85 | 5.66 | 1.85 | 5.36 | 8.89 | 6.25 | 11.54 | **29.63** | 5.56 | 17.14 | 7.25 |
| Claude 3 Haiku + Zero-Shot | 5.77 | 1.89 | 3.70 | 7.14 | 6.67 | 0.00 | 15.38 | 24.07 | **8.33** | 14.29 | 6.31 |
| Llama3 8B Instruct + Zero-Shot | 1.92 | 1.89 | 1.85 | 3.57 | 11.11 | 6.25 | 7.69 | 16.67 | 0.00 | 17.14 | 5.75 |
| Llama3 70B Instruct + Zero-Shot | 3.85 | 5.66 | 1.85 | 7.14 | 8.89 | 6.25 | 15.38 | 16.67 | 2.78 | 25.71 | 7.56 |
| Mistral 7b + Zero-Shot | **7.69** | 7.55 | **12.96** | **14.29** | 11.11 | 9.38 | 15.38 | **29.63** | 5.56 | 17.14 | **10.72** |
| Mistral 8x7b + Zero-Shot | 3.85 | **9.43** | **12.96** | 10.71 | **20.00** | 9.38 | **19.23** | 16.67 | **8.33** | **28.57** | 9.56 |
| Claude 3 Sonnet + Reminder | **96.15** | **98.11** | **94.44** | 85.71 | 88.89 | 100.00 | 96.15 | 98.15 | 91.67 | 100.00 | **93.24** |
| Claude 3 Haiku + Reminder | 61.54 | 66.04 | 59.26 | 67.86 | 53.33 | 43.75 | 65.38 | 68.52 | 61.11 | 91.43 | 60.95 |
| Llama3 8B Instruct + Reminder | 63.46 | 56.60 | 55.56 | 57.14 | 60.00 | 50.00 | 76.92 | 52.78 | 82.86 | 57.28 |
| Llama3 70B Instruct + Reminder | 15.38 | 18.87 | 5.56 | 17.86 | 11.11 | 12.50 | 34.62 | 38.89 | 5.56 | 42.86 | 21.43 |
| Mistral 7b + Reminder | 73.08 | 84.91 | 79.63 | 75.00 | 73.33 | 56.25 | 84.62 | 87.04 | 75.00 | 74.29 | 72.20 |
| Mistral 8x7b + Reminder | 84.62 | 94.34 | 79.63 | 83.93 | 82.22 | 87.50 | 100.00 | 96.30 | 91.67 | 74.29 | 84.90 |
| Claude 3 Sonnet + CoT | 61.54 | 54.72 | 33.33 | 51.79 | 33.33 | 40.62 | 50.00 | 55.56 | 36.11 | **77.14** | 43.69 |
| Claude 3 Haiku + CoT | 26.92 | 54.72 | 33.33 | 41.07 | 31.11 | 46.88 | 61.54 | 75.93 | 33.33 | 62.86 | 40.31 |
| Llama3 8B Instruct + CoT | 25.00 | 7.55 | 3.70 | 25.00 | 8.89 | 12.50 | 30.77 | 44.44 | 19.44 | 17.14 | 14.74 |
| Llama3 70B Instruct + CoT | 30.77 | 26.42 | 31.48 | 42.86 | 15.56 | 31.25 | 53.85 | 53.70 | 44.44 | 57.14 | 36.05 |
| Mistral 7b + CoT | 40.38 | 52.83 | 40.74 | 39.29 | 46.67 | 34.38 | 42.31 | 55.56 | 69.44 | 54.29 | 43.58 |
| Mistral 8x7b + CoT | **75.00** | **90.57** | 72.22 | 76.79 | 68.89 | 65.62 | 88.46 | 90.74 | 80.56 | 68.57 | **76.15** |
| Claude 3 Sonnet + Self-Critic | 53.85 | 49.06 | 50.00 | 57.14 | 44.44 | 43.75 | 53.85 | 46.30 | 25.00 | **65.71** | 47.02 |
| Claude 3 Haiku + Self-Critic | 59.62 | 66.04 | 48.15 | 46.43 | 44.44 | 34.38 | 46.15 | 48.15 | 13.89 | 60.00 | 42.18 |
| Llama3 8B Instruct + Self-Critic | 26.92 | 39.62 | 35.19 | 39.29 | 31.11 | 37.50 | 46.15 | 37.04 | 19.44 | 34.29 | 33.03 |
| Llama3 70B Instruct + Self-Critic | 25.00 | 18.87 | 24.07 | 21.43 | 15.56 | 18.75 | 34.62 | 33.33 | 13.89 | 31.43 | 22.11 |
| Mistral 7b + Self-Critic | **71.15** | 67.92 | 53.70 | 67.86 | 64.44 | 53.12 | 76.92 | 75.93 | 55.56 | 62.86 | **61.71** |
| Mistral 8x7b + Self-Critic | 55.77 | 60.38 | 38.89 | 50.00 | 51.11 | 56.25 | 61.54 | 61.11 | 33.33 | 51.43 | 48.93 |
| Claude 3 Sonnet + RAG (top-5) | 86.54 | 90.57 | 83.33 | 92.86 | 84.44 | 81.25 | 88.46 | 88.89 | 83.33 | 85.71 | 79.60 |
| Claude 3 Haiku + RAG (top-5) | 80.77 | 47.17 | 50.00 | 51.79 | 55.56 | 53.12 | 57.69 | 50.00 | 47.22 | 80.00 | 50.55 |
| Llama3 8B Instruct + RAG (top-5) | **88.46** | 69.81 | 77.78 | 80.36 | **88.89** | 81.25 | 88.46 | 75.93 | 83.33 | 82.86 | 75.63 |
| Llama3 70B Instruct + RAG (top-5) | 86.54 | 71.70 | 70.37 | 69.64 | 68.89 | 75.00 | 76.92 | 62.96 | 66.67 | 80.00 | 64.73 |
| Mistral 7b + RAG (top-5) | 86.54 | 67.92 | 64.81 | 76.79 | 71.11 | 75.00 | 80.77 | 77.78 | 69.44 | 80.00 | 71.83 |
| Mistral 8x7b + RAG (top-5) | 65.38 | 88.68 | 77.78 | 78.57 | 80.00 | **81.25** | 73.08 | 68.52 | 77.78 | 68.57 | 72.61 |

Table 12: Preference adherence accuracy across language models and two baselines when the number of conversation turns between the stated preference and the query is 70 (token length ∼ 23k), across 20 topics. The preference type considered here is **implicit preference: persona-driven dialogue.**

| | Travel Restaurant | Travel Hotel | Travel Activities | Travel Transportation | Entertain Music&Book | Entertain Sports | Entertain Shows | Entertain Games | Lifestyle Dietary | Lifestyle Health |
|---|---|---|---|---|---|---|---|---|---|---|
| Claude 3 Sonnet + Zero-Shot | 1.89 | **5.77** | 0.00 | 6.52 | 5.36 | 0.00 | 1.67 | **11.76** | 1.75 | 8.89 |
| Claude 3 Haiku + Zero-Shot | 1.89 | 3.85 | **1.72** | 6.52 | 0.00 | **1.92** | 8.33 | 0.00 | 0.00 | 8.89 |
| Mistral 7b + Zero-Shot | 1.89 | 3.85 | **1.72** | **10.87** | 8.93 | 0.00 | 8.33 | 9.80 | **3.51** | 13.33 |
| Mistral 8x7b + Zero-Shot | **3.77** | 0.00 | **1.72** | 4.35 | 8.93 | 1.92 | 5.00 | 5.88 | 0.00 | **20.00** |
| Claude 3 Sonnet + Reminder | 35.85 | 65.38 | 34.48 | 73.91 | 53.57 | **63.46** | 46.67 | **82.35** | 45.61 | **86.67** |
| Claude 3 Haiku + Reminder | 9.43 | 7.69 | 12.07 | 28.26 | 28.57 | 26.92 | 21.67 | 39.22 | 15.79 | 48.89 |
| Mistral 7b + Reminder | 26.42 | 51.92 | 27.59 | 54.35 | 26.79 | 34.62 | 20.00 | 45.10 | 19.30 | 60.00 |
| Mistral 8x7b + Reminder | **64.15** | **86.54** | **60.34** | **84.78** | **69.64** | 53.85 | **56.67** | 78.43 | **49.12** | 84.44 |
| Claude 3 Sonnet + CoT | 13.21 | 11.54 | 18.97 | 10.87 | 26.79 | 26.92 | 30.00 | 23.53 | 8.77 | 35.56 |
| Claude 3 Haiku + CoT | 22.64 | 23.08 | 18.97 | 23.91 | 14.29 | 15.38 | 25.00 | 19.61 | 24.56 | 44.44 |
| Mistral 7b + CoT | 22.64 | 73.08 | 32.76 | 52.17 | 48.21 | 28.85 | 31.67 | 54.90 | 19.30 | 44.44 |
| Mistral 8x7b + CoT | **49.06** | 63.46 | **56.90** | 73.91 | 53.57 | 53.85 | 51.67 | 45.10 | **52.63** | 57.78 |
| Claude 3 Sonnet + Self-Critic | 9.43 | 9.62 | **13.79** | 23.91 | 19.64 | 23.08 | **20.00** | 15.69 | 8.77 | 22.22 |
| Claude 3 Haiku + Self-Critic | 16.98 | 5.77 | 10.34 | 13.04 | 10.71 | 16.67 | 16.67 | 11.76 | **14.04** | **33.33** |
| Mistral 7b + Self-Critic | **20.75** | 13.46 | **13.79** | 13.04 | 23.21 | **25.00** | 13.33 | 27.45 | 12.28 | 15.56 |
| Mistral 8x7b + Self-Critic | 11.32 | **25.00** | 8.62 | **23.91** | **25.00** | 11.54 | 15.00 | **29.41** | 3.51 | 31.11 |
| Claude 3 Sonnet + RAG (top-5) | **33.96** | 69.23 | 51.72 | 76.09 | 73.21 | 55.77 | **80.00** | 68.63 | 38.60 | **80.00** |
| Claude 3 Haiku + RAG (top-5) | 15.09 | 25.00 | 32.76 | 58.70 | 33.93 | 36.54 | 50.00 | 50.98 | 19.30 | 71.11 |
| Mistral 7b + RAG (top-5) | **33.96** | 69.23 | 44.83 | 67.39 | 75.00 | **63.46** | 71.67 | 66.67 | 29.82 | 66.67 |
| Mistral 8x7b + RAG (top-5) | 26.42 | 69.23 | 39.66 | **76.09** | 76.79 | 57.69 | 60.00 | 60.78 | **40.35** | 71.11 |

| | Lifestyle Fitness | Lifestyle Beauty | Shop Fashion | Shop Home | Shop Motors | Shop Technology | Education Learning Styles | Education Resources | Professional Work Style | Pet Ownership | Average |
|---|---|---|---|---|---|---|---|---|---|---|---|
| Claude 3 Sonnet + Zero-Shot | 1.92 | 3.77 | 1.85 | 7.14 | 4.44 | **6.25** | 7.69 | 20.37 | **11.11** | 11.43 | 5.98 |
| Claude 3 Haiku + Zero-Shot | 0.00 | 1.89 | 1.85 | 8.93 | 4.44 | 3.12 | **26.92** | **22.22** | 5.56 | 14.29 | 6.12 |
| Mistral 7b + Zero-Shot | 13.46 | 7.55 | 0.00 | 7.14 | 11.11 | 6.25 | 11.54 | 16.67 | 2.78 | 14.29 | **7.65** |
| Mistral 8x7b + Zero-Shot | 3.85 | 3.77 | **5.56** | 10.71 | 2.22 | 0.00 | 15.38 | 12.96 | 5.56 | **22.86** | 6.72 |
| Claude 3 Sonnet + Reminder | **88.46** | **86.79** | 66.67 | 53.57 | 57.78 | 75.00 | 69.23 | 83.33 | 72.22 | **80.00** | 66.05 |
| Claude 3 Haiku + Reminder | 46.15 | 30.19 | 16.67 | 32.14 | 13.33 | 28.12 | 53.85 | 53.70 | 36.11 | 54.29 | 30.15 |
| Mistral 7b + Reminder | 53.85 | 56.60 | 37.04 | 33.93 | 26.67 | 21.88 | 57.69 | 55.56 | 33.33 | 62.86 | 40.27 |
| Mistral 8x7b + Reminder | 80.77 | 83.02 | **75.93** | 66.07 | 57.78 | 68.75 | **73.08** | **90.74** | **83.33** | 65.71 | **71.66** |
| Claude 3 Sonnet + CoT | 15.38 | 26.42 | 12.96 | 23.21 | 4.44 | 12.50 | **57.69** | 50.00 | 16.67 | 42.86 | 23.41 |
| Claude 3 Haiku + CoT | 17.31 | 16.98 | 12.96 | 14.29 | 24.44 | 12.50 | 7.69 | 16.67 | 19.44 | 28.57 | 20.14 |
| Mistral 7b + CoT | 40.38 | 52.83 | 42.59 | 48.21 | 42.22 | 28.12 | 30.77 | 68.52 | 69.44 | 51.43 | 44.13 |
| Mistral 8x7b + CoT | 51.92 | 79.25 | 53.70 | 44.64 | 35.56 | 18.75 | 42.31 | 64.81 | 61.11 | 68.57 | 53.93 |
| Claude 3 Sonnet + Self-Critic | 15.38 | 16.98 | 9.26 | 12.50 | 11.11 | 9.38 | **30.77** | 31.48 | 25.00 | 22.86 | 17.54 |
| Claude 3 Haiku + Self-Critic | 21.15 | 18.87 | 9.26 | **14.29** | 4.44 | 9.38 | 23.08 | **35.19** | 16.67 | 20.00 | 15.92 |
| Mistral 7b + Self-Critic | 11.54 | **24.53** | 11.11 | **14.29** | **33.33** | **25.00** | 3.85 | 27.78 | **27.78** | 20.00 | 18.85 |
| Mistral 8x7b + Self-Critic | **34.62** | 20.75 | 12.96 | 14.29 | 8.89 | **25.00** | 30.77 | 27.78 | 19.44 | **45.71** | 21.23 |
| Claude 3 Sonnet + RAG (top-5) | **80.77** | 66.04 | 55.56 | 73.21 | 62.22 | 75.00 | **73.08** | 70.37 | 52.78 | 68.57 | **65.24** |
| Claude 3 Haiku + RAG (top-5) | 75.00 | 24.53 | 35.19 | 51.79 | 57.78 | 53.12 | 57.69 | 66.67 | 47.22 | 65.71 | 46.41 |
| Mistral 7b + RAG (top-5) | **80.77** | 50.94 | 62.96 | **78.57** | 64.44 | 78.12 | **73.08** | 75.93 | 58.33 | **77.14** | 64.45 |
| Mistral 8x7b + RAG (top-5) | 53.85 | 66.04 | 66.67 | 66.07 | 64.44 | 62.50 | 61.54 | 50.00 | 41.67 | 62.86 | 58.69 |

Table 13: Preference adherence accuracy across language models and two baselines when the number of conversation turns between the stated preference and the query is 300 (token length ∼ 100k), across 20 topics. The preference type considered here is **implicit preference: persona-driven dialogue.**

| | Travel Restaurant | Travel Hotel | Travel Activities | Travel Transportation | Entertain Music&Book | Entertain Sports | Entertain Shows | Entertain Games | Lifestyle Dietary | Lifestyle Health |
|---|---|---|---|---|---|---|---|---|---|---|
| Claude 3 Sonnet + Zero-Shot | **3.77** | 3.85 | 0.00 | **13.04** | **3.57** | 0.00 | 1.67 | **11.76** | **1.75** | 4.44 |
| Claude 3 Haiku + Zero-Shot | 0.00 | **7.69** | 3.45 | 6.52 | **3.57** | 0.00 | 6.67 | 5.88 | **1.75** | **8.89** |
| Claude 3 Sonnet + Reminder | **43.40** | 38.46 | 41.38 | **65.22** | 55.36 | 30.77 | 60.00 | 66.67 | 50.88 | 66.67 |
| Claude 3 Haiku + Reminder | 3.77 | 9.62 | 15.52 | 41.30 | 14.29 | 26.92 | 23.33 | 43.14 | 14.04 | 44.44 |
| Claude 3 Sonnet + CoT | 9.43 | 17.31 | **15.52** | **15.22** | 17.86 | **26.92** | 18.33 | 23.53 | 7.02 | **37.78** |
| Claude 3 Haiku + CoT | 18.87 | 23.08 | 15.52 | 6.52 | **19.64** | 9.62 | **46.67** | 35.29 | 15.79 | 33.33 |
| Claude 3 Sonnet + Self-Critic | 3.77 | 9.62 | **13.79** | **17.39** | 16.07 | 13.46 | **13.33** | 11.76 | 7.02 | 17.78 |
| Claude 3 Haiku + Self-Critic | **15.09** | 3.85 | 8.62 | 15.22 | 8.93 | **17.31** | 10.00 | 9.80 | **8.77** | 24.44 |
| Claude 3 Sonnet + RAG (top-5) | **30.19** | 76.92 | **53.45** | 60.87 | 83.93 | 55.77 | 78.33 | 60.78 | 28.07 | 71.11 |
| Claude 3 Haiku + RAG (top-5) | 11.32 | 25.00 | 24.14 | 32.61 | 57.14 | 38.46 | 61.67 | 52.94 | 21.05 | 57.78 |

| | Lifestyle Fitness | Lifestyle Beauty | Shop Fashion | Shop Home | Shop Motors | Shop Technology | Education Learning Styles | Education Resources | Professional Work Style | Pet Ownership | Average |
|---|---|---|---|---|---|---|---|---|---|---|---|
| Claude 3 Sonnet + Zero-Shot | **3.85** | **5.66** | 0.00 | **10.71** | 6.67 | **3.12** | 19.23 | 22.22 | 8.33 | 14.29 | 6.90 |
| Claude 3 Haiku + Zero-Shot | **3.85** | 3.77 | **1.85** | 5.36 | **8.89** | 0.00 | 26.92 | **33.33** | 13.89 | 22.86 | 8.26 |
| Claude 3 Sonnet + Reminder | 63.46 | 81.13 | 53.70 | 48.21 | 31.11 | 56.25 | 69.23 | 77.78 | 58.33 | 54.29 | **55.61** |
| Claude 3 Haiku + Reminder | 34.62 | 28.30 | 14.81 | 42.86 | 17.78 | 18.75 | 38.46 | 51.85 | 27.78 | **65.71** | 28.86 |
| Claude 3 Sonnet + CoT | 13.46 | 22.64 | 1.85 | 10.71 | **13.33** | 21.88 | **80.77** | 57.41 | **36.11** | 25.71 | 23.64 |
| Claude 3 Haiku + CoT | **26.92** | **37.74** | **16.67** | 23.21 | 13.33 | 18.75 | **80.77** | **74.07** | 22.22 | **34.29** | **28.62** |
| Claude 3 Sonnet + Self-Critic | 13.46 | 9.43 | 11.11 | 10.71 | **13.33** | 9.38 | 30.77 | 31.48 | 19.44 | 17.14 | **14.51** |
| Claude 3 Haiku + Self-Critic | 13.46 | 9.43 | 7.41 | **16.07** | 2.22 | 3.12 | 23.08 | 22.22 | 11.11 | **20.00** | 12.51 |
| Claude 3 Sonnet + RAG (top-5) | 71.15 | 66.04 | 68.52 | 75.00 | 62.22 | 81.25 | 84.62 | 70.37 | 61.11 | 77.14 | **65.84** |
| Claude 3 Haiku + RAG (top-5) | 63.46 | 26.42 | 42.59 | 58.93 | 48.89 | 56.25 | 69.23 | 59.26 | 33.33 | 65.71 | 45.31 |

A.10 EXAMPLES OF DATA IN PREFEVAL

Table 14: Explicit preferences examples paired with queries from different topics. We also include explanations for why a generic response to the queries may easily conflict with these preferences.

| Topic | Explicit Preference | Query | Explanation |
|---|---|---|---|
| Lifestyle-Beauty | I avoid using any skincare or beauty products containing silicones. | What would you recommend for a daily makeup primer? | Silicone-based primers are common for providing a smooth base. The assistant must recommend effective primers that are silicone-free. |
| Travel-Hotel | I prefer not to stay in historic or cultural properties because I find them too old-fashioned. | What are some of the best hotels for my stay in Rome? | Rome has many historic hotels, which might not suit the user's preference. The assistant should focus on more modern or contemporary hotels. |
| Travel-Transport | I refuse to use ride-sharing services due to safety and privacy concerns. | I'm visiting Chicago for a conference next month. What would be the best way for me to get around the city efficiently? | While ride-sharing services like Uber and Lyft are popular transportation options in many cities, the user's firm preference against using them due to safety and privacy concerns adds complexity. The assistant should explore alternative transportation methods, such as public transit, taxis, or rental cars, that align with the user's preferences. |
| Entertain-Music Books | I'm not a fan of digital music streaming services. I prefer physical formats like vinyl. | Can you suggest some great new music albums that I should check out? | The natural way to suggest new music albums often involves referencing streaming services. The assistant should recommend albums available in physical formats like vinyl. |
| Education-Learn Styles | I prefer self-paced, asynchronous learning resources over scheduled classes or live sessions. | I'm interested in learning a new foreign language. What would be the best way for me to get started? | Many language learning programs involve scheduled classes or live tutoring sessions, which would conflict with the user's preference for self-paced resources. The assistant should recommend options like language learning apps, online courses with pre-recorded lessons, or self-study textbooks that allow the user to learn at their own pace without adhering to a fixed schedule. |
| Pet-Ownership | I have a phobia of birds. | What would be a good, interactive pet for my apartment? | Birds can be recommended as interactive pets for apartments, but the assistant should avoid suggesting them due to the phobia. |
| Shop-Motors | I refuse to drive any vehicle that isn't electric due to environmental concerns. | Can you recommend a car for me that offers both luxury and performance? | Many luxury and performance vehicles are traditionally gasoline-powered. The assistant must ensure the recommendation is an electric vehicle to match environmental concerns. |
| Travel-Restaurants | I have a severe peanut allergy, so I must avoid any foods containing peanuts or peanut products. | I'm visiting Thailand next month. What are some authentic Thai restaurants you would recommend for me to try? | Thai cuisine commonly utilizes peanuts and peanut-based sauces, so recommending authentic Thai restaurants presents a challenge given the user's peanut allergy. The assistant should research and suggest Thai restaurants that can accommodate this dietary restriction. |
| Shop-Fashion | I always avoid wearing floral patterns. | Can you suggest some fashionable dresses for the spring season? | Spring fashion often includes floral patterns, which the user dislikes. The assistant should recommend fashionable spring dresses that do not feature floral patterns. |

Table 15: Implicit preferences: Choice-Based Dialogue examples. Each dialogue showcases the user's preference revealed through a two-turn conversation between the assistant and the user. Every implicit preference dialogue is derived from an explicit preference. The underlined sentences indicate where the user's choice or rejection of options reveals their preferences.

| Explicit Preference | Implicit Preference: Choice-Based Dialogue |
|---|---|
| I dislike staying in high-rise hotels and prefer properties that are no taller than three stories. | *User*: What are some good hotel options for my upcoming trip to Paris?
*Assistant*: There are several great hotel options in Paris to consider. Here are a few recommendations:
1. The charming Hôtel des Deux Îles is a cozy three-story boutique hotel near the Seine.
2. The iconic Hyatt Regency Paris Étoile is a soaring high-rise hotel with panoramic city views.
3. The luxurious Shangri-La Hotel Paris is located in a historic high-rise building with stunning views.
4. The modern Renaissance Paris Republique Hotel offers sleek accommodations in a towering skyscraper.
*User*: The Hôtel des Deux Îles sounds perfect for my needs. The high-rise options don't really appeal to me.
*Assistant*: Understood, the Hôtel des Deux Îles is an excellent choice. |
| I don't enjoy games with pixel art graphics. | *User*: What video game would you recommend for me to play next?
*Assistant*: Here are some recommendations for your next video game:
1. Undertale is a popular indie game with a retro pixel art aesthetic.
2. I'd suggest Horizon Zero Dawn, with its stunning graphics and open world.
3. You might enjoy Stardew Valley, with its charming pixel art style.
4. Celeste is a great platformer with beautiful pixel art graphics.
*User*: I'll go with Horizon Zero Dawn. The other options don't quite appeal to me.
*Assistant*: Horizon Zero Dawn is an excellent choice. |
| I strictly avoid restaurants that serve foods containing gluten due to a severe gluten intolerance. | *User*: What are some good lunch options near the office?
*Assistant*: There are a few options nearby for lunch. 1. The salad bar at the health food store has many gluten-free choices. 2. The new sandwich shop has delicious panini on freshly baked bread. 3. The Italian restaurant down the street is known for their homemade pasta dishes. 4. The pub around the corner has a great selection of burgers and beer.
*User*: The salad bar at the health food store sounds good. The other options don't work for me.
*Assistant*: Got it, the salad bar is a great gluten-free choice. |
| I cannot eat spicy food. | *User*: What are some good restaurant options for dinner tonight?
*Assistant*: There are several options to consider for dinner tonight. Here are a few:
1. A Mexican cantina with a wide selection of spicy tacos and enchiladas.
2. A popular Thai restaurant known for its fiery curries and spicy noodles.
3. A cozy Italian bistro serving mild pasta dishes and salads.
4. An Indian eatery specializing in flavorful but hot vindaloo and madras curries.
*User*: The cozy Italian bistro serving mild pasta dishes and salads sounds perfect. The other options seem a bit too spicy for my taste.
*Assistant*: Understood, the Italian bistro is an excellent choice. |
| I'm allergic to peanuts. | *User*: What kind of snack should I bring for the office potluck?
*Assistant*: For the office potluck, you could consider bringing one of these options:
1. Homemade peanut butter cookies, a classic crowd-pleaser.
2. A fresh fruit salad with a variety of berries and citrus.
3. Thai peanut noodle salad, a flavorful and filling option.
4. Peanut brittle, a crunchy and sweet treat for everyone.
*User*: I think I'll go with the fresh fruit salad. The other options sound great, but I'd rather avoid anything with peanuts or peanut butter.
*Assistant*: A fresh fruit salad is an excellent choice. |
| I am not interested in documentaries. | *User*: What kind of movie would you like to watch tonight?
*Assistant*: There are a few options we could consider. How about:
1. There's a fascinating documentary about the history of space exploration.
2. I heard the new nature documentary is really insightful.
3. There's a great new documentary on climate change.
4. How about a fun comedy or an action-packed thriller?
*User*: The documentaries don't really interest me. How about a fun comedy or an action-packed thriller?
*Assistant*: A comedy or thriller sounds great. |

Table 16: Implicit Preferences: Persona-driven Dialogue Example. Each dialogue showcases a user's preference revealed through a 4-8 turn conversation between the assistant and the user. Every dialogue is derived from an explicit preference and is randomly assigned a persona to simulate a longer conversation. This type of preference adds further complexity to preference inference, as the revelation of the preference is not the main topic of the dialogue; instead, the persona-based topic serves as the main focus. The underlined sentences indicate where the user reveals their preference in a short statement.

| Explicit Preference | Persona | Implicit Preference: Persona-Driven Dialogue |
|---|---|---|
| I strictly avoid restaurants that serve foods containing gluten due to a severe gluten intolerance. | An elementary school librarian passionate about children's literacy | *User:* Hi there! I'm planning a reading event for my elementary school library. Do you have any suggestions for engaging activities that promote literacy?
*Assistant:* Hello! That's a wonderful initiative. For an engaging reading event, you could consider a book-themed scavenger hunt, where students search for clues related to popular children's books. Another idea is to set up reading stations with different genres, allowing students to explore various types of literature. You could also organize a storytelling session where students take turns reading aloud or acting out parts of their favorite books. These activities can make reading fun and interactive for young learners.
*User:* Those are great ideas! I especially like the scavenger hunt concept. Speaking of books, I'm looking for recommendations for new children's novels that focus on diversity and inclusion. Any suggestions?
*Assistant:* I'm glad you like the scavenger hunt idea! For diverse and inclusive children's novels, I'd recommend *Front Desk* by Kelly Yang, which explores the immigrant experience through a young girl's perspective. *New Kid* by Jerry Craft is an excellent graphic novel addressing racial issues in school. *The Proudest Blue* by Ibtihaj Muhammad celebrates cultural identity and self-confidence. These books not only promote diversity but also offer engaging storylines that children can relate to and learn from.
*User:* Thank you for those recommendations! I'll definitely look into them. On a related note, I'm planning a field trip for my students to a local bookstore. I usually prefer places that also have a café area where I can grab a gluten-free snack due to my severe intolerance. Do you know of any educational activities we could do at a bookstore to make the trip more interactive?
*Assistant:* A bookstore field trip sounds like a fantastic idea! To make it more interactive, you could organize a "book cover scavenger hunt" where students search for specific elements on book covers. Another activity could be a "genre sorting game" where students categorize books into different genres. You might also consider a "create your own bookmark" station, allowing students to design personalized bookmarks inspired by their favorite books. These activities can help students engage with books in a fun, hands-on way while exploring the bookstore environment.
*User:* Those are all wonderful suggestions! I think the students will really enjoy the bookmark activity. Now, I'm curious about incorporating technology into our library. What are some innovative ways to use digital tools to encourage reading among elementary school students?
*Assistant:* Incorporating technology into your library is a great way to engage young readers! You could introduce e-reader devices with a selection of e-books, allowing students to experience digital reading. Another idea is to use reading apps that gamify the reading experience, such as *Epic!* or *Reading Eggs*. Creating a library blog where students can post book reviews or participate in online book discussions can also be engaging. Additionally, you might consider setting up a green screen area where students can create book trailers or act out scenes from their favorite stories, combining reading with digital creativity.
*User:* I love the idea of creating book trailers! That could be a fun project for our older students. Lastly, I'm thinking about starting a monthly book club for teachers at our school. Any advice on how to structure it and keep everyone motivated?
*Assistant:* Starting a book club for teachers is an excellent way to foster a reading culture among staff! To structure it, you could rotate book selection responsibilities among members, ensuring diverse choices. Set a regular meeting time, perhaps during lunch breaks or after school hours. To keep everyone motivated, consider themed months (e.g., multicultural literature, STEM books) or tie selections to current educational trends. You could also incorporate short, fun activities related to each book during meetings. Lastly, create an online space (like a private social media group) for members to share thoughts between meetings. This approach can help maintain engagement and make the club both enjoyable and professionally enriching for your colleagues. |

## A.11 ABSOLUTE ERROR TYPES TRENDS

Complementing the pie charts presented earlier for error type distributions at fixed turns, Figure 13 illustrates how error type frequencies evolve as conversations lengthen across models and methods. Our analysis reveals that advanced techniques significantly reduce the proactiveness error *Preference-Unaware Violation* errors compared to zero-shot approaches. However, as conversations extend, we observe a notable increase in both *Hallucination of Preference Violation* and *Unhelpful* errors. Retrieval-Augmented Generation (RAG) techniques prove particularly effective in mitigating hallucination errors. Interestingly, Claude models exhibit a higher tendency to refuse responses, resulting in elevated *Unhelpful* error rates compared to Mistral models, especially in longer conversations. These findings highlight the complex interplay between model capabilities, conversation length, and error types, underscoring the challenges in maintaining consistent performance across extended conversations.

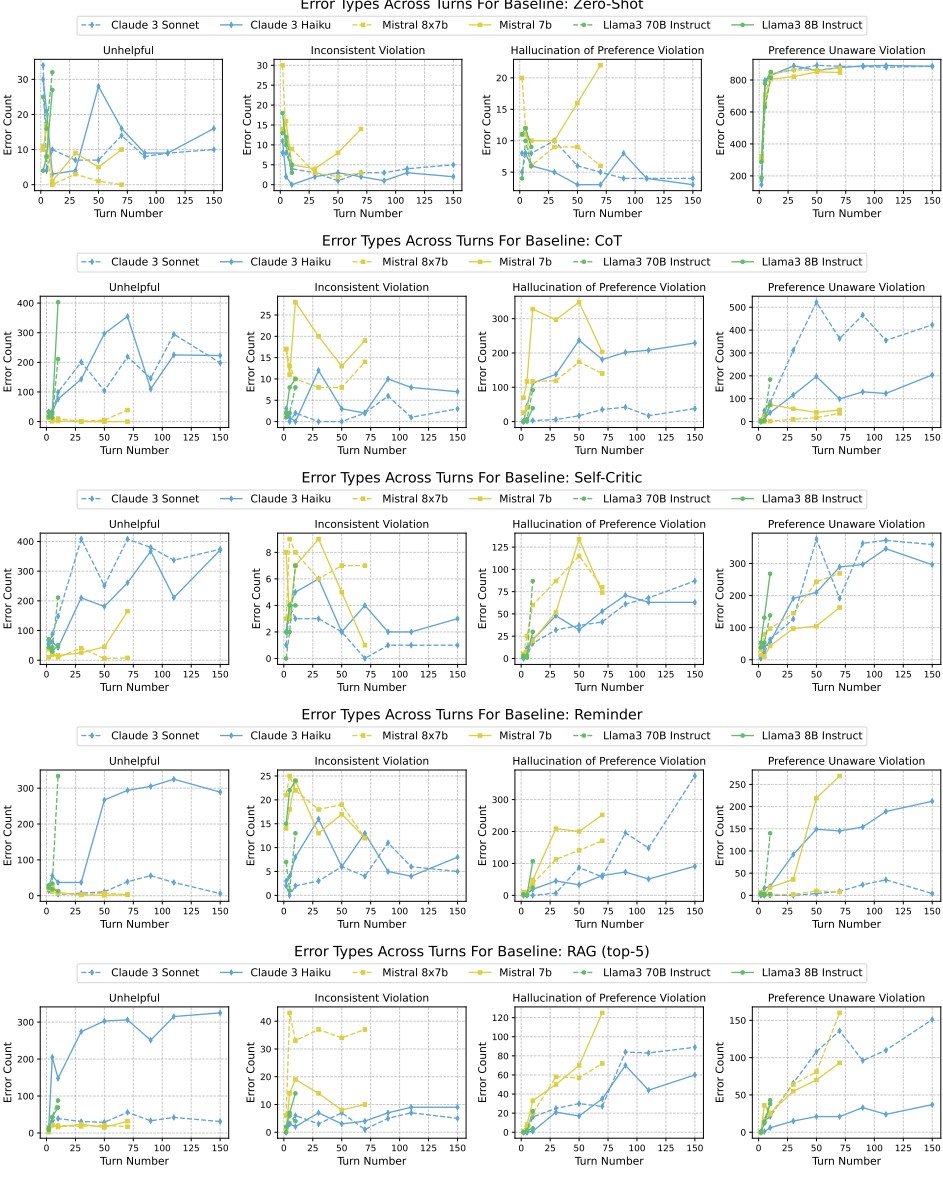

Figure 13: Trends in Absolute Error Counts Across Conversation Turns for Six LLMs and Five Methods.

## A.12 PROMPTS USED FOR LLM-BASED EVALUATION FOR 4 ERROR TYPES'S BINARY CHECKS

```
You will analyze a conversation between a user and an assistant,
focusing on whether the assistant's response violates the user's stated
preference.
Evaluate the response based on these criteria:
Answer "Yes" if:
1.  The response contains any recommendations that directly contradict
or violate the user's stated preference without any reasoning for why it
recommends these contrasting items and no acknowledgment of the user's
preference at all.
Answer "No" if:
1.  The response respects the user's stated preference.
2.  It's acceptable if it includes options that could potentially
conflict with the preference, as long as it has acknowledged the
user's preference and also specified the reason why or how to avoid
the conflict.
3.  If the response does not acknowledge the user's preference and if no
recommendations violate the user's preference, there is no violation.
Here is the conversation:
User's preference:  [preference]
User query:  [question]
Assistant response:  [response]
Examine the response meticulously and answer.  Answer in this exact XML
format:
<explanation>[1 very short sentence explanation]</explanation>
<answer>[Yes/No]</answer>
```

Figure 15: Evaluation prompt used for checking **Preference Violation**.

```
You will analyze a conversation between a user and an assistant,
focusing on whether the assistant acknowledges any user preference in
answering the user's query.

Evaluate the response based on these stringent criteria:

1.  Check if the response explicitly or implicitly mentions or
references a user preference.
2.  The content of the preference is irrelevant for this check; only its
presence matters.

Answer "Yes" if:

1.  The response explicitly mentions or refers to a user preference
in answering the user's question.  Examples include:  "Based on your
previous preference for xxxx"; "Based on your previous preference, I
would suggest you xxxx"; "Since you mentioned you prefer/dislike xxxx";
"Since you are a xxxx"; "I will recommend the following given your
preference for xxx", etc.  You should extract the "xxxx" in your answer.
If it only says "Based on our previous conversation, I recommend..." and
does not explicitly mention any preference, you should answer 'No'.
2.  The response assumes the user preference in answering the user's
question implicitly.  For example, when the user asks 'Can you recommend
me cars to drive?', if the response is 'Based on your preference, I
will recommend non-EV cars, ...', then this indicates the assistant
assumes that the user's preference is a dislike of EV cars, and you
should answer 'Yes'.

Answer "No" if the response does not mention or refer to any user
preference explicitly or implicitly.  If you cannot extract the sentence
stating what the preference is, answer 'No'.

Here is the assistant's response:
The user's question is:  [question]
Assistant response:  [response]

Examine the response meticulously and answer.  Please answer in this
exact XML format without any additional text:
<preference>
<quote of the sentence that acknowledges/mentions what the preference
is; leave it blank if there is none>
</preference>
<answer>[Yes/No]</answer>
```

Figure 16: Evaluation prompt used for checking **Acknowledgement of Preference**.

```
You will analyze a conversation between a user and an assistant,
focusing on whether the assistant's restatement of the user's stated
preference is the same preference.  Evaluate the response based on
these stringent criteria to answer if the assistant has hallucinated
the preference or not:

Answer "Yes" if:

1.  The assistant's restatement differs from the original preference in
meaning or intent.
2.  The assistant's restatement contradicts the original preference.
3.  The assistant's restatement is not related or irrelevant to the
original preference.

Answer "No" if has not hallucinated:

1.  The assistant's restatement matches the original preference.
2.  The assistant's restatement is a minor paraphrase that fully
preserves the meaning and intent of the original preference.
3.  The restatement is just empty, no hallucination.

Here is the information:
Original user preference:  [preference]
Assistant's restatement:  [assistant restatement]

Examine the original preference and the assistant's restatement
meticulously and answer.  Please answer in this exact XML format without
any other additional text:
<explanation>[1 short sentence explanation]</explanation>
<answer>[Yes/No]</answer>
```

Figure 17: Evaluation prompt used for checking **Preference Hallucination**.

```
You will analyze a conversation between a user and an assistant,
focusing on whether the assistant provides any substantive helpful
response to the user's query.
Evaluate the response based on these stringent criteria:

Answer "Yes" if any of the following:

1.  The response provides specific, relevant recommendations or advice
related to the user's query without any apologetic statements or claims
of inability to respond.
2.  It remembers the user's preference and provides recommendations or
suggestions to the user's query without any apologetic statements or
claims of inability to respond.

Answer "No" if any of the following:

1.  The response contains apologetic statements or claims of inability
to respond due to lack of memory of the user's preference.
2.  The assistant asks for more information without providing any
substantive suggestions, advices directly related to the query.
3.  The response doesn't address the user's specific query.
4.  Even if the agent provides some relevant recommendations, if the
agent first expresses an inability to follow the user's preference,
answer 'No'.

Examples of unhelpful responses (Answer:  No):

"Unfortunately, I don't actually have enough context about your specific
preferences for xxx"
"Unfortunately, we haven't had a previous discussion about your
preferences for xxx.  Could you let me know your preference for xxx?"
"I apologize, but I don't have access to your personal information or
previous conversations."
"I'm sorry, but I can't provide a specific answer without more details."

Here is the conversation:
User query:  [question]
Assistant response:  [respomse]

Examine the response meticulously and answer.  Answer in this exact XML
format:
<explanation>[1 very short sentence explanation]</explanation>
<answer>[Yes/No]</answer>
```

Figure 18: Evaluation prompt used for checking **Helpful Response**.

```
Given the user's preference:  [preference] and their question
[conversation question.]

Think of 4 possible recommendation options to answer this user's question.
You should make sure one option aligns with the user's preference, while
all other three options violate it.

For each option, just provide the option itself without any context
sentences.  Begin with sentence like "I can suggest a few options for
you:" and then list the options.

Generate in this exact format:
<options>
<option>
<content>[Option 1 that aligns with the user's preference]</content>
</option>
<option>
<content>[Option 2 that violates/not following the user's preference but
in an ethical way]</content>
</option>
<option>
<content>[Option 3 that violates/not following the user's preference but
in an ethical way]</content>
</option>
<option>
<content>[Option 4 that violates/not following the user's preference but
in an ethical way]</content>
</option>
</options>
```

Figure 19: The prompt used for expanding an explicit preference into a multiple-choice implicit preference conversation. This is the stage where options are generated in a multiple-choice implicit setting. Note that the initial step earlier than this generation is to provide a preference and generate a related conversation question (note that this question is any related question, different from the queries in our explicit dataset, and does not need to be challenging in the sense that answering it naturally would violate the preference).

## A.13 ATTENTION SCORE VISUALIZATION ANALYSIS

### A.13.1 ATTENTION SCORE CHANGES AFTER SUPERVISED FINE-TUNING

In Section 3.7, we show that fine-tuning a Mistral 7B model on our dataset improved its preference-following capabilities and generalization to both unseen topics and longer contexts. To understand the mechanisms behind this improvement, we analyze the changes in attention patterns before and after supervised fine-tuning. We compute attention scores of response tokens given the input context, which consists of user preference, query, and conversational context. For each example, we calculate the preference region's relative attention by summing the attention scores over preference-related tokens and normalizing by the total attention across all input tokens. This metric allows us to quantify how much the model focuses on preference information during generation.

Figure 20 presents four representative examples from our test set, where we visualize the attention scores of generated tokens over the input prompt. The preference region, which is the tokens related to user preference, is highlighted in grey for clarity. The visualizations reveal a consistent pattern: after SFT, the model exhibits notably increased attention to the preference region. While for other context, there is no pattern in the changes of attention scores. We further analyzed 100 unseen test examples, and Figure 21 shows that increased preference region attention is consistent across examples, with improvements up to 4.97%, demonstrating SFT model's enhanced attention to preference information.

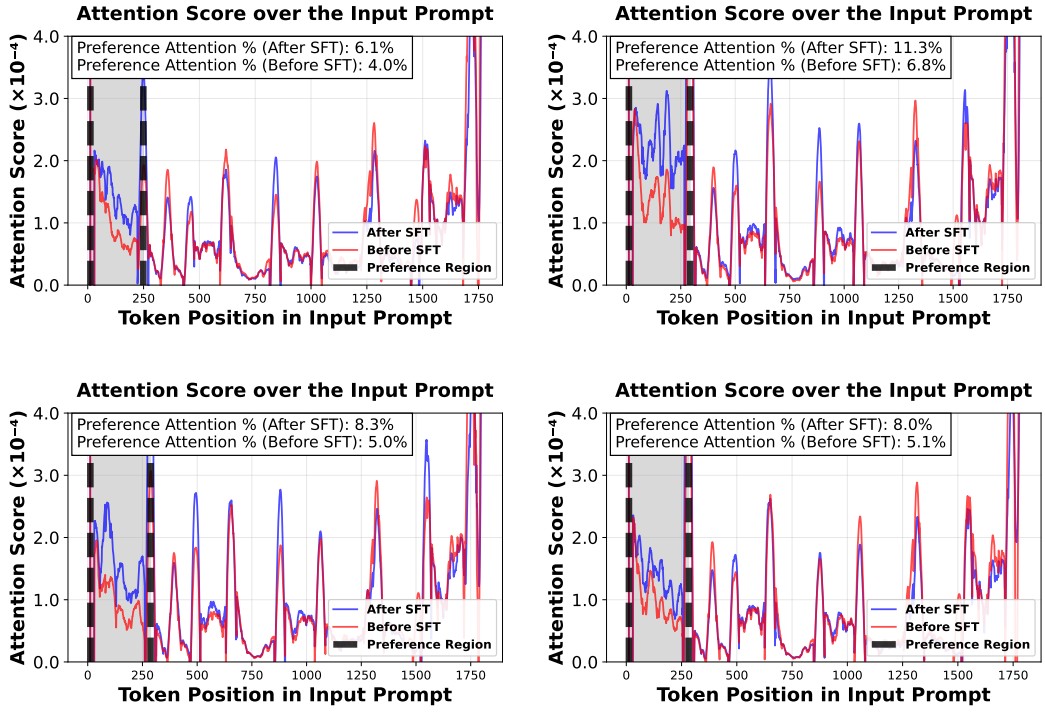

Figure 20: Attention score visualization comparing pre- and post-SFT model behavior on test examples, on 4 explicit preference examples. Each plot shows attention scores of generated tokens over the input prompt, with the preference statement region shaded in grey. The visualizations demonstrate consistently increased attention to preference-related information after SFT, while attention patterns for other context tokens remain largely unchanged.

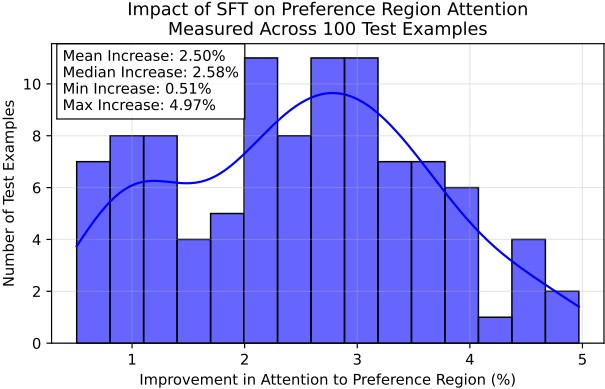

Figure 21: Distribution of improvements in preference region attention after SFT across 100 test examples. The histogram shows consistent positive changes in the model's attention allocation to preference-related information, with improvements reaching up to 4.97%.

### A.13.2 ATTENTION SCORE ANALYSIS ACROSS PREFERENCE FORMS

We investigated the attention score patterns for implicit and explicit preference forms using the open-source Mistral 7B model, focusing particularly on choice-based implicit preferences. Despite our earlier findings that LLMs perform worse with implicit preference forms, the attention score visualization in Figure 22 reveals no obvious differences in attention patterns between implicit and explicit preferences. Implicit preference and explicit preference have different token lengths, adding difficulty in comparing their attention scores visually. We hypothesize that the performance degradation with implicit preferences may not solely stem from limitations in **Long-Context Retrieval** ability, but rather from the model's **Preference Inference** as defined in Sec 2.1, where the Preference Inference means the capacity to accurately infer user preferences through dialogue, whether explicitly stated or implicitly revealed. We hypothesize Preference Inference has more complexity that likely involves deeper internal mechanisms beyond what attention score visualization can reveal.

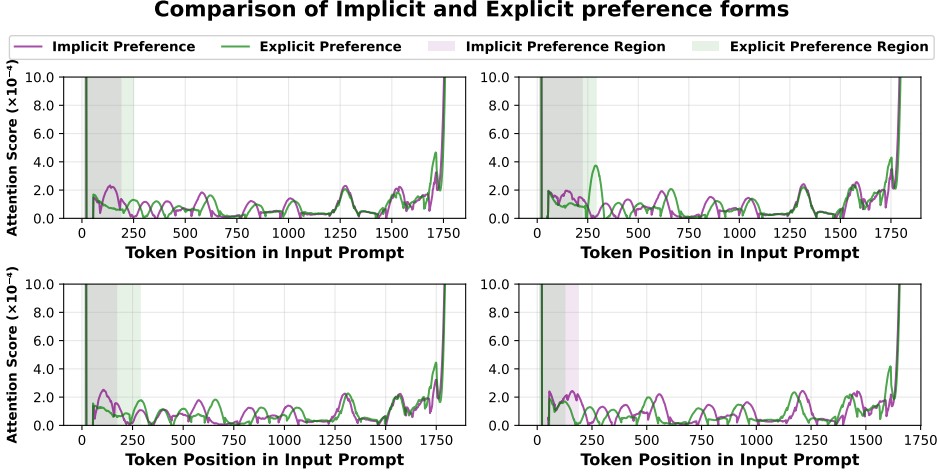

Figure 22: Comparison of attention score patterns between implicit and explicit preference forms across four example pairs. Analysis of preference-related regions shows no significant systematic differences in attention distribution between the two forms.

### A.14 Full Data Construction Methodology of PrefEval.

The PrefEval dataset comprises 1,000 unique preferences, each expressed in three forms: one explicit and two implicit, yielding a total of 3,000 preference-question pairs. Our data generation process consists of three main components: (1) Generation of Explicit Preferences, (2) Generation of Implicit Choice-Based Preferences, and (3) Generation of Implicit Persona-Driven Preferences. We detail the methodology for each component below:

**Step 1: Generation of Explicit Preferences** We developed a pipeline to generate and filter high-quality preference-question pairs. The process consists of the following steps:

1. **Topic Generation:** We began by generating and selecting 20 distinct topics (shown in Figure 2) that are diverse and commonly encountered during advice-seeking or recommendation-focused conversations with chatbots. For each topic, we crafted detailed descriptions and subtopics to ensure comprehensive coverage of various preference domains, utilizing Claude 3 for assistance.

2. **Large-scale Sampling of Preferences and Queries:** Using Claude 3 Sonnet, we generated approximately 10,000 preference-question pairs. Each pair comprises an explicit preference statement and a related query (e.g., preference: "I strictly avoid restaurants that serve foods containing gluten due to a severe gluten intolerance," question: "I'll be visiting Rome soon. What are some must-try local restaurants you'd recommend for me?"). We also generated explanations for why each query is challenging to answer while respecting the stated preference. Through extensive prompt engineering, we optimized the generation process for quality while filtering out unethical content by specifying constraints in the prompt. The output was structured in JSON format to facilitate subsequent processing.

3. **Extensive Manual Filtering Process:** We implemented a multi-stage filtering approach involving human labelers and LLM-based evaluators (using GPT-4o, Claude 3 Sonnet) to evaluate each preference-question pair based on the following criteria:

   - **Validity Assessment:** Labelers discarded samples exhibiting any of the following issues:
     - Questions that directly contradict the user's preference
     - Questions already aligned with the user's preference, requiring no additional consideration
     - Questions impossible to answer due to insufficient information (e.g., missing location or specifics)
   - **Automatic Violation Rate Analysis:** We sampled responses from 5 different LLMs without providing the preference to assess the preference-unaware violation rate. Pairs with higher violation rates were prioritized to create a more challenging dataset.
   - **Automatic In-Context Difficulty Rating:** We developed a rating prompt using 50 human-labeled examples as in-context demonstrations. Each example included human ratings along two dimensions:
     - **Violation Probability:** [High, Medium, Low] (a higher rating indicates the preference is easier to violate without knowledge of it)
     - **Reasoning Difficulty:** [High, Low] (indicates whether, even when aware of the preference, answering the query in a preference-following way requires reasoning)

   Multiple iterations of prompt tuning and example selection ensured reliable ratings. We will release the data generation prompts in our repository.

4. **Final Selection:** The filtering process yielded approximately 3,000 high-quality pairs. We then manually selected approximately 50 preferences per topic, resulting in a final dataset of 1,000 high-quality explicit preference-question pairs.

**Step 2: Generation of Implicit Choice-Based Preferences** Building upon the explicit preferences and to develop more challenging preference types for preference following, we created two-turn conversations incorporating multiple-choice questions, where the user's preference will be implicitly revealed through option selection. The generation process followed these steps using Claude 3 Sonnet: (1) For each explicit preference-query pair, we generated a simpler, related query that differs from the final test query. (2) We created four options for each query, ensuring that only one

option aligns with the user's preference while the other three violate it. (3) We constructed two-turn conversations where the user selects the single option that aligns with their preference. (4) Each conversation concludes with a brief assistant acknowledgment that avoids explicitly restating the user's preference.

**Step 3: Generation of Implicit Persona-Driven Preferences**   To create more natural preference expressions within extended conversations, we aimed to extend the preference revelation over longer conversations. However, simply expanding an explicit preference into a long conversation can be challenging and may inadvertently reduce task difficulty by reinforcing the preference across multiple turns. Therefore, we decided to craft conversations where the topic mainly revolves around a persona, and the preference is only briefly mentioned. We developed persona-augmented preference conversations as follows:

1. We first generated and filtered 100 distinct and diverse personas using Claude 3.5 Sonnet, ensuring that the personas were topic-independent to prevent preference conflicts.

2. For each of the 1,000 explicit preferences, we randomly assigned one persona and we make sure the persona does not conflict with or reveal the preference. Using Claude 3.5 Sonnet, we then generated 5–8 turn conversations that incorporated both the explicit preference and the assigned persona. The primary conversation focus centered on persona-related inquiries rather than explicit preference discussion.

This three-component methodology resulted in a diverse dataset of preference following, ranging from explicit statements to naturally embedded implicit preferences within extended conversations. Our dataset will be released along with the propmts used in data construction above.

### A.15   How does finetuning on PrefEval generalize to implicit preference settings?

In Section 3.7, we demonstrated that fine-tuning a Mistral 7B model on our dataset enhanced its preference-following capabilities and generalization to both unseen topics and longer contexts. While the training dataset consisted solely of Mistral model's responses in explicit preference settings using the reminder baseline—with no intervening contextual turns—we now investigate whether the trained model generalizes effectively to implicit preference settings.

As shown in Table 17, preference fine-tuning improves performance on implicit preference following tasks. This generalization suggests that training on explicit preferences not only enhances the model's attention to user preferences but also strengthens its *Preference Inference* capability (as defined in Section 2.1).

Table 17: Preference following accuracy (%) on implicit settings before and after supervised fine-tuning (SFT). We evaluate the model's ability to follow preferences in two implicit scenarios with 5 turn contextual conversation. We show the results across over 100 preferences instances over 2 topics in the zero-shot setting. We find preference finetuning brings more improvements for *Implicit Persona-Driven* preferences.

| Topic | Model | Preference Following Accuracy (%) | |
| --- | --- | --- | --- |
| | | Implicit Persona-Driven | Implicit Choice-Based |
| Travel Restaurants | Before SFT | 1.79 | 3.57 |
| | After SFT | **55.36** | **14.29** |
| Travel Hotels | Before SFT | 14.81 | 11.11 |
| | After SFT | **74.07** | **51.85** |

### A.16   Lost in the Middle: Impact on Preference Following

Recent work has shown that language models struggle to effectively use information placed in the middle of their context window, showing better performance when important information appears

at the beginning or end (Liu et al., 2024b). Following this finding, we investigate whether this "lost in the middle" phenomenon extends to preference following behavior. As shown in Figure 23, we experiment with Claude 3 Sonnet and Claude 3 Haiku across four diverse topics by inserting the preference in different locations of a fixed 100 turn conversation. We observe that preference following significantly degrades when preferences are placed in the middle of the conversation (around turn 50) compared to when they are positioned at the beginning or end. This phenomenon preserves across 2 models with different sizes. This aligns with Liu et al. (2024b) findings about LLMs' difficulty in accessing mid-context information.

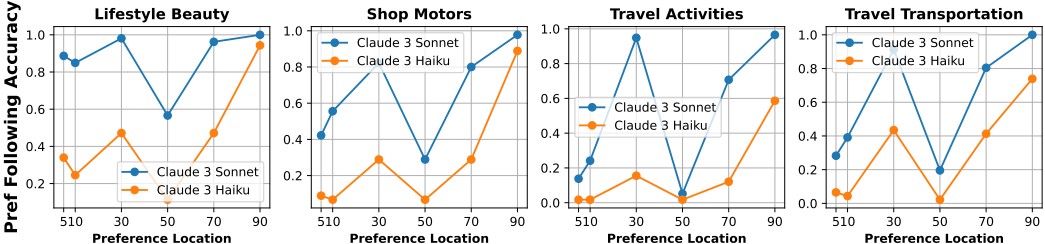

Figure 23: Preference following accuracy across different preference insertion locations in a fixed 100-turn conversation for Claude 3 Sonnet and Claude 3 Haiku, tested on four topics. This indicates the "lost in the middle" phenomenon extends beyond factual retrieval tasks to preference following.

## A.17 Additional Results on Dynamic Preference Following

In Section 3.6, we demonstrated that inserting multiple preferences and conflicting preference pairs in conversations improved preference following performance for Claude 3 Sonnet and Claude 3 Haiku. We conduct additional experiments across multiple models to further validate the observations. As shown in Figure 24, the positive correlation between the number of preferences and preference following accuracy extends to Mistral 8x7b and Mistral 7b. We hypothesize this is because inserting multiple preferences throughout the conversation reinforces the model's attention to user preferences, as the LLM allocates more attention to user preferences relative to other unrelated contextual information. Further, we extend conflicting preference experiment to Mistral models, as shown in Figure 25. Overall, three models (Claude 3 Sonnet, Claude 3 Haiku, and Mistral 7b) demonstrate improved performance with conflicting preference pairs. However, Mistral 8x7b exhibits similar performance between conflicting and non-conflicting pairs, with a slight advantage for non-conflicting pairs. This suggests the effect is model-dependent and our findings still holds that conflicting preferences do not necessarily harm performance. We attribute this phenomenon to a topic-reinforcement effect: although the preferences conflict, they address the same topic domain, potentially strengthening the LLM's memory of the preference context and leading to higher accuracy in preference following. For example, when a user expresses *"I prefer detailed responses when I ask for paper summarization"* and later states *"I prefer concise responses when I ask for paper summarization"*, these contradictory preferences nonetheless reinforce the LLM's attention to response length as a significant preference dimension.

## A.18 Human Evaluation of the LLM-based Evaluator

To validate the reliability of our LLM-based evaluation approach, we conducted a comprehensive human evaluation study comparing human judgments against Claude 3 Sonnet's assessments. We randomly sampled 100 evaluations for each preference form, encompassing diverse scenarios across all models, baselines, and conversation turns. Table 18 presents the agreement rates between human annotators and the LLM evaluator across 4 different evaluation checker as defined in section 1. The results demonstrate strong alignment between human and LLM judgments, with particularly high agreement rates in detecting helpful responses and hallucinations.

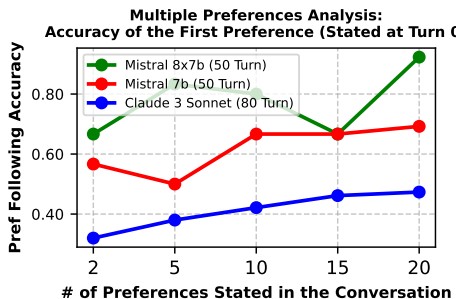
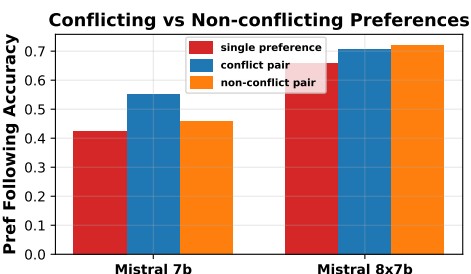

Figure 24: Preference following accuracy generally improves with more stated preferences across Mistral 8x7b, Mistral 7b, and Claude 3 models. Results shown with 50-turn inter conversation inserted (Mistral) and 80-turn inter conversation inserted (Claude 3 Sonnet) due to context length limits.

Figure 25: Effect of adding conflicting versus non-conflicting preferences on adherence. The red bar indicates the performance when only the original preference is present. Results are averaged over five topics using a fixed 100-turn conversation.

Table 18: Human-LLM agreement rates across different error checker as well as the final preference following accuracy. We randomly sampled 100 evaluations from each preference form and calculated the agreement rate between human annotators and the LLM evaluator in their judgments.

| Error Checker | Explicit Preference | Implicit Choice-based | Implicit Persona-driven |
|---|---|---|---|
| Violate Preference? | 0.92 | 0.86 | 0.95 |
| Acknowledge Preference? | 0.88 | 0.90 | 0.97 |
| Hallucinate Preference? | 0.98 | 0.96 | 0.92 |
| Helpful Response? | 0.96 | 0.93 | 0.90 |
| Preference Following Accuracy | 0.97 | 0.92 | 0.96 |

