# OpenReview forum: "Do LLMs Recognize Your Preferences? Evaluating Personalized Preference Following in LLMs"
_ICLR.cc/2025/Conference — ICLR 2025 Oral_

### Official Review · Reviewer_d6T5 · 2024-10-18

**Soundness:** 3
**Presentation:** 3
**Contribution:** 4
**Rating:** 8
**Confidence:** 4

**Summary:**

This paper presents a challenging and important problem of LLMs: memorizing the user preference in a long-context, multi-turn dialogue (could be up to 100k tokens). To achieve this, the authors create a PrefEval dataset containing 3000 examples (i.e., preference-query pairs), distributed across 3 preference forms and 20 different topics related to our daily conversations with chatbots.

Specifically, in their dataset, the stated user preference could be in three different forms (blue box in Figure 1; see also Table 14 to 16 in Appendix):

* one turn (also only one sentence in the user utterance, e.g., I don't like seafood)

* two turns (e.g., a user first asks a question, then after an LLM provides four options, the user chooses one of them and rejects the others)

* $>$ 2 turns (i.e., they insert the user preference within an unrelated 4~8 turn dialogue; for instance, a user "accidentally let slip" and mentions he/she doesn't like seafood in a library-related dialogue)

To make the conversation turn extremely long, they leverage the LMSYSChat-1M dataset by randomly selecting the dialogue up to 100k tokens, and then they insert this long context between the user preference (typically in the first turn) and the query question (at the end of the dialogue).

The authors benchmark the PrefEval dataset using a variety of LLMs (GPT, Claude, Mistral, Llama-3, and Gemini), as well as the latest GPT-o1 system. Concretely, they conduct different experimental settings (see also Appendix A.4):
1. (pure) zero-shot
2. zero-shot + inject a sequence (like “provide an answer that follows my preference”) after the test question (termed "reminder" in this paper)
3. let LLM evaluate its own response and generate the answer again, similar to the self-correct task or simple CoT baseline (termed "self-critic" in this paper)
4. give few-shot examples to the LLM before the LLM response (termed "$CoT$"* in this paper)
5. use RAG method to retrieve relevant sentences in the previous dialogue then provide these sentences before the LLM response.

Additionally, they finetune the Mistral 7B LLM using their dataset with the reminder method and observe that SFT improves in alignment with user preferences. This simple SFT approach could be more robust in long-term dialogue if the training data has more unrelated turns inserted between the user preference and the query dialogue (which is reasonable).

(*) though this could be a bit confusing, see weakness 1 below

**Strengths:**

The strength is mostly summarized in the Summary.

Overall, this paper is well-motivated by the current capabilities of LLMs to follow the user preference in long-term conversation, which is also an emerging issue of contemporary ("general-purpose") LLMs.
To take a step further, they construct this dataset and gauge whether LLMs could implicitly retrieve the user preference and follow it in the long-context, long-term setting.

While the originality of following user preference may not be really novel $-$ since memorizing the user preference has been studied in other NLP tasks such as task-oriented dialogue (TOD) $-$ there is still a lack of comprehensive experiment in this extremely long-term scenario, not to mention various LLMs are conducted in this paper. Multiple baselines are performed when LLMs are frozen, including the pure zero-shot, zero-shot with a "follow my preference" string, few-shot, self-correction/CoT (termed "self-critic"), and RAG.
Different question (i.e., query) types are also tested ("generation" task and (MCQ) "classification" task).
This is perhaps one of the most comprehensive evaluations I've read in recent CoT-prompting papers.

The authors also plan to open-source the dataset and release the prompts used in the Appendix.

**Weaknesses:**

The major weaknesses are that (1) some terms may confuse the readers at first glance and (2) the dataset construction is not very clear in this paper (as it is in the datasets and benchmarks track) $-$ even though the benchmark experiments are exceedingly extensive.

## Weakness 1

(i) In this paper, the "$CoT$"* method is defined as having several (i.e., 5) examples before an LLM response (visualized in Appendix A.4, page 17). However, providing examples to the LLM is generally considered to be few-shot learning (or, previously in-context or demonstration learning). In particular, the term CoT should be more related to the "self-critic" method in this paper (which can also be viewed as a simple self-correct task [1]).

(ii) The evaluation metric of the "generation task" is the accuracy (see page 2, Figure 1's caption) in this paper. If I understand correctly, the generation task in NLP typically refers to translation, summarization, story completion, etc. (hence, accuracy is generally *not* the primary evaluation metric, which could be BLEU/ROUGE/METEOR/BERTscore/etc.). While the authors state that "the LLM generates a response of approximately 300 words in reply to the user's query." on page 4 and provide a real-life scenario (asking for a travel recommendation in New Orleans and see if an LLM knows the user doesn't like jazz) in the Introduction, the evaluation is still mainly the accuracy by leveraging LLM-as-a-judge, which could be done $-$ and is also simpler and cheaper than LLM-as-a-judge $-$ using the answer extraction in [2]. To be more specific, we can prompt the LLM by asking "Does the response contain any [seafood] cuisine?" in Figure 1, provided that a user stated that he/she doesn't like [seafood], then extract the Yes/No response afterward.).

Moreover, perhaps using LLM-as-a-judge is costly and complex, the authors also conducted multiple-choice question (MCQ) experiments and included the relation between the classification task and generation task in Appendix A.5, which demonstrates that there is a high correlation in accuracy. In this vein, while LLM-as-a-judge also reports four error types introduced in this work (see Table 1 and Figure 7), there isn't a significant difference between the "generation" task and the "classification" task in this paper to my knowledge (though those four error types should be useful for error analysis in this sense). As a result, after the readers look at Figures 5 and 6, they may not immediately understand why the authors evaluate two "different" tasks (though I believe they *should be* the same, see explanation below).

Take Figure 1 as an example. The user first asks "What Japanese cuisine dishes should I try on my visit?", then this query (from my understanding) is for the generation task (because LLMs tend to reply with lengthy sentences for general (i.e., non-specific) questions), which is evaluated by LLM-as-a-judge (Claude 3 Sonnet). On the other hand, if, for example, the query becomes "What Japanese cuisine dishes should I try on my visit? (A) beef and mushroom noodles (B) vegetable stir fry (C) seafood with shrimp and crab (D) grilled chicken Caesar salad", then perhaps this is for the classification task? (I cannot find the query related to MCQ so I give an illustrative example above.)

## Weakness 2

(i) Though the three forms of user preference (blue box) and the multi-session conversations (green box) are clearly stated in Figure 1, it is unknown what the **source** of the user preference is, as well as the construction between three different preference forms. After reading the paper, I only found "These pairs were manually curated with the assistance of GPT-4, Claude 3 Sonnet, and Claude 3.5 Sonnet." in Section 2.2.  In Tables 14 to 16, the authors only explain "Every implicit preference dialogue is derived from an explicit preference" and "Every dialogue is derived from an explicit preference and is randomly assigned a persona to simulate a longer conversation."

Specifically, how to first generate the explicit user preference and get the distribution in Figure 2? Are the preferences mined from existing dataset(s) or generated by LLMs? After this, how is the corresponding query generated? Next, what are the criteria for generating the implicit choice-based/persona-driven dialogue? I found in Table 15 that the templates are rather fixed. For example, the chatbot only provides four choices in the first turn, and the user as well as the chatbot response are highly similar across topics. Are there any rules for prompting LLMs or are they human-generated templates (with some text combinations)?

(ii) Another issue might be the problem formulation in Section 2.1, as it is not formalized in math. Although I have no trouble understanding the PrefEval dataset and the proposed framework/experiments in general, other not-too-familiar readers may need to re-read several times to grasp the idea behind this paper. For example, the formulation of the conversation flow in Figure 1. Here, we only know user preference $p$ is in the blue box, and the query $q$ is in the red box. How to generalize it in a $m$-turn conversation (i.e., {$u_1, b_1, u_2, b_2, ..., u_m, b_m$})? Alternatively, as each box is separated, can the PrefEval dataset construction be formalized in a mathematical way?


[1] Large Language Models Cannot Self-Correct Reasoning Yet, Jie Huang, Xinyun Chen, Swaroop Mishra, Huaixiu Steven Zheng, Adams Wei Yu, Xinying Song, Denny Zhou, ICLR 2024

[2] Large Language Models are Zero-Shot Reasoners, Takeshi Kojima, Shixiang Shane Gu, Machel Reid, Yutaka Matsuo, Yusuke Iwasawa, NIPS 2022

Regarding the format, the line number is missing in its current PDF format, so the presentation is somewhat unclear, making it difficult to address the concerns below (not sure if it is because the version is the previous one).

**Questions:**

Despite the above major weaknesses, **these two weaknesses could be easily fixed** to improve this paper's clarity as follows (first migrate this paper to the 2025 template):

### Suggestion for Weakness 1-(i)

Perhaps (a) modify the "$CoT$"* to "few-shot", and (b) either include some (intrinsic) self-correction papers that the authors found related to your "self-critic" or simply state self-critic is also the CoT baseline (with CoT citations).

### Suggestion for Weakness 1-(ii)

I personally would prefer to report accuracy in MCQ experiments, whereas LLM-as-a-judge is for error-type analysis only. However, If replacing the generation task performance with the classification task requires substantial revisions, then it's best to explicitly state something like "the accuracy in the generation task **is highly correlated** to the classification task" either in Figure 1 (on page 2) or in Section 2.5 (on page 4).
Either way should be better than first mentioning it on page 7 (in Section 3.4): "This correlation suggests that despite task differences, ... Classification tasks could thus serve as an efficient proxy for evaluating preference following in complex generation scenarios across models and methods." Moreover, this may not be generalized to other complex generation tasks such as machine translation.

### Suggestion for Weakness 2

Detail the dataset construction in the Appendix, especially the process of how the user preference, its related query, as well as the construction of implicit preference form (Tables 15 and 16). Is the PrefEval dataset constructed in a more standardized way that could be formalized for future research to follow this criterion? If not, then it is also appreciated to include the flowchart or the pipeline process of how the explicit and two implicit forms are generated, perhaps with a walkthrough example, in the Appendix.


I could be convinced if the authors can address the following questions below. No need to conduct additional experiments, just clarify the rationale behind this paper (e.g., the figure shown, experimental setup).


# Questions

1. See Weakness 1. If I am very much mistaken, perhaps the authors could provide some papers that use this term in this scenario?

2. See Weakness 2.

3. In Figure 1
* Do authors conduct experiments where the user preference (blue box) is in between the multi-session conversations (green box)? Specifically, as the multi-session conversations (green box) are from LMSYS-Chat-1M dataset in Section 2.4, I wonder if the top one is also filled in the experiments (just curious if I am missing something in this paper).
* In explicit preference form, as the conversation turn is defined as a pair of user and chatbot utterances in Section 2.1, what is the corresponding "chatbot" utterance in user preference when a user states "I dislike eating seafood due to an allergy" in the first turn (see Figure 1)? Is it an empty string or some pre-defined sentence (because I cannot find it in Table 14)? If so, is it used across all explicit data?
* In the implicit choice-based preference form, does the first user question the same as the query? Specifically, as shown in Table 15, suppose a user asks "What are some good hotel options for my upcoming trip to Paris?" in the first turn, then does the user ask the same question in the last turn (which is the query)?
* When evaluating results, does the green box (multi-session conversations) also present to the evaluator? If so, then the figure is missing, which may cause some confusion as the evaluator would become impractical in real-world scenarios (as we normally don't have labeled data). If not, then could the authors explain why it's excluded in reporting accuracy (I think it is reasonable to exclude this only for error analysis, as shown in Appendix A.11)?

4. In the Introduction, does "prompting" in the second contribution mean "zero-shot, reminder, and $CoT$*" in Section 3.1? Similarly, does "iterative feedback" mean "self-critic"? As for RAG, is the retrieved context "utterance-based" or "sentence-based" (e.g., if a user utterance has 3 sentences, only one of which is related to the query)?

5. In Section 2.5, does LLM-as-a-judge similar to the work in [1]? Are the four error types used in the previous studies, or did the authors pre-define them? In Table 1, why the value of (Preference Hallucination Violation, Hallucinate Preference?) is "No"?

6. In Section 3.1, as few-shot examples are given in the "$CoT$"* method, did authors previously also experiment with placing those examples on top (as many papers first provide LLMs with examples and then ask them to perform some tasks)?

7. In Table 2, Is the number of data conducted in this restaurant 3000 x 5.6% (168) or 1000 x 5.6% (56) in 10 Turns and 300 Turns, respectively? I am also curious how this travel-restaurant topic is reported but not the others. Is it because of the cost, or does GPT-o1 perform better in this topic?

8. In Section 3.3, why does Claude struggle more with implicit persona-driven dialogue compared to the choice-based one? In Figure 5, I found that Claude 3 Sonnet is generally on par with or better than in the reminder scenario (it is worse if the token length is around 16k~23k, but for longer context, there isn't much difference).

9. In Section 3.4, what is the "structure task"?

10. In Section 3.6

* Could the authors provide an example in "Impact of Multiple Preferences Stated in Conversation"? Does it mean that if a user says "I don't like seafood" (restaurant) in the first turn, then he/she also states "I don't like wearing jeans" (fashion), "I like to read" (music & books) in the following turns?

* Similarly, could the authors provide an example in "Effect of Conflicting Preferences Stated in Conversation"?

* What is the number of data tested in Figures 8 and 9, respectively?

11. In my opinion, even though the $(p, q)$ pair is indeed different in this sense, I would expect 3,000 to be the "totally" different dataset (if only glancing at the Abstract). Maybe it is better to rephrase something like 1,000 data, each with three "levels of difficulty" (user preference forms).

12. Is the temperature set to 0 for all LLMs?

13. Do the authors include the Ethics Statement?

[1] Judging LLM-as-a-Judge with MT-Bench and Chatbot Arena, Lianmin Zheng, Wei-Lin Chiang, Ying Sheng, Siyuan Zhuang, Zhanghao Wu, Yonghao Zhuang, Zi Lin, Zhuohan Li, Dacheng Li, Eric P. Xing, Hao Zhang, Joseph E. Gonzalez, Ion Stoica, NIPS 2023

# Typos and Format Suggestions

1. In Figure 1 "shnmp" $\rightarrow$ "shrimp"
2. The em dash ($-$) in Section 3.7 should be consistent with the previous ones.
3. In Figure 10, "W/" $\rightarrow$ "w/"
4. Could try to fit GPT-4o in Section 3.1 since there are only nine models mentioned in this section.
5. Should use wrapfigure in Figures 3, 7, and 10 so that the caption is below the figure.
6. From my understanding, when citing the papers in Appendix A.2, \citet{} should not be used even if they are in the middle of the sentence (consistently use \citep{}).
7. In Table 14~16, if Table 14 shows 10 examples while Table 15 could only display 6 within a page, perhaps it would be more visually comparable to select the first 6 examples from Table 14 (in the same order).
8. For color consistency in Figure 10, the author could draw the 5 w/o SFT methods first so it will match the color in Figure 4. The authors could also consider drawing two legends in this figure (for example, a legend displays the marker: star denotes w/ SFT and circle denotes w/o SFT).

---

> ### Author Response · Authors · 2024-11-21
> **Rebuttal Reply 1/n (with additional results and paper rewriting)**
>
> Dear reviewer d6T5,
>
> Thank you for your detailed, thorough and insightful feedback\! We are encouraged by you recognizing our paper's strengths regarding (1) "one of the most comprehensive evaluations" across LLMs and baselines, (2) well-motivated problem setting addressing real-world LLM capabilities, and (3) "exceedingly extensive" experimental analysis.
>
> We appreciate your constructive suggestions and have addressed your concerns with clarifications and additional details.
>
> > **W1.1: "Terminology confusion around 'CoT'..providing examples to the LLM is generally considered to be few-shot learning (or, previously in-context or demonstration learning). "**
> We agree that our usage of certain terms could be confusing and have made the following revisions: we have renamed our "CoT" method to "few-shot CoT." In the original CoT paper, CoT prompting is defined as “*augmenting each exemplar in few-shot prompting with a chain of thought for an associated answer*”. Following this definition, we designed our few-shot CoT prompting, as detailed in Appendix A.4, where we provide manually constructed chains of thought (created using Claude 3.5) to guide the LLM in step-by-step reasoning to adhere to user preferences. Recognizing the potential overlap with few-shot prompting, we explicitly clarified the distinction by renaming it "few-shot CoT”.
>
> We've also clarified that our "self-critic" approach is similar to prior work in intrinsic self-correction, where the LLM itself generates critiques to its own previous generation, citing relevant papers \[1\].
>
> \[1\] Large Language Models Cannot Self-Correct Reasoning Yet, Huang et al, ICLR 2024
>
> > **W1.2: “The evaluation metric of the "generation task" is the accuracy in this paper. If I understand correctly, the generation task in NLP typically refers to translation, summarization, story completion, etc. (hence, metric could be BLEU/ROUGE/METEOR/BERTscore/etc.)”**
>
> While traditional NLP generation tasks often employ metrics like ROUGE when clear ground truth responses exist, our benchmark addresses open-ended queries (e.g., restaurant recommendations) where curating ground truth responses is infeasible due to the multiplicity of valid outputs. Using accuracy as our evaluation metric enables clear comparisons of model performance across turn numbers, models, and baselines, which is particularly valuable for understanding preference adherence in open-ended scenarios.
>
> > **W1.2 cont’d: “LLM-as-a-judge could be done − and is also simpler and cheaper than LLM-as-a-judge − using the answer extraction in \[2\]. To be more specific, we can prompt the LLM by asking "Does the response contain any \[seafood\] cuisine?" in Figure 1, provided that a user stated that he/she doesn't like \[seafood\], then extract the Yes/No response afterward.).**
>
> Thank you for suggesting answer extraction as an evaluation method. We'd like to clarify our "violation preference?" checker in Table 1 already implements this type of binary evaluation. However, we found that relying solely on this violation checker was insufficient for comprehensive evaluation. Our framework includes three additional binary checkers because models can fail in ways beyond simple preference violations – they may provide unhelpful responses, refuse to answer, hallucinate preferences, or exhibit inconsistent behavior across turns. These failure modes are explicitly captured through our additional error types. Therefore, the preference-following accuracy we report provides a more comprehensive assessment of model performance in generation tasks.
>
> > **W1.2 cont’d: “I personally would prefer to report accuracy in MCQ experiments, whereas LLM-as-a-judge is for error-type analysis only. However, If replacing the generation task performance with the classification task requires substantial revisions, then it's best to explicitly state something like "the accuracy in the generation task is highly correlated to the classification task" either in Figure 1 (on page 2\) or in Section 2.5.**
>
> We agree with the reviewer's suggestion to explicitly state the correlation between generation and classification task performance. We have added this clarification in Figure 1\.
> However, we want to emphasize that there is no need to “replace” the generation task with the classification task in our paper. While both tasks show correlated performance, they serve distinct and complementary purposes. The generation task allows for open-ended question answering where models can provide detailed explanations and handle scenarios not limited by predefined options. The classification task, while more computationally efficient, has the inherent limitation that it cannot exhaustively enumerate all possible answer choices. We evaluated both tasks with the same amount of effort in terms of the same set of models, baselines, preference forms, and turn numbers for each. This allows users to choose between the tasks based on their computational resources.

---

> ### Author Response · Authors · 2024-11-21
> **Rebuttal Reply 2/n (with additional results and paper rewriting)**
>
> > **W2.1: Missing Data construction details: “...how to first generate the explicit user preference and get the distribution in Figure 2? Are the preferences mined from existing dataset(s) or generated by LLMs? After this, how is the corresponding query generated? Next, what are the criteria for generating the implicit choice-based/persona-driven dialogue? …”**
>
> Thank you for this important question. We have added a new section (Appendix A.13) that comprehensively describes our three-component data generation methodology. Here we give a brief summary: we first generated a set of explicit preferences, we started with 20 diverse topics, generated \~10,000 initial preference-question pairs using Claude 3 Sonnet, and implemented rigorous multi-stage filtering using both human labelers and LLM evaluators to select 1,000 high-quality pairs. For implicit preferences, we created two variants: choice-based preferences expressed through multiple-choice conversations, and persona-driven preferences embedded within 5-8 turn conversations using 100 diverse personas. Our quality control measures included validity assessment by human labelers, automated violation rate analysis using 5 different LLMs, and a systematic difficulty rating system using human-labeled examples as in-context examples. During the above construction and filtering process, we have spent large efforts in designing the prompts at each stage.
>
> We will release all data generation prompts in our code repository for reproducibility.
>
> > **W2.2: “problem formulation in Section 2.1, as it is not formalized in math. \[…\] can the PrefEval dataset construction be formalized in a mathematical way?”**
>
> We agree that additional mathematical formulation would improve the clarity of our problem setup. We have revised Section 2.1 to include formal mathematical notation that defines the relationships between the key components of our multi-session framework: queries, preferences, turns, sessions, and conversations.
>
> > **Q1: “In Figure 1, do authors conduct experiments where the user preference (blue box) is in between the multi-session conversations (green box)?”**
>
> For most experiments, we place the user preference at the beginning of the conversation and the query at the end. We also conduct experiments where preferences are inserted in the middle of the conversation to evaluate dynamic preference following, as discussed in Section 3.6. And we explored how the position of preference within the conversation affects the final performance in Appendix A.15.
>
> > **Q2: “In explicit preference form, what is the corresponding "chatbot" utterance when a user states "I dislike eating seafood due to an allergy" in the first turn? Is it an empty string or a predefined sentence?”**
>
> The corresponding chatbot utterance is an actual response generated by each LLM, which typically acknowledges the user's preference. We will include additional examples in the appendix for clarity.
>
> > **Q3: “In the implicit choice-based preference form, is the first user question the same as the query?”**
>
> No, the first user question differs from the final query. Repeating the same question would revealing the ground truth to the query so we used different query.
>
> > **Q4: “When evaluating results, is the multi-session conversation (green box) also presented to the evaluator? If not, why is it excluded in accuracy reporting?”**
>
> The multi-session context is not presented to the evaluator. This is a deliberate design choice because our focus is specifically on assessing whether responses correctly follow and implement the user's stated preferences in a helpful manner. Including the multi-session context would not only add unnecessary computational overhead, especially in long-context settings, but would also be irrelevant to the core evaluation objective of preference-following accuracy.
>
> > **Q5: “In the Introduction, does "prompting" mean "zero-shot, reminder, and CoT" in Section 3.1? Does "iterative feedback" mean "self-critic"? For RAG, is the retrieved context "utterance-based" or "sentence-based"?\*”**
>
> Yes, "prompting" includes zero-shot, reminder, and few-shot CoT methods. "Iterative feedback" corresponds to self-critique. The retrieval context in RAG is exchange-based, with each turn containing two exchanges.
>
> > **Q6: “In Section 2.5, is LLM-as-a-judge similar to prior work, and were the error types pre-defined or adapted from existing studies? In Table 1, why the value of (Preference Hallucination Violation, Hallucinate Preference?) is "No"? ”**
>
> Our LLM-as-a-judge framework is similar to prior work, but the four error types were newly defined in our study to capture critical failure modes. For Table 1, the "No" value for "Preference Hallucination Violation, Hallucinate Preference?" is a typo, we have corrected this, thanks for catching\!

---

> ### Author Response · Authors · 2024-11-21
> **Rebuttal Reply 3/n (with additional results and paper rewriting)**
>
> > **Q7: “In Section 3.1, did authors experiment with placing few-shot examples on top instead of CoT?”**
>
> Yes, we experimented with placing reminder and CoT prompts in the system prompt (on top). However, performance was significantly worse in this configuration.
>
> > **Q8: “In Table 2, is the number of data points 3000 x 5.6% (168) or 1000 x 5.6% (56)? Why is the travel-restaurant topic reported but not others?”**
>
> In Table 2, we only report results for explicit preferences, therefore it is 1000 x 5.6% (56). We focused on the travel-restaurant topic due its relevance to real-world dietary restrictions. The topic was not cherry-picked based on results. As conducting experiments on more topics under the 300 turn (\~100k tokens) settings is costly due to our computation budget.
>
> > **Q9: In Section 3.3, why does Claude struggle more with persona-driven dialogue compared to choice-based dialogue?**
>
> That’s a good observation. We hypothesize that Claude struggles more with preferences expressed through persona-driven dialogue because it gets distracted by the persona-driven conversation and ignores briefly mentioned preferences, compared to other models.  In contrast, choice-based dialogues are less complicated for Claude to process, as they involve just two turns of conversation with a clear preference statement and it outperforms other models.
>
> > **Q10: In Section 3.4, what is the "structure task"?**
>
> "Structured task" refers to the MCQ task, which is more constrained than the generation task, as it requires selecting one answer (e.g., a, b, c, or d).
>
> > **Q11: In Section 3.6, Could the authors provide an example in "Impact of Multiple Preferences Stated in Conversation"? Does it mean that if a user says "I don't like seafood" (restaurant) in the first turn, then he/she also states "I don't like wearing jeans" (fashion), "I like to read" (music & books) in the following turns?**
>
> Yes, your listed example is correct. In the multiple preference experiment, we randomly sample preference from each topic, since we have 20 topics, we can maximally sample 20 preferences to insert within a conversation. We only sample one preference from each topic in this experiment to avoid conflicting preferences that are from each topic.
>
> > **Similarly, could the authors provide an example in "Effect of Conflicting Preferences Stated in Conversation"? What is the number of data tested in Figures 8 and 9, respectively?**
>
> Yes, an actual example of a conflicting preference we used is: “*I dislike any form of transportation that involves being on water due to motion sickness*” versus “*I love taking cruises and sailing trips because being on the water is so relaxing.*” Note that one preference is from our dataset, while the other was generated by Claude 3.5 Sonnet to intentionally conflict with the first preference.
>
> In Figure 8, each data point represents an average of 100 conversations. In Figure 9, the results are averaged over 5 topics, comprising approximately 250 data points. Additionally, since Figures 8 and 9 focus exclusively on Claude models, we have conducted further experiments to validate our findings on dynamic preferences. These new experiments include evaluations with Mistral 8x7b and Mistral 7b, and the results are included in Appendix A.16.
>
> > **Q12: Rephrase to emphasize that the dataset has 1,000 unique user preferences with three levels of difficulty.**
>
> Yes, we have clarified this in Section 2.2 and also in our detailed data construction section in Appendix A.13.
>
> > **Q13: Is the temperature set to 0 for all LLMs?**
>
> Yes, we use a temperature setting of 0\.
>
> > **Q14: Did the authors include an Ethics Statement?**
>
> We have added an Ethics Statement in the revised paper.
>
> > **Typos.**
>
> Thanks for catching the typos! We will fix them in the revision.
>
> > **Could try to fit GPT-4o in Section 3.1 since there are only nine models mentioned in this section.**
>
> Yes, we have additionally included GPT-4o's results in Table 4.
>
>
> We greatly appreciate your feedback and believe that these new experiments and clarifications strengthen our paper. We hope this addresses your concerns and please let us know if you have any questions.

---

> ### Comment · Reviewer_d6T5 · 2024-11-26
>
> Thank you for your detailed response. Most of the concerns are addressed.
>
> > For most experiments, we place the user preference at the beginning of the conversation and the query at the end. We also conduct experiments where preferences are inserted in the middle of the conversation to evaluate dynamic preference following, as discussed in Section 3.6. And we explored how the position of preference within the conversation affects the final performance in Appendix A.15.
>
> Are the experiment in Appendix A.15 explicit preference?
>
> I would suggest mentioning the # data shown in each figure and table (especially when the data is not "large" enough, like Table 2) since sometimes it needs further re-reading to understand (or, guess) the actual number.
>
> > The multi-session context is not presented to the evaluator. This is a deliberate design choice because our focus is specifically on assessing whether responses correctly follow and implement the user's stated preferences in a helpful manner. Including the multi-session context would not only add unnecessary computational overhead, especially in long-context settings, but would also be irrelevant to the core evaluation objective of preference-following accuracy.
>
> It would be even nicer, in the final version, if using "small" LLMs (like Mistral 7b or LLaMA 3 8b) to run this experiment and see how they handle this long-context scenario, which is more realistic regarding "lost in the middle" task. (As the discussion period is ending soon, there is no need to report the result this time.)
>
> > "Structured task" refers to the MCQ task, which is more constrained than the generation task, as it requires selecting one answer (e.g., a, b, c, or d).
>
> I would recommend change it to classification (MCQ) task.

---

> > ### Author Response · Authors · 2024-11-26
> > **Thank you!**
> >
> > Dear Reviewer d6T5,
> >
> > Thank you for replying to our response and for raising score!
> >
> > > Are the experiment in Appendix A.15 explicit preference?
> >
> > yes, they were conducted using explicit preferences.
> >
> > > I would suggest mentioning the # data shown in each figure and table
> >
> > We agree, we will mention the exact number of preferences or topics experimented in each caption in the final paper.
> >
> > > It would be even nicer, in the final version, if using "small" LLMs (like Mistral 7b or LLaMA 3 8b) to run this experiment and see how they handle this long-context scenario, which is more realistic regarding "lost in the middle" task.
> >
> > Yes, we will conduct this with smaller models and add to the final version.
> >
> > > I would recommend change it to classification (MCQ) task.
> >
> > True, "Structured task" may be unclear.
> >
> > Thank you again for reviewing and helping us improve our paper! We will incorporate the above suggestions into our final paper.

---

### Official Review · Reviewer_PRWG · 2024-10-30

**Soundness:** 3
**Presentation:** 4
**Contribution:** 3
**Rating:** 8
**Confidence:** 3

**Summary:**

This paper introduces PREFEVAL, a benchmark for evaluating large language models' (LLMs) ability to infer, remember, and follow user preferences in multi-turn conversations. The main contributions are:

(1) Introduction of a comprehensive benchmark assessing LLMs' ability to follow user preferences in conversational settings, comprising 3,000 manually curated question-preference pairs across 20 topics and 3 preference forms.

(2) Through extensive evaluation of 10 state-of-the-art LLMs, the paper reveals current challenges in actively following user preferences during conversations and presents important findings about personalized preference adherence.

**Strengths:**

The paper's strengths are as follows:

(1) **In terms of originality**, it proposes a comprehensive benchmark for evaluating LLMs' personalized preference adherence capabilities, which is a relatively novel research direction.

(2) **Regarding quality and writing**, the authors thoroughly validated LLMs' capabilities through extensive evaluation of multiple models' preference adherence across various conversation topics. The paper structure is clear, with well-presented experimental design and results.

(3) **In terms of significance**, the findings reveal key challenges in current LLMs' personalized preference adherence. In daily use and experiments, we also observe that LLMs' adherence capability diminishes with increased conversation length. Broadly speaking, ensuring LLMs consistently follow user characteristics or instructions is crucial for their real-world applications.

**Weaknesses:**

The paper's weaknesses are as follows:

(1) **The data construction process is not detailed**. A major contribution of this paper is the manually constructed personalized preference dataset. The construction process directly affects data quality and thus impacts model results and experimental reliability. However, I did not find a section detailing the dataset construction process.

(2) **Additional experiments could enhance the comprehensiveness of findings**. The authors verified LLMs' adherence to explicit preferences through fine-tuning. However, implicit preferences are often more challenging to identify and follow. Experiments in this area are recommended for a more comprehensive understanding of LLMs' preference recognition and adherence capabilities.

(3) **Line numbers are missing**. The paper lacks line numbers, making it difficult to precisely reference specific content locations.

**Questions:**

(1) **The detailed dataset construction process is not introduced**. Multiple methods should be employed to verify the dataset's high quality, which would make subsequent experiments more convincing. I recommend adding a dedicated section that outlines the data preparation process and quality control measures. This should include details on selection criteria, query development, and validation procedures used to ensure data quality.

(2) **How were the reasoning steps for CoT obtained?** Since reasoning steps are crucial for CoT effectiveness, the authors did not discuss this detail. I suggest the authors provide examples of CoT prompts, or describe their process for generating these prompts.

(3) **When multiple preferences are injected, why does the adherence to original preferences improve?** Intuitively, when multiple preferences are introduced, LLMs would struggle to accurately identify user preferences. This should make it harder for LLMs to recognize the original preferences. However, the paper's results show the opposite.(Section 3.6, Page 8). I recommend the authors explore potential explanations for this phenomenon, or conduct additional experiments to verify if this effect is consistent across different models.

(4) **When conflicting preferences are introduced, model performance improves compared to single preferences, which is counter-intuitive**. Why does this phenomenon occur, and do other LLMs besides Claude 3.5 Sonnet exhibit similar behavior? (Section 3.6, Page 9). It is recommended that the authors expand their analysis to include more models beyond Claude 3.5 Sonnet. Additionally, they should provide potential explanations for these counter-intuitive results and discuss their implications for understanding LLM behavior in complex preference scenarios.

---

> ### Author Response · Authors · 2024-11-21
> **Rebuttal Reply 1/n (with additional results and paper rewriting)**
>
> Dear reviewer PRWG,
>
> Thank you for your thoughtful feedback and for recognizing our paper's strengths in terms of (1) originality in benchmark creation, (2) thorough evaluation methodology with well-presented experimental design, and (3) significance of our findings for real-world applications. We appreciate your constructive suggestions and have addressed your concerns with additional details and analysis.
>
> > **W1 and Q1: "The data construction process is not detailed. A major contribution of this paper is the manually constructed personalized preference dataset \[...\] However, I did not find a section detailing the dataset construction process." “Regarding the detailed dataset construction process, Multiple methods should be employed to verify the dataset's high quality, which would make subsequent experiments more convincing. I recommend adding a dedicated section that outlines the data preparation process and quality control measures. This should include details on selection criteria, query development, and validation procedures used to ensure data quality.”**
>
> Thank you for this important question. We have added a new section (Appendix A.13) that comprehensively describes our three-component data generation methodology. Here we give a brief summary: for explicit preferences, we started with 20 diverse topics, generated \~10,000 initial preference-question pairs using Claude 3 Sonnet, and implemented rigorous multi-stage filtering using both human labelers and LLM evaluators to select 1,000 high-quality pairs. For implicit preferences, we created two variants: choice-based preferences expressed through multiple-choice conversations, and persona-driven preferences embedded within 5-8 turn conversations using 100 diverse personas. Our quality control measures included validity assessment by human labelers, automated violation rate analysis using 5 different LLMs, and a systematic difficulty rating system using human-labeled examples as in-context examples. During the above construction and filtering process, we have spent large efforts in designing the prompts at each stage. We will release all data generation prompts in our code repository for reproducibility.
>
> > **W2: "Additional experiments fine-tuning for implicit preferences are often more challenging to identify and follow. Experiments in this area are recommended for a more comprehensive understanding of LLMs' preference recognition and adherence capabilities.."**
>
> Thank you for this feedback. We have added investigations on how fine-tuning impacts implicit preference following. As detailed in Appendix A.14, while our training data only included explicit preference examples, the fine-tuned Mistral 7B model showed significant improvements in handling implicit preferences. Across both travel and hotel domains, we observed substantial improvements in implicit preference following after fine-tuning, with accuracy gains of 40-60% for persona-driven preferences and 10-40% for choice-based preferences as shown in the table 1 below. These results suggest that training on explicit preferences enhances not only the model's attention to user preferences but also its general preference inference capabilities.
>
> *Table 1: Preference following accuracy (%) on implicit settings before and after supervised fine-tuning (SFT). We evaluate the model's ability to follow preferences in two implicit scenarios. We show the results across over 100 preferences instances over 2 topics in the zero-shot setting. We find preference finetuning brings more improvements for Implicit Persona-Driven preferences.*
>
> | Topic | Model | Implicit Persona-Driven | Implicit Choice-Based |
> |-------|-------|------------------------|---------------------|
> | Travel Restaurants | Before SFT | 1.79 | 3.57 |
> | | After SFT | **55.36** | **14.29** |
> | Travel Hotels | Before SFT | 14.81 | 11.11 |
> | | After SFT | **74.07** | **51.85** |
>
> > **W3: Line numbers**
>
> We sincerely thank the reviewer for pointing this out and have added line numbers.

---

> > ### Author Response · Authors · 2024-11-21
> > **Rebuttal Reply 2/n (with additional results and paper rewriting)**
> >
> > > **Q2: “How were the reasoning steps for CoT obtained? Since reasoning steps are crucial for CoT effectiveness, the authors did not discuss this detail. I suggest the authors provide examples of CoT prompts, or describe their process for generating these prompts.”**
> >
> > Yes, we have provided the CoT prompts used in Appendix A.4.
> >
> > For example, a single CoT example can be:
> >
> > *User’s preference: ”I’m a vegan looking for new dinner recipes. Any ideas?”*
> >
> > *Good assistant response: ”As a vegan, you’ll need plant-based recipes without animal products. I’ll focus on nutrient-rich, diverse ingredients to ensure balanced meals. Consider: quinoa and black bean burrito bowls, lentil and vegetable curry, roasted vegetable and hummus wrap, or zucchini noodles with avocado pesto. These options offer protein, fiber, and essential nutrients while adhering to vegan principles…”*
> >
> > For generating the chain-of-thought reasoning steps, we use Claude 3.5 Sonnet to generate and refine several few-shot examples (5-shot in our experiments) that demonstrate the reasoning process for preference-following. These examples were carefully designed to show the LLM how to analyze and incorporate the user's preferences before providing a final answer. The prompts serve as explicit demonstrations of how to break down preference-following into clear reasoning steps.
> >
> > > **Q3: “When multiple preferences are injected, why does the adherence to original preferences improve? Intuitively, when multiple preferences are introduced, LLMs would struggle to accurately identify user preferences. This should make it harder for LLMs to recognize the original preferences. However, the paper's results show the opposite.(Section 3.6, Page 8). I recommend the authors explore potential explanations for this phenomenon, or conduct additional experiments to verify if this effect is consistent across different models.”**
> >
> > Yes, we conducted additional experiments to validate our observation as detailed in Appendix A.16. We compared across multiple models, including Mistral 8x7b, Mistral 7b and Claude 3 Sonnet. Our findings in Figure 24 and in the table 2 below confirm that this effect is consistent across different models, with a clear positive correlation between the number of preferences and preference-following accuracy.
> >
> > *Table 2: Preference following accuracy generally improves with more stated preferences in the conversation.*
> > | Model | 2 Prefs | 5 Prefs | 10 Prefs | 15 Prefs | 20 Prefs |
> > |-----------------------|---------|---------|----------|----------|----------|
> > | Mistral 8x7B (50 Turn)| 0.67 | 0.83 | 0.80 | 0.67 |**0.93** |
> > | Mistral 7B (50 Turn) | 0.57 | 0.50 | 0.67 | 0.68 | **0.70** |
> > | Claude 3 Sonnet (80 Turn)| 0.32 | 0.38 | 0.42 | 0.46 | **0.47** |
> >
> > We hypothesize that this improvement arises because introducing multiple preferences throughout the conversation prompts the LLM to allocate proportionally more attention to user preferences than to the unrelated contextual information. This increased focus enables the model to better adhere to user preferences throughout the conversation. And another key reason is the multiple preferences we inserted are from different topics, therefore does not conflict with each other, creating less ambiguity for the LLM to memorize these preferences.

---

> ### Author Response · Authors · 2024-11-21
> **Rebuttal Reply 3/n (with additional results and paper rewriting)**
>
> > **Q4: “When conflicting preferences are introduced, model performance improves compared to single preferences, which is counter-intuitive. Why does this phenomenon occur, and do other LLMs besides Claude 3.5 Sonnet exhibit similar behavior? (Section 3.6, Page 9). It is recommended that the authors expand their analysis to include more models beyond Claude 3.5 Sonnet. Additionally, they should provide potential explanations for these counter-intuitive results and discuss their implications for understanding LLM behavior in complex preference scenarios.”**
>
> To explore this further, we conducted additional experiments extending the analysis to Mistral 8x7b and Mistral 7b, in addition to Claude 3 Sonnet and Claude 3 Haiku, as presented in Appendix A.16 and also in the Table 3 below. Overall, three models (Claude 3 Sonnet, Claude 3 Haiku, and Mistral 7b) demonstrate improved performance with conflicting preference pairs. However, Mistral 8x7b exhibits slightly lower performance with conflicting pairs. This suggests the effect is model-dependent and our findings still holds that conflicting preferences do not necessarily harm performance.
>
> *Table 3: Effect of adding conflicting versus non-conflicting preferences on adherence. Results are averaged over five topics using a fixed 100-turn conversation for Claudes and 70-turn for Mistrals.*
>
> | Model        | Single Preference | Conflict Pair | Non-Conflict Pair |
> |--------------|-------------------|---------------|--------------------|
> | Mistral 7B   | 0.423             | **0.550**     | 0.458             |
> | Mistral 8x7B | 0.657             | 0.708         | **0.722**         |
> | Claude 3 Haiku   | 0.196             | **0.405**     | 0.313             |
> | Claude 3 Sonnet    | 0.202             | **0.438**     | 0.312             |
>
> We hypothesize the increased performance with conflicting pair is due to a "topic-reinforcement effect." Even though the preferences conflict, they focus on the same topic domain, which potentially strengthens the LLM’s memory of the preference context. For instance, if a user states, “I prefer detailed responses for paper summarization” and later specifies, “I prefer concise responses for paper summarization,” the conflicting preferences both emphasize response length as a critical dimension. This repeated focus reinforces the LLM’s attention to that specific aspect, leading to improved accuracy in preference following despite the apparent contradictions.
>
>
>
> We greatly appreciate your feedback and believe that these new experiments and clarifications strengthen our paper. We hope this addresses your concerns and please let us know if you have any questions.

---

> > ### Comment · Reviewer_PRWG · 2024-11-26
> > **Response to Authors**
> >
> > Thank you for your response. You have resolved my concerns.

---

> > > ### Author Response · Authors · 2024-11-26
> > > **Thank you!**
> > >
> > > Dear Reviewer PRWG,
> > > Thank you for your thoughtful reviews and increased score!

---

### Official Review · Reviewer_L7qv · 2024-11-04

**Soundness:** 3
**Presentation:** 3
**Contribution:** 3
**Rating:** 8
**Confidence:** 3

**Summary:**

The authors introduce PREFEVAL, a benchmark for evaluating if LLMs follow personal preferences in conversational settings. PREFEVAL assesses how well LLMs can infer, remember, and adhere to personalized preferences across multi-session conversations. It includes 3,000 preference-query pairs across 20 topics, with preferences presented explicitly or implicitly. Ten LLMs were evaluated using prompting, iterative feedback, and RAG. LLMs struggle to maintain preference adherence over longer dialogues, especially without _reminders_, with accuracy dropping below 10% for 10-turn interactions. Finally, advanced prompting and fine-tuning improved model performance, particularly in handling long contexts and multiple or conflicting preferences.

**Strengths:**

I think this paper covers a timely and interesting topic. The paper and dataset yields several strengths:

1. The error analysis is quite interesting! The authors look at violations of preference applications that are particularly aggravating when interacting with instruction following LLMs

2. The interventions are well-motivated, simple, and moderately effective (though there is still space for improvement).

3. The authors validate their prompts with human annotators.

4. The paper is well-written and generally easy to follow.

**Weaknesses:**

I have three concerns. If the first two are resolved, this would improve the paper!

First, I'd like to see some analysis of the "Lost in the Middle" effect (see: https://arxiv.org/abs/2307.03172). I suspect that some of the failures can be attributed to how models use long context. If we include the reminder at the start instead of the end, or in both places---does that affect performance? I'd also encourage the authors to draw distinctions between needle-in-a-haystack benchmarks for long-context LLMs and this dataset. One reason might be that the implicit / persona settings in the dataset require one to _reason_ over preferences, not just retrieve a needle---but I would make this comparison more explicit in the main text.

Second, the long-context integration feels a bit heavy-handed. The authors seem to randomly sample conversations and randomly intersperse (p, q) pairs within this context. This seems somewhat artificial. Were there any measures taken to make sure that the bridge between the pre-existing context and the (p, q) pairs were natural?

Finally (and I don't expect or require the authors to directly fix this), the (p, q) pairs are entirely synthetically generated. This is an unfortunate limitation: a personalization dataset generated with no "real" people. One could argue that generations from LLMs "simulate" some aspects of human behavior. Still, I'd recommend mentioning this in the limitations section.

**Questions:**

A particularly irritating effect of LLM "personalization" / "memory" is that it assumes that some attribute (e.g. I like pizza) should now apply to all my queries, even if not relevant (e.g. high recall, low precision). Am I correct in assuming this is a "preference hallucination" violation?

> To validate our LLM-based evaluation method, we manually checked 200 randomly sampled evaluations, with an observed 5% error rate.
How was this error rate computed? What was the distribution of errors in the 200 randomly sampled evals?

How were the implicit personas generated and validated? How much variance is there between the generated personas?

With the additional space in the camera ready, I would move some examples of (p, q, explanation) pairs to the main text.

---

> ### Author Response · Authors · 2024-11-21
> **Rebuttal Reply 1/n (with additional results and paper rewriting)**
>
> Dear reviewer L7qv,
>
> Thank you for your thoughtful feedback and for recognizing the strengths of our work in terms of (1) timely and interesting topic coverage, (2) insightful error analysis, (3) well-motivated interventions, and (4) clear presentation.
>
> We appreciate your constructive suggestions and have addressed your concerns with additional experiments and analysis.
>
> > **W1: "I'd like to see some analysis of the 'Lost in the Middle' effect...and needle-in-a-haystack comparison.”**
>
> * To address this suggestion, we conducted experiments analyzing the impact of preference positioning on preference-following performance, which we believe the position of the preference will affect the retrieval of it. As detailed in Appendix 15 and Figure 23, we evaluated Claude 3 Sonnet and Claude 3 Haiku across four diverse topics by inserting the user preference at different points in a fixed 100-turn conversation. The results show a significant decline in preference-following accuracy when preferences are introduced in the middle of the conversation (around turn 50), compared to preferences placed at the beginning or end. These findings corroborate previous research on the challenges LLMs face with mid-context retrieval.
> * Additionally, we revised the main text in Section 4 to explicitly compare our dataset to "needle-in-a-haystack" benchmarks. While those benchmarks primarily test retrieval capabilities from long contexts, PrefEval requires reasoning over preferences, particularly in implicit and persona-driven settings. This distinction underscores the unique challenges posed by our benchmark.
>
> > **W2: “Second, the long-context integration feels a bit heavy-handed…Were there any measures taken to make sure that the bridge between the pre-existing context and the (p, q) pairs were natural?”**
>
> * We understand and agree with the concern about the unnaturalness of inserting preferences into the LMSys dataset when viewed as a single session multi-turn conversation. However , we would like to clarify our setting is multi-session, multi-turn conversations. We believe this also represents the  real-world chatbot interactions from the perspective of users shift between different topics across multiple sessions..
> * In reality, it is hard for users to maintain a single-topic conversation over long context like 100K tokens. Therefore, to assess long context preference following, we designed our setup as a multi-session conversation. Each conversation is divided into distinct sessions, with each session focusing on a specific topic across multiple turns (please refer to our formalized problem setup in Sec 2.1 of our revised paper). For instance, a conversation might begin with 10 turns discussing a debugging challenge, shift to 2 turns expressing dining preferences while seeking restaurant recommendations, and later move to travel planning that incorporates the preference.
> * During preference insertion, we ensure topic transitions occur only after completing a turn. For example, each preference is inserted as a complete session, where the session contains the same number of exchanges from the user and LLM. This makes preference revelation a natural part of the multi-session structure. By modeling dynamic, multi-session interactions, we capture how users naturally engage in diverse, topic-spanning conversations. This setup enables testing of the LLM's ability to maintain and apply user preferences across varied contexts.
>
> > **W3: “the (p, q) pairs are entirely synthetically generated. This is an unfortunate limitation: a personalization dataset generated with no "real" people … I'd recommend mentioning this in the limitations section.”**
>
> We acknowledge that synthetic data cannot fully capture the nuances of preferences expressed by real users, and we have incorporated this point into the limitations section of the paper.
>
> That said, while the (p, q) pairs in our dataset are synthetically generated, they underwent a rigorous selection process from an initial pool of approximately 10,000 generated preferences. This extensive filtering combined manual human evaluation with LLM-assisted ratings to ensure that the preferences and queries are as natural and realistic as possible. As detailed in Appendix A.13, we employed a comprehensive approach, including manual human filtering and leveraging human-provided in-context examples to rate each preference-query pair. This thorough curation process aims to mitigate the limitations of synthetic data.

---

> ### Author Response · Authors · 2024-11-21
> **Rebuttal Reply 2/n (with additional results and paper rewriting)**
>
> > **Q1: “A particularly irritating effect of LLM "personalization" / "memory" is that it assumes that some attribute (e.g. I like pizza) should now apply to all my queries, even if not relevant (e.g. high recall, low precision). Am I correct in assuming this is a "preference hallucination" violation?”**
>
> We agree that this behavior is undesirable. The scenario you described is more like "task hallucination", where irrelevant preferences are unnecessarily applied. In our framework, "preference hallucination" primarily refers to cases where the LLM, when prompted to follow a preference (e.g., using a reminder), incorrectly invents a completely non-existent preference. This “task hallucination” does not exist in our setup since we designed our tasks (e.g., the queries) to be relevant to the stated preference.
>
> While the hallucination you mentioned falls outside the scope of our work, we recognize its importance in the broader context of personalization and consider it a valuable direction for future works.
>
> > **Q2: “To validate our LLM-based evaluation method, we manually checked 200 randomly sampled evaluations, with an observed 5% error rate. How was this error rate computed? What was the distribution of errors in the 200 randomly sampled evals?”**
>
> The 200 samples were randomly selected, and the error rate was computed by examining preference-following accuracy alone. We acknowledge that this method is not fully comprehensive. In response to this concern, we significantly expanded our manual validation efforts by conducting a more thorough evaluation of our LLM-based evaluator. Specifically, we examined human-LLM agreement rates across four error-type binary checkers and three preference forms, as detailed in Appendix A.17. By sampling responses evenly from six models, five baselines, and all benchmarked turns, we manually validated 100 samples per preference form and calculated human-LLM agreement rates. For each response, we applied four error checkers, resulting in a total of 1,200 error checks. The results, presented in the table below, demonstrate strong alignment between human and LLM judgments, with particularly high agreement rates in detecting helpful responses and hallucinations.
>
> *Table: Human and LLM Judge agreement rates across different error checker as well as the final preference
> following accuracy.*
> | Error Checker | Explicit Preference | Implicit Choice-based | Implicit Persona-driven |
> |--------------|--------------------:|---------------------:|----------------------:|
> | Violate Preference? | 0.92 | 0.86 | 0.95 |
> | Acknowledge Preference? | 0.88 | 0.90 | 0.97 |
> | Hallucinate Preference? | 0.98 | 0.96 | 0.92 |
> | Helpful Response? | 0.96 | 0.93 | 0.90 |
> | Overall Preference Following Accuracy | 0.97 | 0.92 | 0.96 |
>
> > **Q3: “How were the implicit personas generated and validated? How much variance is there between the generated personas?”**
>
> We generated the personas using Claude 3.5 Sonnet, initially prompting it to create approximately 200 personas. These were then manually filtered down to 100 to ensure maximum diversity. During the generation process, we carefully ensured that no persona contradicted or explicitly revealed any preferences. We plan to release the prompts used to generate the personas used in our dataset for transparency. We have added detailed steps regarding persona generation in Appendix A.13, Step 3\.
>
> > **Q4: “With the additional space in the camera ready, I would move some examples of (p, q, explanation) pairs to the main text.”**
>
> Yes, we will add more demonstrating examples in the camera ready.
>
>
> We greatly appreciate your feedback and believe that these new experiments and clarifications strengthen our paper. We hope this addresses your concerns and please let us know if you have any questions.

---

> ### Comment · Reviewer_L7qv · 2024-11-25
> **Thank you!**
>
> Great work on the needle-in-the-haystack experiments! I looked over the other authors' concerns, and I'm happy with how you've addressed them, too.
>
> Is there an anonymous dataset link I could look at so I can look at some examples myself? I think that would help! I'm gonna bump my score up since I'd like to see this paper accepted, but I'd still like to see the anonymous link regardless.
>
> Finally, I'd recommend explicitly adding W2 in the final paper as a limitation---so despite your efforts, you might end up with awkward instantiations. But that's fine, since realistic conversations with LLMs also have a ton of topic switches.

---

> > ### Author Response · Authors · 2024-11-26
> > **Thank you!**
> >
> > Dear Reviewer L7qv,
> >
> > Thank you for replying to our response and for raising score!
> >
> > > Is there an anonymous dataset link I could look at so I can look at some examples myself?
> >
> > Yes, we have made samples from our dataset available through an anonymous GitHub repository for your reference: https://github.com/iclr2025anonymous-rebuttal
> >
> > > Finally, I'd recommend explicitly adding W2 in the final paper as a limitation
> >
> > We fully agree this is a limitation of our work, and will add this into the limitation in final revision.
> >
> > Thank you again for your valuable feedback and for helping us improve the quality of our paper!

---

### Official Review · Reviewer_Uqsa · 2024-11-07

**Soundness:** 3
**Presentation:** 3
**Contribution:** 3
**Rating:** 6
**Confidence:** 4

**Summary:**

This Datasets and Benchmarks paper presents PREFEVAL, a benchmark designed to evaluate large language models’ capabilities in following lifestyle preferences in long-context, multi-turn, personalized conversational settings. PREFEVAL includes 3,000 manually curated preference-query pairs spanning 20 diverse topics, assessing models’ ability to remember, infer, and follow explicit and implicit preferences over long conversational contexts (up to 100,000 tokens) and tests various settings such as zero-shot, reminding, CoT, RAG and Self-critic approaches. Using generation and classification tasks, the authors evaluate 10 LLMs, including prominent models like Claude, GPT-4, and LLaMA. The results indicate significant challenges for LLMs in following user preferences over extended dialogues, with a steep decline in preference-following accuracy as context length increases.

**Strengths:**

- 3,000 manually curated (with the help of LLM tools) preference-query pairs across 20 diverse topics is a good dataset contribution, covering various preference forms (explicit and implicit) and contexts, making it a valuable tool for future research on personalized conversational AI.
- The paper presents valuable insights across major open and proprietary models along with an understanding of various strategies that work to help improve preference following.
- The error categorization (preference-unaware violation, preference hallucination, inconsistency violation, and unhelpful response) provides actionable insights into the types of failures occurring during preference following. These insights allow targeted intervention methods to improve the performance of the models and reduce hallucinations and increase helpfulness.
- Personalization is a very important topic for LLMs that's still underrepresented and scope for major advancements. Need for more benchmarks in this field will help with faster progress to create more personalized LLMs.

**Weaknesses:**

- The LLM-based evaluation framework introduces a subjective component and lacks qualitative and manual validations for in assessing preference adherence (despite a very small manual checks on the responses with 5% error rate)
- inserting preferences within the LMSys-Chat-1M dataset may be unnatural. It is not clear what strategies were used for maintain semantic coherence of the dataset while introducing preferences and queries.
- There is some overlap with long context recall and reasoning based benchmarking. There is no analysis of existing datasets and benchmarks in the area that may qualify for personalization based benchmarking. Also, there is no discussion around other efforts like 'LaMP: When Large Language Models Meet Personalization' or others (authors mention these in Appendix, however an exhaustive personalization benchmark could cover some aspects of these other benchmarks mentioned in Appendix).
- This study emphasizes improvements with inserting 'Reminders' and other strategies like self-critique. This might skew model development more towards prompt engineering instead of focusing on understanding intrinsic improvements methods in memory and retrieval capabilities.
-

**Questions:**

- Authors mention breakdown of queries by explicit vs implicit preferences (choice based, persona driven) and share results in Figure 5. Shedding more light on some visualization of attention (and the lack of it) could have shed more light on understanding why the results degrade so much in different scenarios.
- Authors use LMSYS- Chat-1M dataset for the experiments (where they intersperse persona/preference based queries) - some of the models they test may have seen this dataset before during training - what would be the impact of such exposure?
- Since this is a dataset contribution, some more clarity on understanding dynamic user preferences might have been useful. For example, mining instances where personalization instances already exist in LMSys dataset and understanding preference following there would have made the work very convincing.

---

> ### Author Response · Authors · 2024-11-20
> **Rebuttal Reply 1/n (with additional results and paper rewriting)**
>
> Dear reviewer Uqsa,
>
> Thank you for your constructive feedback! We appreciate that you recognize our work's strengths in terms of (1) valuable dataset contribution, (2) presenting insights across major models with various strategies for improving preference following and actionable insights from error analysis, and (3) addressing the important yet underrepresented topic of LLM personalization.
>
> We have conducted extensive additional analysis and experiments to address your concerns. Below, we respond to your specific concerns:
> > **W1: "subjective LLM-based evaluation lacks qualitative and manual validations despite very small manual checks on the responses with 5% error rate."**
> * We acknowledge this concern and have significantly expanded our human evaluation efforts beyond just preference-following accuracy. We conducted comprehensive manual validation of our LLM-based evaluator (Claude 3 Sonnet) by examining human-LLM agreement rates across four error type binary checkers and three preference forms. By sampling responses evenly from six models, five baselines, and all benchmarked turns, we manually validated 100 samples per preference form and calculated human-LLM agreement rates. For each response, we checked 4 error checkers, resulting in 1200 error checks. The results (shown in the table below and in Appendix A.17) reveal strong alignment between human and LLM judgments, with particularly high agreement rates in detecting helpful responses and hallucinations.
>
> *Table: Human and LLM Judge agreement rates across different error checker as well as the final preference following accuracy.*
> | Error Checker | Explicit Preference | Implicit Choice-based | Implicit Persona-driven |
> |--------------|--------------------:|---------------------:|----------------------:|
> | Violate Preference? | 0.92 | 0.86 | 0.95 |
> | Acknowledge Preference? | 0.88 | 0.90 | 0.97 |
> | Hallucinate Preference? | 0.98 | 0.96 | 0.92 |
> | Helpful Response? | 0.96 | 0.93 | 0.90 |
> | Overall Preference Following Accuracy | 0.97 | 0.92 | 0.96 |
>
> * To further address concerns about subjectivity, our benchmark also includes Multiple Choice Question tasks for each preference, which has a high correlation between performance with generation tasks (in Fig. 11). Our MCQ tasks provide an objective, deterministic evaluation metric that complements and validates our LLM-based evaluations, thereby strengthening the overall reliability of our benchmark.
>
> > **W2: “Inserting preferences within the LMSys dataset may be unnatural. what strategies were used for maintain semantic coherence of the dataset while introducing preferences and queries.”**
>
> We understand and agree with the concern about the unnaturalness of inserting preferences into the LMSys dataset when viewed as a single session multi-turn conversation. However , we would like to clarify our setting is multi-session, multi-turn conversations. We believe this also represents the  real-world chatbot interactions from the perspective of users shift between different topics across multiple sessions..
>
> In reality, it is hard for users to maintain a single-topic conversation over long context like 100K tokens. Therefore, to assess long context preference following, we designed our setup as a multi-session conversation. Each conversation is divided into distinct sessions, with each session focusing on a specific topic across multiple turns (please refer to our formalized problem setup in Sec 2.1 of our revised paper). For instance, a conversation might begin with 10 turns discussing a debugging challenge, shift to 2 turns expressing dining preferences while seeking restaurant recommendations, and later move to travel planning that incorporates the preference.
>
> During preference insertion, we ensure topic transitions occur only after completing a turn. For example, each preference is inserted as a complete session, where the session contains the same number of exchanges from the user and LLM. This makes preference revelation a natural part of the multi-session structure. By modelling dynamic, multi-session interactions, we capture how users naturally engage in diverse, topic-spanning conversations. This setup enables testing of the LLM's ability to maintain and apply user preferences across varied contexts.

---

> > ### Author Response · Authors · 2024-11-20
> > **Rebuttal Reply 2/n (with additional results and paper rewriting)**
> >
> > > **W3: “no analysis of existing datasets and benchmarks for personalization based benchmarking  like 'LaMP” (authors mention in Appendix, however an exhaustive personalization benchmark could cover some aspects of these other benchmarks in mentioned in Appendix).”**
> >
> > We agree with the need for a more extensive discussion of long-context recall and personalization-based benchmarks in the main paper. Due to page constraints, we initially provided these details in the appendix. In response to your suggestion, we have expanded the discussion in Appendix A.2 to cover personalization benchmarks more comprehensively, particularly highlighting recent works like LAMP \[1\] (which evaluates personalized task like movie tagging through explicit user profile conditioning), RPBench-Auto \[2\] (focusing on character consistency across personas), TIMECHARA \[3\] (addressing temporal consistency in character representation), and RoleLLM \[4\] (examining fine-grained role-playing across 100 diverse characters). This expanded analysis helps contextualize how our work differs by focusing on lifestyle-oriented preferences in long-context, multi-turn conversations, rather than single-turn tasks (like movie tagging, citation identification) or character-based role-playing. We have also added additional context in the main text in Section 4 to better highlight and integrate this analysis.
> >
> > > **W4: “This study emphasizes improvements with inserting 'Reminders' and other strategies like self-critique. This might skew model development more towards prompt engineering instead of focusing on understanding intrinsic improvements methods in memory and retrieval capabilities.”**
> >
> > * Our primary goal is to comprehensively evaluate state-of-the-art models, such as GPT4o, O1, and Claude 3.5, which are closed-source and do not permit gradient-based modifications. As such, we benchmark these models using various prompting methods, including widely-adopted techniques like few-shot CoT, self-correction and RAG, to establish meaningful comparisons.
> > * While the focus of our work is not directly on intrinsic model improvements, these prompting baselines provide valuable response data that can be used for future model improvements.
> > * In Section 3.7, we fine-tuned an open-source model, Mistral 7B on our dataset and demonstrated intrinsic improvements in preference-following capabilities, effectively bridging the gap between prompting strategies and intrinsic model advancements.
> >
> > > **Q1: “Shedding more light on some visualization of attention (and the lack of it) could have shed more light on understanding why the results degrade so much in different scenarios.”**
> >
> > Thank you for your insightful suggestion. We conducted an in-depth analysis of the model's attention patterns in Appendix 12 of the revised paper. We visualized the average attention scores of response tokens over input prompt tokens. We compared attention patterns between implicit and explicit preferences, and between pre and post-SFT models. The findings are summarized below:
> >
> > * **Implicit vs. Explicit Preferences:**
> >   While visualizing attention score patterns for implicit and explicit preferences, we observed no significant systematic differences in how the model distributes attention over preference-related tokens. The lack of disparity suggests that the performance degradation for implicit preferences may stem mostly from challenges in ***Preference Inference*** than ***Long-Context Retrieval*** capabilities as defined in Sec 2.1, where the ***Preference Inference*** means the capacity to accurately infer user preferences through dialogue, whether explicitly stated or implicitly revealed. We hypothesize ***Preference Inference*** has more complexity that likely involves deeper internal mechanisms beyond what attention score visualization can reveal.
> > * **Pre- vs. Post-SFT Comparison:**
> >   Motivated by the insight above, we proceed to compare attention score after supervised fine-tuning on the Mistral 7b model in Figure 20\. On explicit preference following, **it exhibited a consistent increase in attention scores allocated to preference-related tokens**, with up to ~5% increase in attention. This increase correlates with the observed improvements in preference-following capabilities, suggesting that attention scores can indeed reflect the model's ability in ***Long-Context Retrieval.***
> >
> > For more details on the attention visualization experiments, please refer to Appendix A.12.

---

> > > ### Author Response · Authors · 2024-11-20
> > > **Rebuttal Reply 3/n (with additional results and paper rewriting)**
> > >
> > > > **Q2: “..LMSYS- Chat-1M dataset, some of the models they test may have seen this dataset before during training \- what would be the impact of such exposure?”**
> > >
> > > This is a thoughtful consideration. However, the trained datasets of these closed-source LLMs are not publicly released, making direct overlap assessment impossible. Additionally, our evaluation primarily focuses on the preference-following aspects of conversations, which differ significantly from the general information exchange in LMSYS-Chat-1M's intermediate conversations, as success in objective QA does not necessarily translate to improved preference adherence. Furthermore, since we sampled conversations from the dataset, even if the LLMs were trained on it, the exact ordering of the conversation flow would differ, reducing potential memorization effects.
> > >
> > > > **Q3: Since this is a dataset contribution, some more clarity on understanding dynamic user preferences might have been useful. For example, mining instances where personalization instances already exist in LMSys dataset and understanding preference following there would have made the work very convincing.**
> > >
> > > * We have thoroughly examined the conversations sampled from the LMSYS-Chat-1M dataset and ensured that these are task-oriented (e.g., grammar corrections, factual queries, or general knowledge questions) without involving personal preferences, thereby eliminating potential conflicts related to dynamic user preferences.
> > > * In addition, we **conducted additional experiments to validate our findings on dynamic preferences,** as detailed in Section 3.6 of the main paper and Appendix A.16. We demonstrated that inserting multiple preferences and conflicting preference pairs improved preference-following performance for models such as Claude 3 Sonnet, Claude 3 Haiku, Mistral 8x7b, and Mistral 7b. These results confirm a positive correlation between the number of preferences and preference-following accuracy. Furthermore, our findings suggest that the impact of conflicting preferences is model-dependent, with conflicting preferences not necessarily harming performance.
> > >
> > > We greatly appreciate your feedback and believe that these new experiments and clarifications strengthen our paper. We hope this addresses your concerns and please let us know if you have any questions. Thank you!
> > >
> > >
> > >
> > >
> > >
> > > \[1\] A. Salemi, S. Mysore, M. Bendersky, and H. Zamani, "LaMP: When Large Language Models Meet Personalization," in Proc. of ACL, 2024\.
> > >
> > > \[2\] Boson AI, "RP-Bench," [https://boson.ai/rpbench/](https://boson.ai/rpbench/), 2024\.
> > >
> > > \[3\] J. Ahn, T. Lee, J. Lim, J.-H. Kim, S. Yun, H. Lee, and G. Kim, "TimeChara: Evaluating Point-in-Time Character Hallucination of Role-Playing Large Language Models," in Findings of ACL, 2024\.
> > >
> > > \[4\] N. Wang, Z. Peng, H. Que, J. Liu, W. Zhou, Y. Wu, H. Guo, R. Gan, Z. Ni, J. Yang, M. Zhang, Z. Zhang, W. Ouyang, K. Xu, W. Huang, J. Fu, and J. Peng, "RoleLLM: Benchmarking, Eliciting, and Enhancing Role-Playing Abilities of Large Language Models," in Findings of ACL, 2024\.

---

> > > > ### Author Response · Authors · 2024-11-28
> > > > **gentle reminder**
> > > >
> > > > Dear Reviewer Uqsa,
> > > >
> > > > Thanks again for your insightful feedback on our work! We've carefully addressed your comments with new experiments and clarifications, including attention score visualizations, extensive human validation across 1,200 error checks, clarification of our multi-session conversation setup, expanded related works on personalization benchmarks, and additional experiments on dynamic preferences.
> > > > We hope the new experiments and clarifications address your concerns. Could you please let us know if you have further questions? We would be happy to discuss further. Thank you!

---

### Author Response · Authors · 2024-11-27
**General Response (Summary of Rebuttal Discussion, Additional Experiments and Revisions)**

Dear Reviewers and ACs,

Thank you again for overseeing our submission and your thoughtful feedback! In this post, we want to summarize the rebuttal discussion and our key responses. We first thank the reviewers for recognizing our work as "one of the most comprehensive evaluations" with "exceedingly extensive experimental analysis" (d6T5), providing "actionable insights from error analysis" (Uqsa), and featuring "timely and interesting topic coverage" (L7qv), and "significance of our findings for real-world applications" (PRWG).

To address the concerns raised, we have expanded our work by: (1) detailing data construction process, (2) added extensive human validation of our evaluator, (3) clarifying our multi-session problem setup to address long-context coherence concerns, and (4) new experiments on attention score visualization, dynamic preferences and SFT. Below, we detail our responses:

**1: Added Comprehensive Data Construction Process** *(reviewers PRWG, L7qv, d6T5)*

We have added a new section (Appendix A.13) that thoroughly details our data generation process for all preferences forms. We aim to make the data construction as reproducible as possible. This includes the generation process, selection criteria, and validation procedures. Our dataset was curated through multi-stage filtering involving human annotators and LLM-assisted evaluations to ensure high quality and naturalness. We have included the prompts used for each evaluator and for generating the implicit preferences, and we will release all other detailed prompts in the repository upon release.

**2: Conducted More Manual Validation for LLM-based Evaluator** *(reviewers Uqsa, PRWG, d6T5)*

* **Expanded Human Evaluation Efforts**: We conducted a more comprehensive manual validations of our LLM-based evaluator across error types and preference forms, involving 1,200 error checks (in Appendix A.17). The results showed strong alignment between human judgments and LLM evaluations.
* **Our MCQ Task Provides Deterministic and Objective Evaluation**: In addition to generation tasks, our benchmark incorporated MCQ tasks, providing deterministic evaluation metrics that complement our LLM-based evaluations and is less computationally expensive.

**3: Semantic Coherence When Inserting Preferences** *(reviewers Uqsa, L7qv)*

We clarified that:

* **Our Setup Is Multi-Session, Multi-Turn Conversations**: Our setting is multi-session, multi-turn conversations. We believe this also represents the  real-world chatbot interactions from the perspective of users shift between different topics across multiple sessions. In reality, it is hard for users to maintain a single-topic conversation over long context like 100K tokens. Therefore, to assess long context preference following, we designed our setup as a multi-session conversation.
* **Preferences are inserted in a complete session**: During preference insertion, we ensure topic transitions occur only after completing a turn. For example, each preference is inserted as a complete session, where the session contains the same number of exchanges from the user and LLM. This makes preference revelation a natural part of the multi-session structure.
* **Formalizing Problem Setup**: To make our problem setup clearer, we have updated Section 2.1 with mathematical formalization to better define the relationships between conversations, sessions, turns, queries, and preferences.

**4: Additional Experiments on Attention Visualization, Dynamic Preferences, SFT and Lost-in-the-Middle Effect** *(reviewers Uqsa, L7qv, PRWG)*

* **Analyzing 'Lost in the Middle' Effect**: We conducted experiments (Appendix A.15) showing how the position of preferences within conversations affects model performance, corroborating the 'Lost in the Middle' phenomenon.
* **Visualizing Attention Patterns**: We performed in-depth analysis of attention score (Appendix A.12), comparing pre- and post-fine-tuning, and implicit and explicit preferences. We found that fine-tuning increases attention to preference-related tokens, providing mechanistic insight into why finetuning enhances preference-following capabilities.
* **Investigating Multiple and Conflicting Preferences**: Additional experiments (Appendix A.16) were conducted to examine these effect across different models and the new results follow our previous conclusion.
* **Fine-Tuning for Implicit Preferences**: We extended our fine-tuning experiments (Appendix A.14), demonstrating that training on explicit preferences generalizes effectively to implicit ones.

**Additional Revision:**

* **Ethics Statement** has been added.
* **Expanded Related Works**:  We have clarified baseline terminologies and included additional related works.

We sincerely appreciate your constructive suggestions, which helped strengthen our paper a lot. We are happy to provide additional clarifications. Thank you for your time!

---

### Meta-Review · Area_Chair_FGWo · 2024-12-18

**Metareview:**

This paper presents a new dataset PrefEval that contains multi-turn conversational data in which users express specific preferences in context, for example "I always avoid floral patterns."  The authors detail the construction of this dataset and evaluate the ability of multiple state of the art LLMs in being able to learn these conversationally expressed preferences in context and incorporate this knowledge into their recommendations.

The authors provide extensive evaluation and instructions on how to create the dataset including releasing the prompts and they say that they plan to release the dataset at future date, but have not done so at this point.

The paper is clear, well written and represents a contribution to the ICLR community.  Hopefully the dataset will be released soon, but the process of constructing it is clear and perhaps others may simply want to use this process to construct their own multi-turn datasets tailored to their needs.  Being able to evaluate individual user preferences in "real conversational time" is essential to many recommendation chatbots that respect customer privacy and are therefore faced with a "cold start" personalization problem with every customer conversation.

**Additional Comments On Reviewer Discussion:**

The reviewer discussion was engaged and positive.  Two reviewers increased heir scores during the rebuttal period.  All reviewer seem enthusiastic about the paper.

---

### Decision · Program_Chairs · 2025-01-22

Accept (Oral)